# Chaperone biomarkers of lifespan and penetrance track the dosages of many other proteins

Nikolay Burnaevskiy [1], Bryan Sands[1], Soo Yun[1], Patricia M. Tedesco[2], Thomas E. Johnson[2], Matt Kaeberlein[1], Roger Brent [3,4]* & Alexander Mendenhall [1,4]*

Many traits vary among isogenic individuals in homogeneous environments. In microbes, plants and animals, variation in the protein chaperone system affects many such traits. In the animal model *C. elegans*, the expression level of *hsp-16.2* chaperone biomarkers correlates with or predicts the penetrance of mutations and lifespan after heat shock. But the physiological mechanisms causing cells to express different amounts of the biomarker were unknown. Here, we used an in vivo microscopy approach to dissect different contributions to cell-to-cell variation in *hsp-16.2* expression in the intestines of young adult animals, which generate the most lifespan predicting signal. While we detected both cell autonomous intrinsic noise and signaling noise, we found both contributions were relatively unimportant. The major contributor to cell-to-cell variation in biomarker expression was general differences in protein dosage. The *hsp-16.2* biomarker reveals states of high or low effective dosage for many genes.

[1] Department of Pathology, University of Washington, Seattle, WA, USA. [2] Department of Integrative Physiology, University of Colorado, Boulder, CO, USA. [3] Division of Basic Sciences, Fred Hutchinson Cancer Research Center, Seattle, WA, USA. [4] These authors jointly supervised this work: Roger Brent, Alexander Mendenhall. *email: rbrent@fredhutch.org; alexworm@uw.edu

Genes and environments do not explain all the differences in heritable traits. Experiments with isogenic model systems kept in homogeneous environments have clearly demonstrated this[1]. Variation in traits among isogenic animals was perhaps first noticed in 1925 in *Drosophila funeberis* by Romaschoff when he noticed that not all animals in a "pure bloodline" (inbred strain) exhibited the mutant phenotype for *Abdomen abnormalis*; this is called incomplete penetrance. The severity of the phenotype was different as well, a phenomenon that came to be termed expressivity[2]. Also in 1925, Timofeeff-Resskovsky noticed that the penetrance and expressivity of another mutation, *Radius incompletes*, was altered in different genetic backgrounds. Thus, the action of genes could affect the penetrance and expressivity of discreet traits conferred by other genes[3]. Complex traits such as lifespan also vary among isogenic individuals, both in model systems[4] and in human monozygotic twins[5]. The mechanisms underlying this variation remain poorly understood.

Experimental and natural variation of genes in the protein chaperone system alters the manifestation of complex and discreet traits. In the late 1990s and 2000s scientists found that perturbing the chaperone system had broad effects on variation in traits, including the penetrance of some mutations. Scientists used geldanamycin to perturb the N-terminus of HSP90, a chaperone with important specific clients, and a master regulator of the heat-shock response[6]. They found that the general variation in traits is affected in different genetic backgrounds of *Drosophila melanogaster*[7] and *Arabidopsis thalania*[8]. Another group found chaperone-altering geldanamycin treatment alters penetrance and expressivity of specific mutations in the vertebrate *Danio rerio*[9]. In *Caenorhabditis elegans*, which is immune to the effects of geldanamycin, differences in levels of *hsp-90* reporter gene expression levels predicted differences in the penetrance of loss of function mutations[10,11].

In previous work, we explored the consequences of differences in expression of another chaperone on expression of genetic traits. In *C. elegans*, the *hsp-16.2* reporter gene is expressed only after heat shock. We found that adult animals that make more of the *hsp-16.2* reporter gene have differences in complex traits—lifespan and lethal thermal stress tolerance in *C. elegans*[12,13]. Using the another construct of the *hsp-16.2* reporter in *C. elegans* (the same promoter fused to fluorescent protein, inserted elsewhere in the genome), another group found that increased *hsp-16.2* reporter expression was associated with differences in the penetrance of a number of hypomorphic point mutations in distinct types of genes[10]. For the most part, these *hsp-16.2* and *hsp-90* reporter gene biomarkers correlated with the penetrance of distinct mutations, but both correlated with penetrance of at least one mutation, *lin-31*[10], indicating that there are both distinct and overlapping fractions of the proteome affected by these chaperones. As expected, but important to note, both *hsp-90*[11] and *hsp-16.2*[12] reporters properly correlate with the expression levels of the chaperones on which they are reporting (*hsp-16.2* & *hsp-90*).

We previously showed that differences in expression levels of *hsp-16.2* lifespan/penetrance biomarkers in adult animals were likely due to differences in transcription; notably, this did not include *hsp-90*. We also identified genes[14] and environmental variables[15] that affected the amount of interindividual variation in the *hsp-16.2* reporter. Yet, we did not know how the cells of animals came to express more or less of this lifespan/penetrance biomarker. Therefore, we set out to dissect the mechanisms of cell-to-cell variation in gene expression to understand how differences in the expression of *hsp-16.2* lifespan/penetrance biomarker arise. We focused on gene expression in the intestine cells of adult animals because that is the tissue that makes the most signal for *hsp-16.2* reporters[13,16,17], because it is the point in life

we used to predict lifespan and thermotolerance[12,13], and because we had developed technical methods for in vivo reproducible quantification of gene expression in single intestine cells[16].

Here, we adapted and extended an experimental design and analytical framework we developed in yeast[18], to quantify sources of variation in gene expression in a metazoan. This analytical framework is an expansion of the intrinsic/extrinsic noise framework pioneered in *E. coli*[19]. The concept of intrinsic noise in gene expression arose from measuring the correlation between two differently colored but otherwise identical reporter genes in work in prokaryotic bacteria by Elowitz and Swain[19]. This approach for quantifying intrinsic noise was then adapted by Raser and O'Shea to quantify variation in the expression of differently colored alleles in eukaryotic single celled yeast[20]. Our expanded experimental and analytical framework allowed us to distinguish among three hypotheses for how differences in *hsp-16.2* reporter expression might arise. The three hypotheses were that the differences in biomarker expression level arose from intrinsic noise, signaling noise or differences in general protein expression capacity. The first hypothesis was that differences in the *hsp-16.2* lifespan biomarker might arise from differences in intrinsic noise in gene expression. Previous work with human autosomal genes[21] showed that individual cells may only express much less of, or only one, of their two distinct copies of each allele. Therefore, animals might express more or less of a gene by expressing different amounts of each allele—anywhere from full expression of both alleles to no expression of either allele.

Our second hypothesis was that differences in the *hsp-16.2* reporter and associated chaperones might be due to differences specific to the signaling pathway that activated chaperone expression. That is, we hypothesized we would see relatively high covariation for expression of the *hsp-16.2* reporter gene and other chaperone reporters like *hsp-90*, but not covariation with non-heat-shock pathway reporters like a yolk-protein reporter.

Finally, our third hypothesis was that elevated chaperone levels would increase general protein dosage, by affecting expression capacity or protein turnover. Chaperones have many clients and might affect folding, maintenance and turnover of these proteins[6], thereby altering their effective dosage. This idea was appealing, since, work from the Kim Lab using *C. elegans* found that several distinct non-chaperone reporter genes could predict lifespan, and that these distinct reporters were highly correlated[22]. Moreover, work by us in *S. cerevisiae* had shown that these general effects on protein dosage[18] are important contributors to extrinsic noise in gene expression in *E. coli*[19] and *S. cerevisiae*[20]. A detailed description of this analytical framework is given in Supplementary Notes 1–5. Cartoon images of what cell autonomous and cell nonautonomous differences in each of the three aforementioned categories (intrinsic noise, signaling noise, and general protein dosage/protein expression capacity) are shown as Supplementary Figs. 1–3.

Below, we detail results showing that, for *hsp-16.2* reporter expression in the adult worm intestine, two components of cell-to-cell variation are minimal. The other component, differences in protein dosage, accounts for the majority of variation in gene expression in intestine cells. We provide experimental evidence that shows how differences in this component may arise after heat shock in the context of a working model integrating data from this and other reports, and suggest how these differences might account for observed effects on expressivity and penetrance of different alleles.

## Results
### The adapted analytical framework and experimental design.
Here we adapted an approach we used in yeast[18], wherein we compared the outputs of two differently colored (different

fluorescent proteins) versions of the same reporter gene expressed from two identical loci on homologous chromosomes (Type I experiments) or the outputs of two different reporter genes (Type II experiments), shown in Fig. 1. Measuring the pairs of reporter genes in these two different kinds of experiments allows us to quantify the amount of cell-to-cell variation in gene expression attributable to three distinct, experimentally tractable bins, each with distinct underlying molecular causes. Figure 1 shows how we can use this reporter gene measurement scheme to quantify differences in intrinsic noise[19,23] ($\eta^2(\gamma)$), signaling through different pathways[18] ($\eta^2(P)$), or general protein expression capacity[18], a measure of general protein abundance ($\eta^2(G)$). See Supplementary Notes 1–5 and Supplementary Figs. 1–3 for additional explanation and cartoon depictions of different kinds of variation in gene expression.

**Type I experiments quantify intrinsic noise**. Type I experiments allow us to measure how much cell-to-cell variation there is in apparently stochastic to-promoter or to-transcript binding events, or allele access. Type I experiments reveal how much cell-to-cell variation there is in allele expression/access as uncorrelated variation, often referred to as intrinsic noise ($\eta^2(\gamma)$). Figure 1 shows an example of a scatterplot for the same cell type (same lineage/fate) measured from many different animals from one of these experiments. If there were a lot of uncorrelated variation (a cloud like appearance of points) it would mean that there were significant cell-to-cell differences in to-promoter binding, to-transcript binding, or allele access. If there was little uncorrelated variation (alignment of points along the major axis; Fig. 1), it would mean that most cells would have had access to, and expressed, both alleles. It would also mean that there were few differences attributable to probabilistic differences in biochemical binding events (to-promoter/to-transcript) that tend to dominate when there are small numbers of molecules involved[23]; we would say that intrinsic noise was low. For additional details, please refer to Supplementary Note 4.

**Intrinsic noise is relatively constrained**. In Type I experiments for *hsp-16.2*, and for all the other reporter genes we measured, intrinsic noise was low. The expression level of one allele accounted for 90% or more of the expression of the other allele in animals. Figure 2 shows the average intrinsic noise levels for a few genes we measured. Intrinsic noise was a minor component of the cell-to-cell variation, and relatively constrained compared with yeast[20]. We show individual Type I scatterplots and boxplots organized by cell type and experiment in Supplementary Figs. 4, 5. A few relatively deviant cells can be seen off the trend line in Supplementary Fig. 4, which shows the same cell measured from many different animals across multiple experiments, demonstrating that this experimental system is capable of detecting intrinsic noise in gene expression at the protein level. However, large differences in allele bias simply do not occur at a high frequency in the intestine cells, also shown by Supplementary Fig. 4.

**Quantifying variation in signaling and protein dosages**. Type II experiments compare the expression levels from two distinct genes to tell us about cell-to-cell differences in signaling through different pathways ($\eta^2(P)$). In the scatterplot from a Type II experiment shown in Fig. 1, the uncorrelated variation (dispersion across the major axis) is a measure of cell-to-cell differences in signaling through distinct pathways. We subtract the average intrinsic noise ($\eta^2(\gamma)$) for the gene pairs we are measuring, so we know that the remaining dispersion is due to signaling/pathway noise ($\eta^2(P)$) (shown in Fig. 1). For additional details, please also

refer to Supplementary Fig. 2 and Supplementary Note 5. Type II experiments also quantify how much cell-to-cell variation is attributable to differences in protein expression capacity ($\eta^2(G)$), which we define as the ability to express and maintain proteins. If a cell is able to produce relatively more protein from two or more distinctly regulated genes, compared with other cells in which the same proteins are measured, then we observe that cell as having relatively high protein expression capacity. In the scatterplot of a Type II experiment on the bottom right of Fig. 1, the correlated variation (dispersion along the major axis) is a measure of ($\eta^2(G)$). For additional details, please also refer to Supplementary Fig. 3 and Supplementary Note 3.

**Stoichiometry and signaling noise are constrained**. In Type II experiments, we found that cell fate (which ring a cell was in) was the primary determinant of a rigid cell-fate-specific ratiometric setpoint for any given gene pair. Supplementary Fig. 6 shows scatterplots demonstrating how grouping all the cells together artificially inflates signaling noise, compared with grouping cells by ring. So, we grouped cells we measured by the ring from which they came to prevent artificial inflation of ($\eta^2(P)$). For the most part, we found that there was little cell-to-cell (and thus, animal-to-animal) relative deviation in signaling through distinct pathways to activate distinct genes.

Figure 3 shows that there was little signaling noise at the cell level ($\eta^2(P)$), relative to intrinsic noise. However, we did notice that some pathways seemed to have more noise; for example there appears to be more signaling noise through *vit-2* than through the *hsp-16.2* pathway. Figure 3 also shows there is relatively more signaling noise between *hsp-17* and *mtl-2*. This is just not a feature of most reporter pairs we examined. Furthermore, we observed entire populations of animals signal properly. The *hsp-16.2* reporters only expressed after a heat shock and the yolk-protein reporters only expressed in intestine cells at the onset of reproductive maturity. For another example of proper signaling, Supplementary Fig. 7 shows an animal resolution experiment demonstrating that, in response to heat shock, animals' intestine tissue will upregulate eukaryotic translation elongation factor 1 alpha (*eft-3/eef-1A*), and downregulate yolk-protein production (*vit-2*). Supplementary Figs. 8–10 show individual scatterplots for Type II experiments and boxplots of individual cell types. Supplementary Fig. 9 also shows that some cells have relatively more or less noise through some pathways, such as higher signaling noise observed in the intestine cells of ring one (for signaling noise examples see Supplementary Fig. 9 *hsp-16.2 & mtl-2* ring one, and *hsp-17 & vit-2* ring one). Thus, this experimental system is capable of detecting differences in signaling at the cell and population levels, we just did not detect a large degree of signaling noise in young adult intestine cells.

**Cells vary most in general protein expression capacity**. The large ($\eta^2(G)$) components in Fig. 3 shows that we found that most of the variation in gene expression is attributable to cell-to-cell (and thus, animal-to-animal) differences in protein expression capacity ($\eta^2(G)$). We show individual Type II scatterplots and boxplots organized by cell type and experiment in Supplementary Figs. 8–10. We then investigated these results further.

**Protein expression capacity with fusion proteins**. These intestine cells express more protein from other, distinctly regulated genes, even if they are on other chromosomes, or fused to other genes (*hsp-16.2* reporter covaries with Emerin::GFP; Fig. 4a). We wanted to confirm our findings with additional fusion proteins. We also wanted to measure fusion proteins that would not have

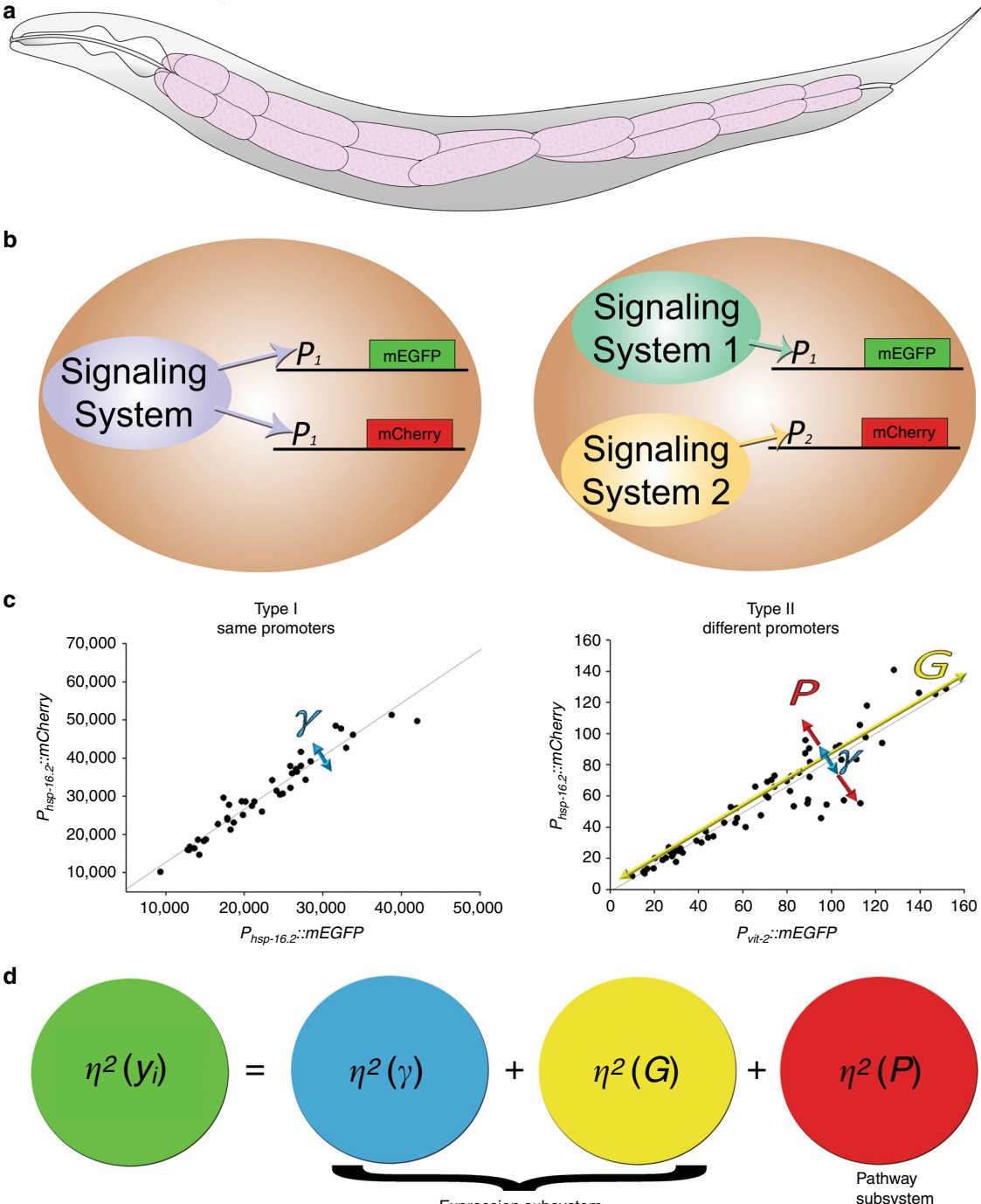

**Fig. 1 Experimental design and analytical framework. a** Schematic view of the *C. elegans* body. Intestine is colored pink. The twenty cells of the intestine are organized into nine segments, called rings. **b** Overview of reporter gene measurement scheme. In the left panel, two differently colored copies of the same gene respond to the same signaling system (Type I experiments), or, in the right panel two differently colored reporters controlled by two different promoters respond to two distinct signaling systems (Type II experiments). **c** Data collection schematic. We imaged intestine cells in rings one through four (the cells that fit in a single field of view with a ×40 objective) in animals on day two of adulthood, at the point when lifespan can be predicted from $P_{hsp-16.2}$::*gfp* expression levels. We then plotted expression levels of reporter pairs expressed in the intestine cells, grouped by cell type; the same cell types (e.g., cells in ring three; int3 cells) from different animals are plotted. Left scatterplot shows typical results of type I experiment. Since both genes receive the signals from the same upstream regulators and share the same downstream expression machinery, uncorrelated variation of expression results only from stochastic noise of transcription/translation or variable allele access $\eta^2(\gamma)$. Right scatterplot shows typical results of type II experiment. In the case of two different *genes*, uncorrelated variation of expression can result from stochastic noise of transcription/translation or variable allele access $\eta^2(\gamma)$, and variation in activation of particular signaling pathways $\eta^2(P)$. Correlated variation results from variation in shared gene expression machinery $\eta^2(G)$. **d** A simplified version of the analytical framework derived from Colman-Lerner et al 2005 is shown. It depicts the three experimentally tractable bins into which we can attribute cell-to-cell variation in gene expression. Each bin has different underlying molecular causes and is colored differently; $\eta^2(\gamma)$ is red, $\eta^2(P)$ is blue, and $\eta^2(G)$ is yellow. A complete, mathematically detailed description of the analytical framework is shown in Supplementary Notes 1–5.

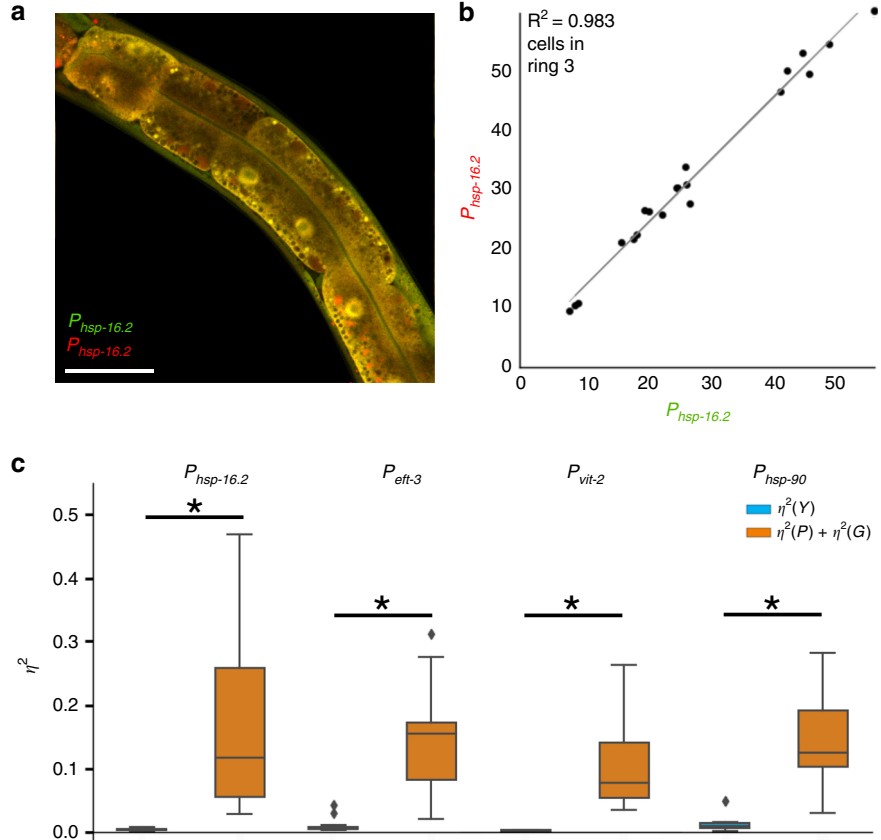

**Fig. 2 Type I experiments reveal low intrinsic noise. a** An example of the animal producing fluorescent reporters (mEGFP and mCherry) from two identical promoters ($P_{hsp-16.2}$); white scale bar on bottom right is 50 μm. We did not observe patchwork of green and red cells. Cells were nearly uniformly yellow, indicating low intrinsic noise. **b** Scatterplot of $P_{hsp-16.2}$::GFP and $P_{hsp-16.2}$::mCherry expression in cells in ring three of the intestine from ten animals examined in one experiment. **c**. Boxplots show correlated and uncorrelated variations for $P_{hsp-16.2}$, $P_{eft-3}$, $P_{vit-2}$, and $P_{hsp-90}$ in the intestine cells in rings 1–4. y axis is unitless $\eta^2$; see Supplementary Note 4 for a detailed description of how to calculate intrinsic noise. The boundary of the box closest to zero indicates the 25th percentile, a line within the box marks the median, and the boundary of the box farthest from zero indicates the 75th percentile. Whiskers above and below the box indicate the 90th and 10th percentiles. Outliers are shown as diamonds. In type I experiment uncorrelated variation arises from stochastic noise of transcription/translation or variable allele access—$\eta^2(\gamma)$ (colored red as in Fig. 1); correlated variation is a combined result of variation in gene expression capacity $\eta^2(G)$ and variation in pathway activation $\eta^2(P)$ (colored orange to represent a combination of the two terms from Fig. 1). Asterisks indicate statistical significance of $p < 0.05$ by two-tailed $t$-test. For each pair of alleles, data are from three independent experiments measuring intestine cells from thirty animals in three independent experiments (intestine cells from ten animals were measured in each experiment for each allele pair).

signal convoluted by abstract spatial patterns and auto-fluorescence. Therefore, we chose to quantify proteins that localized to the relatively low autofluorescence, and entirely segmentable, nucleus. We quantified a His2B knockin in conjunction with Emerin and Lamin fusion proteins in the nuclei of animals' intestine cells. We found that there was significant covariation of these proteins among the cells of animals, with the major axis of variation being the general abundance, shown in a 3D scatterplot in Fig. 4b; additional experiments shown in Supplementary Fig. 11.

**Protein expression capacity at animal resolution**. To confirm that the differences in general protein dosage we saw at cell resolution constitute the major axis of variation in gene expression, we examined large populations of animals. These were not straightforward experiments.

When we tried to quantify covariation between signals in unconstrained animals in the COPAS Biosort, or in a micro-fluidic, image-based quantification device we developed[24], we found little covariation between genes we know to be highly correlated inside individual intestine cells. This fact was due to

the rotational freedom and variable angular positioning of the animals along their long axes with respect to the detectors. This artefact allowed the higher throughput devices to get the right answer for the average expression level of the two different reporters, but not to detect covariation within a tissue. To circumvent these problems, we developed means to mount large numbers of animals in soft agarose pads, anesthetizing them and then imaging them, constrained in the agarose, using an objective with a large field of view. These measurements used the same optics used in our microfluidic imaging instrument, but revealed significant covariation between reporters, attributable to the simple fact that they were immobilized and flattened on agarose pads.

When we examined distinctly regulated reporters in hundreds of animals, we found that animal-to-animal variation in $G$ was dominant. In Fig. 4c, d, the same trend for differences in $G$ can be seen, even at animal resolution, provided animals are constrained and a tissue specific marker is used to extract relatively purer signal from a focused region of interest (the intestine tissue). Signals from other tissues differentially contributed to the intestine signal we measured using the

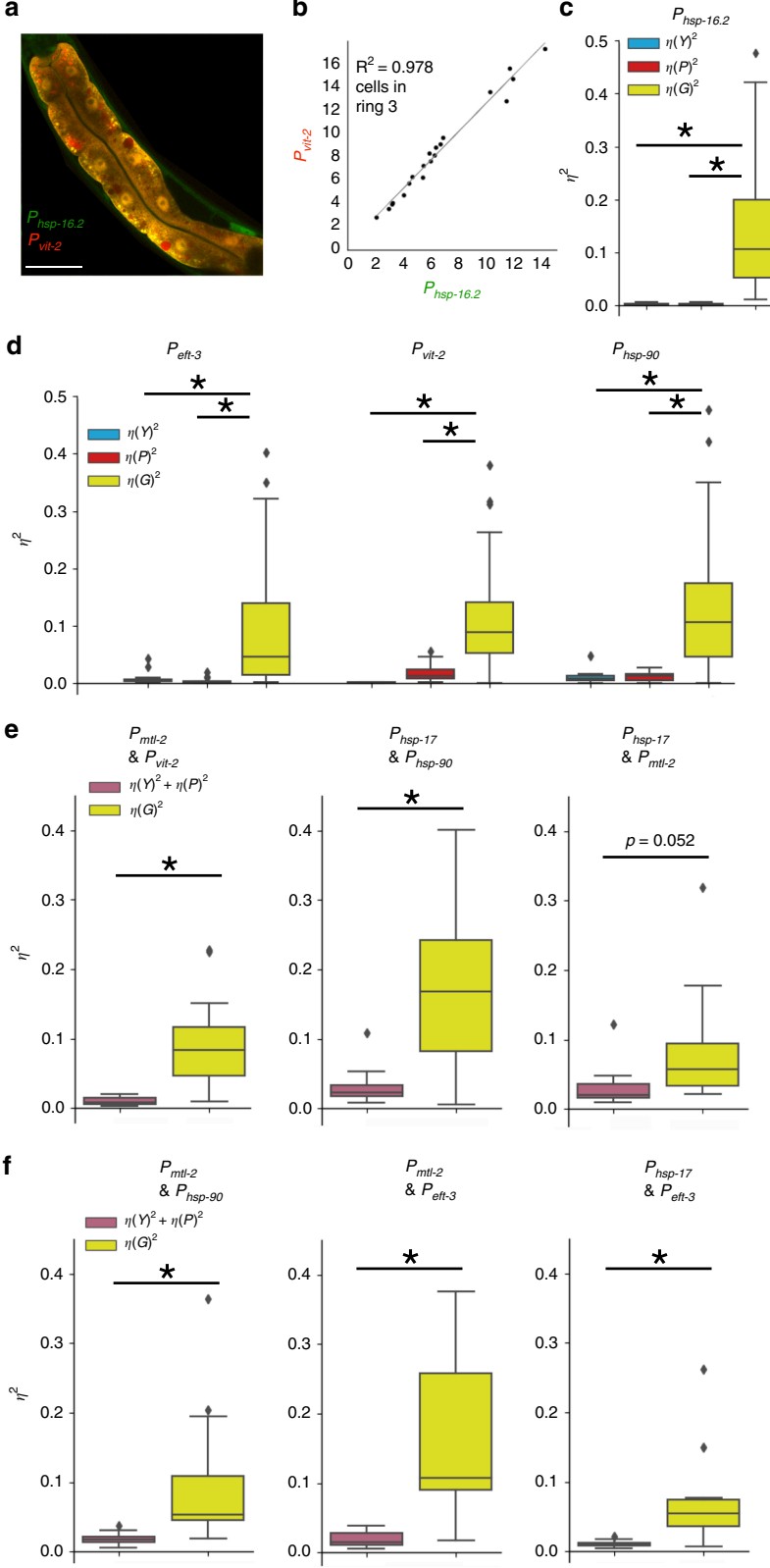

epifluorescent scope method to examine larger populations of animals. We believe the deviations from the trend line with the *eft-3* and *vit-2* reporters are from out of plane non-intestinal signals contributing to the intestine signal, and remaining rotational differences in mounting. We think the *vit-2* and *hsp-16.2* reporters are more representative of what we see at the cell level because both reporters are mostly expressed in the intestine (90% for *hsp-16.2* reporters in adults[16,17] and exclusively in the intestine for *vit-2*), whereas *eft-3* reporter signal is also relatively stronger in muscle and hypodermal tissues surrounding the intestine, compared with *hsp-16.2* reporter signal. To test the idea that the uncorrelated variation was attributable to out of plane signal from other tissue, we analyzed the cell resolution expression data from the 193 cells

**Fig. 3 Type II experiments reveal low signaling noise and high variation in protein expression capacity.** Boxplots are plotted as described in Fig. 2. **a** An example of the animal producing fluorescent reporters (mEGFP and mCherry) from two different promoters ($P_{hsp-16.2}$ and $P_{vit-2}$); white scale bar on bottom right is 50 μm. **b** Scatterplot of $P_{hsp-16.2}$::GFP and $P_{vit-2}$::mCherry expression in the intestine cells in ring three. **c** Average stochastic noise $\eta^2(\gamma)$ (red), variation in pathway activation $\eta^2(P)$ (blue) and variation in protein expression capacity $\eta^2(G)$ (yellow) for expression from $P_{hsp-16.2}$ in cells in intestine rings 1–4. y axis is unitless $\eta^2$; see Supplementary Note 1–5 for a detailed description of the analytical framework used here. Asterisks indicate statistical significance of $p < 0.05$ analyzed by one-way ANOVA with post hoc Tukey's HSD test. **d** Average stochastic noise $\eta^2(\gamma)$, variation in pathway activation $\eta^2(P)$ and variation in protein expression capacity $\eta^2(G)$ for $P_{vit-2}$, $P_{eft-3}$, $P_{daf-21}$ in the intestine cells in rings 1–4 (cells pooled together). Asterisks indicate statistical significance of $p < 0.05$ analyzed by one-way ANOVA with post hoc Tukey's test. **e, f** Average correlated and uncorrelated variations for different promoters couples in the intestine cells in rings 1–4. Uncorrelated variation combines stochastic noise of transcription/translation or variable allele access $-\eta^2(\gamma)$ and variation in pathway activation $\eta^2(P)$ (colored purple to represent the combination of the two terms from Fig. 1). Correlated variation results from variation in protein expression capacity $\eta^2(G)$. Asterisks indicate statistical significance of $p < 0.05$ by two-tailed $t$-test. For each pair of reporter genes, for $\eta^2(\gamma)$ and $\eta^2(G)$, data are from three independent experiments measuring cells from ten animals per experiment for a total of thirty animals in each group. For each explicit, ungrouped $\eta^2(P)$, data are from nine independent experiments measuring each pair of reporters from cells in ten individual animals in each individual experiment; see Supplementary Note 5 to calculate $A$ values to determine $\eta^2(P)$. Cells from 450 total animals, from 45 independent experiments (measuring a pair of reporter genes in ten animals in each experiment) were used to make the plots in this figure.

we measured from animals expressing the same *vit-2* and *eft-3* reporters and found the correlation to be $r = 0.927$.

**Protein expression capacity in images and a global analysis.** Figure 4e shows ($\eta^2(G)$) visually; two different types of animals expressing two distinct sets of reporter genes from Type II experiments are arranged from dimmest to brightest. Note that the patterns of expression are maintained but that the expression level of both reporters is different between animals; to observe this phenomenon, there must be little intrinsic noise ($\eta^2(P)$) and little signaling noise ($\eta^2(\gamma)$). Figure 4f shows that bright cells come from bright animals, which we previously observed with *hsp-16.2* alone[16]. Figure 4g shows that when all cells from all Type I and Type II experiments are normalized and plotted together, the Type II experiment data points surround the Type I experiment data points, indicating that there is general signaling noise beyond the intrinsic noise limits we measured. The large axis of correlated variation comprises the dominant axis of variation in protein expression among adult intestine cells.

**Protein expression capacity and production vs. turnover.** We wanted to test the hypotheses that high gene expression capacity was due to either better protein production or better maintenance/decreased turnover. Therefore, we tested to see if animals that made more protein did so by increased production or decreased turnover in both normal and heat shocked conditions. We quantified the age of protein in vivo using a fluorescent timer protein that matures from green to red in about 48 h[25]. We used whole animal image cytometry to quantify the fractions of relatively young and old protein in individual animals, detailed in Supplementary Materials and Methods. We controlled the reporter with the *eft-3* promoter, which is both constitutive and most highly correlated with both *hsp-16.2* and *hsp-90* chaperone reporters. If animals that contained higher concentrations of $P_{eft-3}$::timer did so because of increased synthesis of new protein, then they would have a higher ratio of new (green signal) to old (red signal) timer protein. If animals that had higher concentrations of $P_{eft-3}$::timer did so because of decreased protein turnover, then they would have a relatively higher ratio of old protein (red) to new protein (green).

Supplementary Fig. 12 shows that this timer protein does indeed detect a higher proportion of relatively older protein in relatively older animals; it works as advertised. We found that animals that express more or less reporter have similar ratios of both new and old protein (e.g., exactly 2.71 new to old ratio in both the top and bottom 10% of heat shocked animals). Supplementary Fig. 12 shows the trend line is linear and the

animals at the top and bottom do not deviate. Consistent with the dual roles of chaperones in protein production and maintenance, we find that animals that express more of the timer reporter are better at both protein production and maintenance to a greater degree than dimmer animals.

**Additional related data.** As previously reported[10,11,26], we also found that chaperone or chaperone-related reporters can correlate with or predict biological outcomes. Prior work showed that higher expression of chaperones is correlated with the effective activities of loss of function mutations[10], because these hypomorphic alleles were less penetrant in animals that expressed more *hsp-16.2* chaperone biomarker reporter gene. Expressing more *hsp-16.2* chaperone biomarker indicates these high biomarker expressing animals have more of some types of actual chaperones[12,27]. This suggests that gain of function mutations would also sometimes have higher activities with higher abundance of some chaperones. This work thus extends previous work with loss of function mutations to include alleles that conferred gains of function. In Supplementary Note 6: Additional Correlations between Phenotypes and Reporter Genes, we show that adult expression of the *hsp-16.2* biomarker correlates with the penetrance of a Ras gain of function mutation that acted during larval development in Supplementary Figs. 13, 14. We also show that the expression of the highly chaperone-correlated *eft-3* reporter in L1 larvae predicts growth to adulthood on neomycin in animals also expressing a NeoR gene (gaining the function of neomycin resistance) in Supplementary Fig. 15. In Supplementary Note 7: Persistence of States, we examine evidence of persistence, finding that the embryonic state of high yolk load or the L1 larval state of high protein expression capacity (discerned via *eft-3* reporter levels) are not correlated with adult gene expression levels, shown in Supplementary Fig. 16. In Supplementary Fig. 17 we show that intestine cells in larvae show differences in $G$, but also have seemingly autonomous bursts of protein expression, which we do not see in 2-day old adults. Developmental variation in gene expression will require further study to understand how these bursts fit in the context of developmental progression and other kinds of noise in gene expression.

**Discussion**
The natural variation in the *C. elegans* chaperone system is a property that is consequential, and likely selected for over geological time, if we subscribe to the prior conclusions of Klaus Gartner when he was pondering the origins of nongenetic variation in murine systems[28] it seems reasonable to do so. In our working model, animals with higher concentrations of the

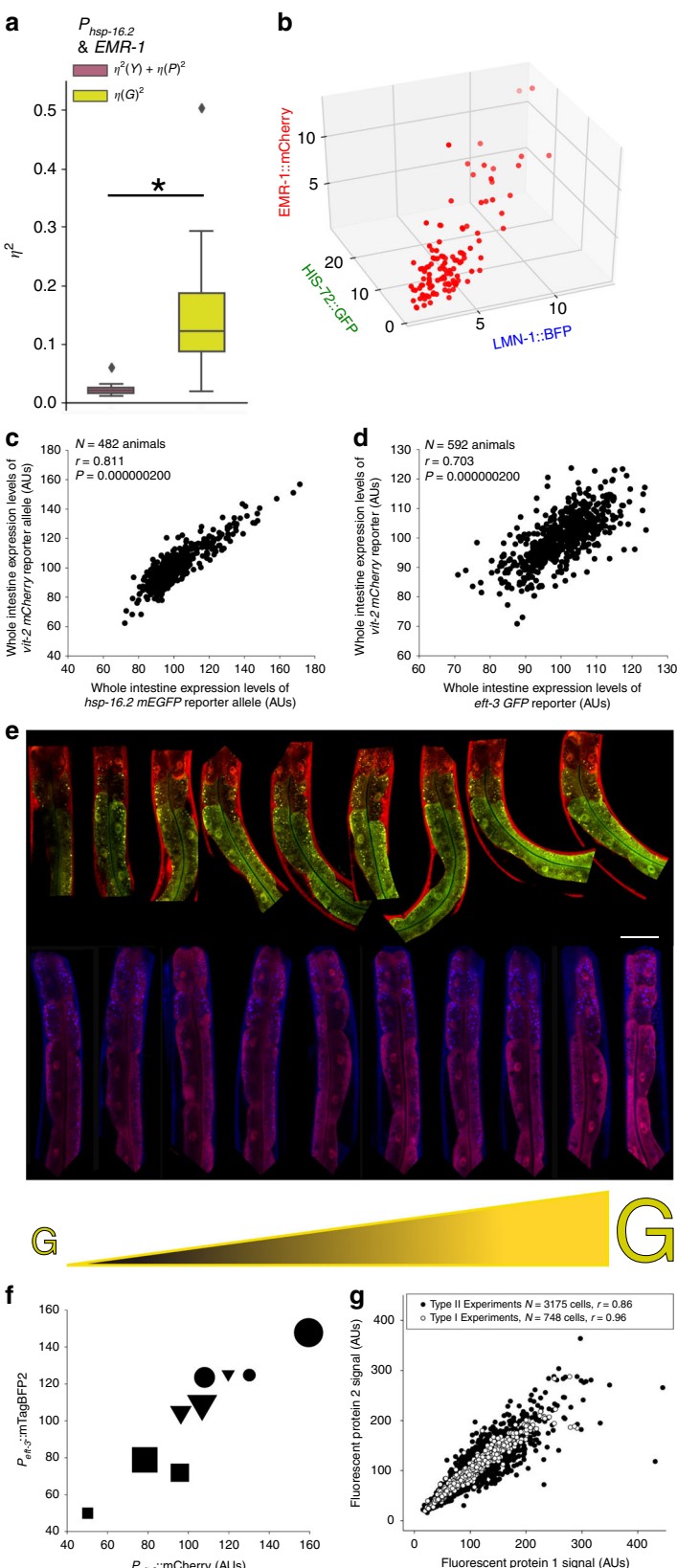

*hsp-16.2* lifespan/penetrance biomarker in their intestine cells have higher concentrations of other proteins in those cells.

In this model, differences might arise from epigenetic differences[27] and seemingly stochastic environmental perception differences[14,15]. In support of this idea, we previously found

that animals whose parents had been heat shocked express more chaperone biomarker[27]. Moreover, we also previously found that thermosensory neuron function is required for generating biomarker differences between individuals after heat shock[14]. In addition, we previously found that animals at

**Fig. 4 Protein expression capacity is the major axis of variation in gene expression in intestine cells. a** Boxplots of the amount of different types of variation for $P_{hsp-16.2}$::mCherry reporter and $P_{EMR-1}$::EMR-1:GFP in intestine cells; boxplots are plotted as described in Fig. 2. Here, $\eta^2(\gamma)$ and $\eta^2(P)$ are combined and colored purple; $\eta^2(G)$ is yellow as described in Fig. 1. y axis is unitless $\eta^2$. A two-tailed $t$-test between combined $\eta^2(\gamma)$ and $\eta^2(P)$ vs $\eta^2(G)$ produced $p < 0.05$; data are from three independent experiments measuring cells from thirty animals (cells from ten animals in each independent experiments). **b** 3-dimensional scatterplot of expression levels of LMN-1::BFP, HIS-72::GFP, and EMR-1::mCherry in intestine cells. Data are from all intestine cells measured in the field of view for each of ten animals in one experiment; additional experiments and 2D scatterplots are shown in Supplementary Fig. 11. **c** Scatterplot of $P_{hsp-16.2}$::GFP and $P_{vit-2}$::mCherry expression at whole animal level from three independent experiments. **d** Scatterplot of $P_{eft-3}$::mTagBFP2 and $P_{vit-2}$::mCherry expression at whole animal level compiled from three independent experiments. In (**d**) and (**e**), $r$ values shown are Pearson's correlation coefficients and Ns are listed in each panel. Panel (**e**) shows images of animals from individual experiments quantifying $P_{mtl-2}$::GFP and $P_{daf-21}$::mCherry (top panel), or, $P_{eft-3}$::BFP and $P_{vit-2}$::mCherry (bottom panel). Animals are arranged from dimmest (left) to brightest (right); white scale bar on bottom right of top panel is 50 μm. Panel (**f**) shows a scatterplot of average expression levels for cells in each ring of individual animals from the bottom row of panel (**e**). Different animals are represented by shapes (circle (brightest), triangle (median), square (dimmest)). Different sized shapes indicate the ring of cells each is representing; ring one cells are the smallest, ring two cells are the medium sized, and ring three cells are the biggest. Panel (**g**) shows a scatterplot of all cells from all animals from all Type I (120 animals; white dots) and II experiments (480 animals; black dots), normalized and scaled.

different temperatures have different amounts of lifespan biomarker variation and lifespan variation[15]. Finally, in the same report in which we examined heritability, we also found that whole animals that make more biomarker have significantly elevated transcripts for five chaperones including *hsp-16.2*[27]. These previous studies led us to believe that differences in the life history of the parents and differences in the perceived or real environment can lead to differences in heat-shock protein biomarker expression. In particular, the fact that we previously detected transcriptional differences between animals that had more and less of the *hsp-16.2* biomarker 24 h after heat shock, suggests that there was an initial thermosensory neuron signaling difference that manifested at the transcript level. But, because chaperones generally affect protein production and turnover, when an intestine produces more chaperones, the increased abundance of chaperons causes a generally greater abundance of other proteins in its cells. Our results suggest that this is from both increased production and decreased turnover. That is, in this view, the initial signaling difference we think comes from neurons results in a physiological state of high protein expression capacity or high *G* in the peripheral intestine tissue, and possibly other tissues.

Figure 5a, b illustrates this model. Given the known role of chaperones in protein production, maintenance and turnover, we speculate that chaperone levels may sometimes determine differences in global protein dosage[6]. In this model, differences in environmental perception result in differences in AFD sensory neuron firing[14,29,30], which results in differences in insulin signaling in the peripheral intestine tissue[14,29,30], causing an upregulation of chaperones[27,31], which changes general protein expression capacity, increasing the dosages of many proteins. Figure 5c shows a general model of how hypomorphic and hypermorphic protein activities may be increased or decreased by chaperones (via protein expression capacity) to cause differences in the manifestation of traits. Alternative model interpretations are discussed in Supplementary Discussion. We also discuss what might be happening with intestinal protein expression capacity at other points during development and aging in the Supplementary Discussion. In addition, in the Supplementary Discussion, we further discuss how chaperones could influence the dosages of other proteins in intestine cells, including discussion of evidence arguing against variation in intestine cell ploidy causing variation in protein dosage.

How might cell-to-cell variation in gene expression manifest in other cell types? Other reports have shown that intrinsic noise can be a significant component of cell-to-cell variation in gene expression. In yeast, intrinsic noise was an order of magnitude higher than in worm intestine cells[20]. In human cell culture, intrinsic noise, to the point of monoallelic gene expression,

has been reported quite frequently now[32]. We believe that in diploid tissues, intrinsic noise of gene expression may be a large and significant component of cell-to-cell variation in gene expression. Indeed, in other work, we found that intrinsic noise is significantly higher in diploid muscle cells, compared with the polyploid intestine cells examined here[33]. Furthermore, developing a greater understanding of such intrinsic noise (stochastic autosomal allele biases, even to the point of monoallelic expression) may have implications for understanding how people with and escape the consequences of dominant negative mutations, like some oncogenes[34]. Specifically, silencing a dominant negative allele on one particular autosome can prevent the manifestation of the associated negative trait(s).

How could variation in chaperone activity affect health and disease? Natural or pharmaceutical manipulation of chaperone systems in different tissues has the potential to affect both health, in terms of robust living, and disease, in terms of affecting the activity of specific disease alleles. Anti-HSP90 drugs have been used to attempt to attenuate the activity of Ras gain of function mutations in cancer treatments for almost a decade[35,36]. Unfortunately there are side effects when targeting the master regulator of the heat-shock response; for example, the HSP90 inhibitor AUY922 had the side effects of diarrhea, skin rash, hyperglycemia, and night blindness[37,38]. It may be that other components of the protein chaperone system can be targeted with less collateral damage than targeting the master regulator of the heat-shock response. Alternatively, increasing the activity of some genes or the dosage of large portions of the proteome may be desirable in other biological scenarios. For example, increasing the dosage of chaperones through increasing the abundance of individual chaperones[39] or the master regulatory transcription factor, HSF-1[40,41], has been shown to increase lifespan in *C. elegans*. Designing small molecule therapies that elicit specific chaperone responses affecting specific subsets of traits may be a worthwhile endeavor for improving human health.

How could the natural variation in chaperone expression affect penetrance and missing heritability? Natural variation in chaperone subsystems affects the penetrance and expressivity of some traits, and is therefore likely to be responsible for the missing heritability of some traits. We know distinct chaperones affect the manifestation of distinct sets of traits[10]. We know that there is an epigenetic component to heritability[42]. And we know that chaperone levels can be epigenetically heritable, at least in *C. elegans*[27,43] and locusts[44]. We now know, at the resolution of single cells, operating at the protein level, for one metazoan tissue, that differences in the dosages of many proteins correlate with and are likely influenced by natural variation in chaperones. So, it may be that the current inability of genetic variants to account for differences in the manifestation of traits is due to another factor

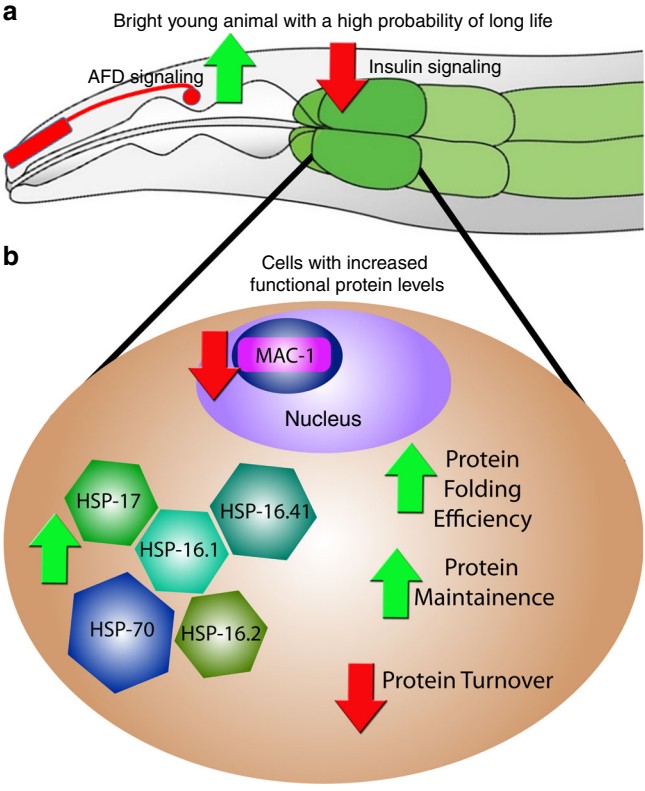

**a** Bright young animal with a high probability of long life

AFD signaling    Insulin signaling

**b** Cells with increased functional protein levels

MAC-1

Nucleus

HSP-17    HSP-16.41
HSP-16.1
HSP-70    HSP-16.2

Protein Folding Efficiency

Protein Maintainence

Protein Turnover

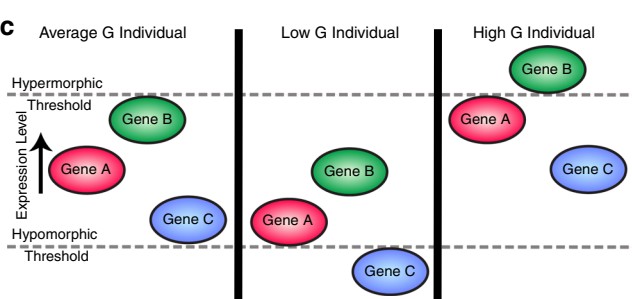

**c**

Average G Individual | Low G Individual | High G Individual

Hypermorphic Threshold

Expression Level

Hypomorphic Threshold

Gene B
Gene A
Gene C

Gene B
Gene A
Gene C

Gene A
Gene B
Gene C

**Fig. 5 Working model. a** Genetic experiments from Mendenhall et al. show that interindividual variation in the *hsp-16.2* reporter gene arises from differences in insulin signaling and depolarization of the AFD neuron pair[14]. Thus, animals with low insulin signaling and high neuronal depolarization would have the highest expression of the *hsp-16.2* lifespan/penetrance biomarker and have the longest lifespans, highest penetrance of hypermorphic phenotypes and lowest penetrance of hypomorphic phenotypes. **b** Diagram of the molecular and physiological differences in cells that underlie global differences in protein dosage. Solid arrows are directly supported by experiments in *C. elegans* using reporter genes (here in Figs. 3, 4, and Supplementary Fig. 12) and microarrays (previously[27]). **c** Three panels showing consequences of different states of protein expression capacity (*G*). Left panel shows an individual with average *G* and wild-type phenotypes. Middle panel shows an individual with low *G* and penetrance of a hypomorphic phenotype for the allele of Gene C. Right panel shows an individual with high *G* and penetrance of a hypermorphic phenotype for the allele of Gene B.

accounting for such differences, such as intrinsic, heritable physiological variation in chaperone levels.

## Methods

**Strains used or created**. We generated strains for these studies using MosSCI transgenesis[45]. We also used some knockins created via CRISPR (his-72::GFP, vit-2::GFP). We also used some standard genetic mutants: CB1370 (*daf-2*

*(e1370)*), CF1038 (*daf-16(mu86)*), NS3099 (*nhr-49(nr2041)*), PS3551 (*hsf-1 (sy441)*). We also used some transgenes from prior studies that were multicopy; RBW2 contains a biolistically integrated transgene (*emr-1::gfp*) and was described in another publication[16]. TJ2741 contains a multicopy version of $P_{hsp-16.2}$::*gfp*, *zIS2735* [$P_{hsp-16.2}$::*egfp*::$T_{unc-54}$] V, which shows the same physiological worm-to-worm variation as the several other $P_{hsp-16.2}$ reporters[16]. We used this particular version because it was genetically compatible with the *let-60(n1046)* IV mutation and enabled our sorter to detect signal in the heat-shock-pathway-attenuating *let-60(n1046)* background. We made the DNA constructs fusing promoter sequences to the coding sequence of different fluorescent proteins (XFPs) and the terminator of *unc-54* by using yeast gap repair[46]. All reporter constructs carried the 5′UTR (the upstream regulatory sequences including the promoter) of the selected genes (i.e., $P_{hsp-16.2}$) fused to an XFP coding sequence (*megfp*, *mcherry*, *mtagbfp2*, *mneptune*) and the 3′UTR of *unc-54*. We integrated fluorescent reporters into Chromosome II in the parent strain RBW6699 (an outcrossed version of EG6699 without an extrachromosomal array; *ttTi5605 II*; *unc-119(ed9)*), unless otherwise noted. To construct DNA for transgenes, we introduced reporter sequences into a vector that targeted the DNA to be inserted at the chromosome II MosSCI transposon at site *ttTi5605*. We then injected each construct at 50 ng/µL into EG6699 animals and recovered single copy insertions that we size and site validated by PCR. Thus, through transformation and/or standard genetic crosses, we created the following strains:

RBW2661 (*hutSi2661[$P_{hsp-90}$::megfp::$T_{unc-54}$ + Cbr-unc-119(+)]* II), RBW2642 (*hutSi2642[$P_{hsp-90}$::mcherry::$T_{unc-54}$ + Cbr-unc-119(+)]* II), RBW2601 (*hutSi2601 [$P_{hsp-16.2}$::megfp::$T_{unc-54}$ + Cbr-unc-119(+)]* II), RBW2561 (*hutSi2561[$P_{hsp-16.2}$:: mcherry::$T_{unc-54}$ + Cbr-unc-119(+)]* II), RBW2621 (*hutSi2621[$P_{vit-2}$::megfp:: $T_{unc-54}$ + Cbr-unc-119(+)]* II), RBW2581 (*hutSi2581[$P_{vit-2}$::mCherry::$T_{unc-54}$ + Cbr-unc-119(+)]* II), RBW2531 (*hutSi2531[$P_{mtl-2}$::mcherry::$T_{unc-54}$ + Cbr-unc-119(+)]* II), ARM1 (*let-60(n1046)*) IV; *jjIs699[$P_{lmn-1}$::emr-1::gfp::$T_{unc-54}$ + unc-119(+)]*?, not II), ARM6 (*wamSi6[$P_{eft-3}$::mtagbfp2::$T_{unc-54}$ + Cbr-unc-119(+)]* II), ARM5 (*wamSi5 [$P_{eft-3p}$::mneptune::$T_{unc-54}$ + Cbr-unc-119(+)]* II), RBW3211 (*hutSi3211[$P_{hsp-17}$:: megfp::$T_{unc-54}$ + Cbr-unc-119(+)]* II), TJ2741 ((*zIs2735[$P_{hsp-16.2}$::gfp::$T_{unc-54}$]*) V; *let-60(n1046)* IV), RBW99 (*oxSi259[$P_{eft-3}$::megfp::$T_{unc-54}$ + Cbr-unc-119(+)]* I; *oxTi179 [$P_{rps-27}$::Neo$^R$::$T_{unc-54}$ + unc-18(+)]* II); RBW2 (*zSi3002[$P_{hsp-16.2}$::mcherry::$T_{unc-54}$ + Cbr-unc-119(+)]* II; *jjIs699[$P_{lmn-1}$::emr-1::gfp::$T_{unc-54}$ + unc-119(+)]*?), ARM179 is the cross of NB147 ([*emr-1::mCherry + unc-119(+)]*)II, ARM164 ([*lmn-1:: mtagBFP2 + unc-119(+)*]) I and LP148 ([*his-72::gfp + LoxP unc-119(+) LoxP*] III), BCN9071 ([*vit-2::gfp*]), ARM183 is a stable cross of BCN9071 and RBW2642, ARM180 ($P_{eft-3}$::timer), ARM215 is a stable cross of RBW6173 ($P_{eft-3}$::GFP) and RBW2581 ($P_{vit-2}$::mCherry).

**Culture conditions**. We maintained animals on NGM plates seeded with live OP50 *E. coli* at 20°[47]. We ensured that the stocks were not starved for at least two generations prior to use for experimentation. Single-color reporter animals (see above) were mated to produce two-color F1 progeny for microscopic analysis. As a result we analyzed expression of reporters driven by identical or different promoters integrated at the identical loci on both copies of chromosomes II. The only exception is RBW2 strain that carried a transcriptional reporter ($P_{hsp-16.2}$::*mCherry*) and *emr-1::gfp* fusion transgene on different chromosomes and was maintained as a stable strain. Animals that carried $P_{hsp-16.2}$-driven reporters were heat shocked on the first day of adulthood by exposure to 35 °C for 1 h on a solid NGM plate. All reporter strains were analyzed on the second day of adulthood. We analyzed crosses listed in Supplementary Table 4.

**Mounting animals for microscopy**. We prepared a 1% agarose pad (about 1–2 cm²) on a standard glass microscope slide. We used worm anesthesia solution containing 0.2% tricaine and 0.02% tetramisole (Sigma, Inc., St. Louis) in M9. We carefully picked animals off their NGM plates and into the drop of anesthesia on the pad. We allowed the worms to swim off the pick into the drop. After that, we covered animals with a glass coverslip. We waited until worms ceased most active movements (about 15–25 min), and placed the prepared slide on the slide holder mounted on the microscope stage. Our methods for mounting were derived from ref. [16].

**Microscopic image acquisition**. We acquired images of animals expressing multiple fluorescent proteins in a similar fashion to Mendenhall et al.[16]. Briefly, we used a ×40 1.2 NA water immersion objective and acquired a z stack of images of each animal in one field of view. Thus, we were able to capture intestine cells in rings one through four of the intestine. We used minimal laser power to prevent photobleaching and heat damage to the cells. We adjusted gain and laser power to ensure that the reporter signals fit within the dynamic range of the photomultiplier tubes and did not saturate them. We only imaged animals lying on their left side to prevent signal loss attributable to imaging through the gonad. We digitally rotated the field of view to arrange the animals into diagonal orientation to fit maximal number of intestine cells in a field of view (for animals oriented in a fairly straight position; some animals adopted an omega or C pose). We then acquired sequential z-slices to span most of the intestine cells' depth. We acquired an optical slice every 2 µm from the proximal starting point at the objective-proximal side of the intestine, and continued imaging until we acquired signal from intestine cell nuclei

in the first four rings. We set optical slice thickness to 2 μm (all photons from ±2 μm of the image plane). Imaging of each animal typically took 2 min at 1024 × 1024 resolution. Each voxel value was the average of four samples. We imaged ten animals per experiment. We captured signals from eight of the cells in the first four rings. In some experiments, for some animals, we were unable to sample all the cells intended due to the orientation of the worm intestine. The cells we were not able to capture images of were almost always in rings two and four, due to depth or stretched orientation of the intestine, respectively.

**Image cytometry.** We first determined the orientation of the animals in images and then identified individual cells, following the guidelines detailed in these methods for image cytometry, which originated from ref. [16]. As the intestine is anchored anteriorly and posteriorly we used the known positions of the anteriormost cells to determine the identity of adjacent cells. We used the location of the vulva and the spermatheca to denote ventral. Once we identified a cell, we measured signal within an equatorial slice of the cell's nucleus, as a proxy for the whole cell; the nuclear signal is nearly perfectly correlated with the cytoplasmic contents[16]. We used the ImageJ software as well as custom built Nuclear Quantification Support Plugin for nucleus segmentation and signal quantification, we call C. Entmoot (Alexander Seewald, Seewald Solutions, Inc., Vienna). The algorithm uses a meeting of decision trees (an Ent Moot—a meeting of tree-beings; from J. R. R. Tolkien's *The Lord of the Rings*) used to delineate a nuclear boundary based on the drop in signal intensity, trained on user delineated images. The values of the nuclei from binucleate cells were averaged.

**Sorting animals in flow.** These experiments require the growth of large amounts of animals for sorting—more so than for simply measuring the distribution of a population. For the animals bearing the *let-60* mutation to be sorted on $P_{hsp-16.2}$:: *EGFP* expression level, we grew a hypochlorite-synchronized population of about 4000 animals to gravid adulthood on 10 cm NGM plates, @175 animals per plate, as in refs. [13,16]. For animals we would sort during the L1 diapause on $P_{eft-3}$::*GFP* expression level, we performed a hypochlorite synchronization and allowed the animals to hatch and enter the L1 diapause in a 3–5 mm deep pool of S-basal swirling at ~30 rpms on an unseeded 10 cm NGM petri dish. At the start of worm flow sorting, we first flowed a sample of the population of animals to determine the distribution of values in order to select the animals at the extremes of the distribution of fluorescent reporter gene signal; we measured between 2 and 300 animals to get an estimate of the distribution of the population. We then selected animals at the top or bottom 5–10%, or selected all animals, and sorted between 50 and 250 animals per group (bright dim, unselected). Actual numbers of animals examined were often less due to loss of animals from escape onto the side of the plate. We hid the identity of each group from the person scoring to prevent bias in scoring. For sorting based on *hsp-16.2* reporter expression in the Ras/*let-60(n1046)* mutant background, animals received a 1 h 35° heat shock in liquid S-Basal and recovered for 24 h in liquid S-Basal seeded with concentrated *E. coli* OP50 before sorting as we have done previously[13,16]; sorting otherwise proceeded as described above. Scoring proceeded immediately after sorting for $P_{hsp-16.2}$::*EGFP* expression levels for adult animals bearing the *let-60(n1046)* mutation. To score, we counted the number of distinct growths emanating from the ventral hypodermis. A distinct growth was considered a protrusion from the soma unconnected to any other apparent protrusion; we verified that these protrusions were caused by the growth of extra cells using strain ARM1, shown in Supplementary Fig. 14. The *let-60* gain of function mutation decreases the expression level of the heat-shock response, necessitating the use of the multicopy insertion to ensure detection of GFP signal in this genetic background. We showed previously that the single copy and multicopy reporters have identical lifespan prediction capabilities[13] and the same amount of worm-to-worm variation in biomarker expression[16]. For larvae expressing NeoR, we used a high magnification stereoscope and scored animals for development to gravid adulthood (at least one fertilized embryo in the uterus) after 96 h of development on food, after being sorted on $P_{eft-3}$::*GFP* expression while in the L1 diapause (72 h for experiments on regular NGM). For an additional, more extensive description of the nuances of worm flow cytometry methods, please see Mendenhall et al.[16].

**Whole animal and whole intestine image cytometry.** For whole animal/intestine analysis animals were anesthetized with 0.2%Tricaine/0.02%Tetramisole and then mounted on 1% agarose gel pads and covered with cover slips. Animals were imaged on Leica Fluorescent Dissecting scope using MicroManager program for image acquisition. For whole intestine analysis of fluorescence in $P_{vit-2}xP_{hsp-16.2}$ and $P_{vit-2}xP_{eft-3}$ animals we segmented intestines with thresholding in ImageJ using $P_{vit-2}$ fluorescent channel and measured average signals from both reporters in the segmented area. For whole body analysis of fluorescence in $P_{eft-3}$::*timer* animals we segmented entire animals with thresholding in ImageJ and measured average signals in each channel in the segmented area.

**Correlations between embryos, larva, and adults.** For correlations between embryos and adults or between larva and adults, we compared average signal intensities from the cells measured. To measure VIT-2::GFP in embryos, we isolated 2-cell embryos and focused on the equatorial plane of the two nuclei, captured an image, and then got the average voxel value for the whole embryo. We did the same thing for whole L1 larvae without anesthesia, focusing on the equatorial plane of the pharynges of animals. We compared larval or embryonic values with the average of adult animals' intestine cells by averaging cell voxel intensity values from the adult cells we measured.

**Lifespan and fecundity after UV irradiation.** We placed populations of approximately 100, age-synchronized young adult animals (60 h development on NGM with OP50 post starved L1 diapause) in a StrataLinker and irradiated them with 1000 Joules of ultraviolet radiation. We took 50 animals from each group to conduct lifespans and we took one animal from each group to perform fecundity assays, in three, or five independent experiments, respectively. We also performed self-fertility fecundity assays on individual wild-type animals that had never been irradiated in five independent experiments.

**Statistical analysis.** We used Sigma Stat (Systat Software, Inc., San Jose) or Python Seaborn (https://doi.org/10.5281/zenodo.1313201) for statistical analyses and plotting. For statistical analyses, we first determined if measurements comprising each dataset were normally distributed, then, depending on the results of those tests, used appropriate parametric or non-parametric statistics to determine if there was significant difference in any measured parameters. Details of specific tests are shown in figure legends. Additional details regarding grouping of different types of measurements for calculations of $\eta^2$ ($G$, $P$, or $\gamma$) and statistical analyses of different variation bins see Supplementary Notes 1–5, which describe the analytical framework. For all reported coefficients of correlation or determination, we used a Pearson correlation.

**Reporting summary.** Further information on research design is available in the Nature Research Reporting Summary linked to this article.

## Data availability
The datasets generated during and/or analyzed during the current study are available from the corresponding author on reasonable request.

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

## Acknowledgements

Some strains were provided by the CGC, which is funded by NIH Office of Research Infrastructure Programs (P40 OD010440). We thank the other laboratories and scientists that have contributed to the understanding of cell-to-cell variation in gene expression not referenced in this paper; a complete literature review on nongenetic, non-environmental biological variation was beyond the scope of this study. We thank George Martin for thoughtful discussions about the paper. We thank the Lehner lab for providing their *vit-2::GFP* knockin. We thank the Goldstein Lab for providing their *his-72::GFP* knockin. Funding was provided by the National Institutes of Health, National Institute on Aging R01AG039025 to T.E.J., National Institutes of Health, National Institute of General Medical Sciences R01GM97479 to R.B., the National Institutes of Health National Cancer Institute R21CA22390 to R.B., the National Institutes of Health National Institute on Aging R00AG045341 to A.M., the National Institutes of Health National Cancer Institute R01CA219460 to A.M., P50AG005136 to M.K., a training grant from the National Institute on Aging, T32AG000057, to support N.B., National Institute on Aging K99AG061216 to N.B,. and Pilot grant from the Nathan Shock Center for Excellence in the Basic Biology of Aging to A.M. (NIA Grant P30AG013280 to M.K.).

## Author contributions

A.M., N.B., T.E.J., and R.B. designed the study. N.B., A.M., and R.B. analyzed and interpreted the results. A.M., R.B., T.E.J., and M.K. provided funding. B.S., N.B., P.T., A.M., and S.Y. generated DNA and transgenic nematode strains. A.M. and N.B. conducted the imaging experiments and performed image cytometry. A.M., P.T., and B.S. conducted the flow sorting and phenotyping experiments. N.B. and A.M. analyzed the raw data. A.M., R.B., and N.B. wrote the initial paper. R.B., M.K., T.E.J., P.T., B.S., S.Y., A.M., and N.B. reviewed and revised the paper.

## Competing interests

The authors declare no competing interests.
