## [Peer Review File · Nature Communications]

Editorial Note 1: Parts of this Peer Review File have been redacted as indicated to remove third-party material where no permission to publish could be obtained.

Editorial Note 2: The attachment referenced on page 34 of this Peer Review File is not included here, as the comments therein are the same as those in the subsequent report.

Reviewers' comments:

Reviewer #1 (Remarks to the Author):

The authors compare single cell expression variation from fluorescent reporter strains to partition expression variance into that between alleles of the same reporter ('intrinsic'), variance uncorrelated between two different genes ('pathway') and correlated between different genes ('G') in *C. elegans*. They conclude for several genes expressed in the intestine that 'G' dominates as a source of variance i.e. it is variation across animals that dominates. The cause of this is unknown. They then suggest that this variance predicts the outcome of incompletely penetrant mutant phenotypes based on the analysis of two gain-of-function mutations and of lifespan based on previous data. Overall I think this is an interesting and novel study. However, I think at the moment the claims of the paper are overstated given that they are based only on the analysis of 1 gain of function mutation and one antibiotic resistance transgene. It is essential that the authors test whether their measurements of 'G' predict the outcome of multiple loss of function mutations and lifespan. I think they have discovered something potentially important, but it may not be as general as they think and/or it may not be the mechanism that they are proposing.

[1] Incomplete penetrance. The idea that a general source of variance affecting expression levels predicts incomplete penetrance is provocative. However the authors need to actually test this idea. At the moment they only do so for two gain-of-function (GOF) mutations (neomycin resistance and a *let-60/Ras* GOF mutation). However most mutations with phenotypic effects and most disease mutations are loss-of-function. The authors must test whether variation in 'G' predicts the outcome of multiple loss-of-function mutations so that we have an idea of how general the result is that they are proposing.

[2] Lifespan. If animal-to-animal variation in 'G' predicts lifespan, then variation in any reporter that quantifies 'G' should predict lifespan i.e. should account for their previous very interesting demonstration that *hsp-16.2::GFP* expression predicts lifespan after a heat shock. Is this true? Is this true without a perturbation (heat shock)?

[3] Conceptual model. The model they are proposing is not quite clear. If 'G' represents a general increase in protein production capacity, then one would expect most or all proteins to have their expression level increased. If 'G' affects the expression level of all or many different genes then the relative expression level of many genes will actually be constant and we would expect little consequence for penetrance or some changes in physiology that affect the penetrance of some genes but not others but not for the 'dosage' hypothesis suggested here. In contrast, if 'G' affects the expression of some genes more than others, why is this and which ones? Which mutations is this relevant for? It strikes me that they have found something interesting, but the mechanism they are proposing is not quite clear. Indeed it may be the 'G' reflects some general physiological variation that in turn impacts on the outcome of some mutations.

Is there any evidence of this for any of the proteins or many proteins? e.g. in Rea et al. (DOI: 10.1038/ng1608) sup fig 1 there is a large difference in GFP between high and low *hps16.2::GFP* animals, but the amount of other proteins appears very similar. i.e. it looks like a specific not a general response. Is this correct if other proteins are quantified individually? Is there actually higher expression of *let-60* in 'high' animals? What about for other proteins?

[4] Protein dosage. This is a bit misleading as it normally implies alterations in the levels of one protein relative to others e.g. due to amplification/deletion.

[5] Figure 4: Shouldn't we expect that dim worms do worse than unselected worms? I wouldn't say anything if that would be one mutant phenotype out of five - but they only looked at two.

[6] As far as I can tell the sorting experiments are all early sorts and looking at traits in adulthood. So the authors should also make some effort to demonstrate that low/high G larvae become low/high G adults (ie that this state is not very dynamic over development).

[7] Discreet or discrete? (!)
line 217 'that that', line 257 'Activation the'.

[8] The last sentence of the introduction is unusual.

Reviewer #2 (Remarks to the Author):

NCOMMS-18-15137

Title Differences in protein dosage underlie nongenetic differences in traits

The authors present a method that allows them to dissect different sources of variation in expression, based on similar studies performed in unicellular organisms. Worms carrying single-copy reporters at the same locus were analysed using microscopy to capture fluorophore expression in the first 4 pairs of intestinal cells in 10 individual worms per replicate.

They measure the fluorescence coming from the same promoter with two different fluorophores, in order to monitor the "intrinsic noise", which is explained by allele expression/access. As expected, the intrinsic noise is very low for four different pairs of identical promoters.

They also measure the fluorescence coming from two different promoters driving two fluorophores in heterozygous worms. By subtracting intrinsic noise and estimating uncorrelated variation they can now measure the "pathway/signalling noise" which is surprisingly very low. In addition, from this data they can estimate correlated sources of variation which represents common sources of variation caused by protein expression capacity.

Their conclusion is that most of the variation in gene expression comes from the "protein expression capacity" of the cells. This conclusion is based on several pairs of reporters including stress reporters (*hsp-16.1*, *daf-21mcherry*) and others (*vit-2*, *eft-3*).

This conclusion challenges the conclusion/assumptions behind their previous work where they showed that variation in *hsp-16.2* was a biomarker of lifespan, supposedly because it was linked to the levels of chaperone expression and to variability in the activation of the stress pathways. What they say here is that those worms that express more chaperones just have a higher expression capacity in general (not only for chaperones). They suggest that the variation in protein expression capacity comes from differences in protein turnover rate. They suggest a model where protein turnover is decreased when ILS is activated, and this increases *hsp-16.2* expression and also affecting the proteome.

Overall, I consider a very positive step to perform careful quantitative experiments to determine the sources of variability in multicellular organisms. The authors present a comprehensive statistical framework to analyse sources of variance, which is an asset for the field. However, although modelling points towards an alternative explanation for previous experiments, two main issues remain, the first is that by studying variability using a very low throughput technique, they may be under-sampling and biasing the results. The second is that there is no mechanistic basis to sustain their claims and therefore more experimental evidence should be provided.

Major issues:

Issues with the methodology:

1. It is important to point out that their method specifically captures the behaviour of the reporters under particular experimental conditions.

(a) Measurement relates to fluorophores (GFP, mcherry) and not protein fusions or endogenous proteins. To make a general claim, correspondence between fluorophore and endogenous proteins has to be presented.

(b) All transcriptional reporters contain an *unc-54* 3'UTR. This is by all means has to be properly controlled, because their measurements are likely biased by it.

(c) The reporters used in this study include inducible reporters (*hsp16*) and house-keeping (HK) genes. HK genes are known to be coregulated by the overall system's expression capacity, whereas inducible genes are known to form part of noise regulons that respond to upstream transcriptional reporters (PMID: 22365828). The authors are however measuring induced GFP 24 hours after transcription and therefore they are likely to be measuring post-transcriptional events, perhaps explaining why *hsp16: GFP* aligns well with HK genes. They should measure sources of variance for *hsp16* and other inducible systems shortly after induction. They should also present evidence of what is being measured, for example, pre-spliced/spliced ratio of transcripts by PCR or smFISH to clarify if their measurements represent solely posttranscriptional events.

(d) The heat shock experiments are performed at day 1 but measured at day 2 of adulthood, after the proteostasis collapse has taken place. We do not know if at this stage molecular events are very different and should therefore be re-done at earlier L4-d1 or later d2-d3.

2. The technique is missing quality controls. Can it capture transcriptional noise regulons? and if it does not, it does not invalidate their studies, it simply explains exactly what they are measuring. For example, *Vit-2* and *hsp-16.2* are known to be anti-correlated at the transcriptional level but as shown in Figure 3, here are highly correlated, so the window of time at which these experiments are performed is possibly too late to capture the variability related to (transcriptional) signalling, explaining why the measured N(P) is so (surprisingly) low. So, the authors need to show high N(P) for genes that are related by means of a common upstream regulator (same pathway) and low N(P) by genes that are not related. For both subsets of genes, provide clear evidence that genes are or not controlled by upstream genes (show causation, epistasis or the lack of it). It will be good practice also to check what happens when GFP is measured as soon as it comes up after transcription, it may be a useful method to show transcriptional and post-transcriptional events in this way.

3. Their method is low throughput. They make inter-individual variability conclusions based on measuring 4 pairs of intestinal cells in 10 individuals per replicate, which seems quite low (I assume that overall, they have 3 biological replicates, that is why they have more than 10 dots in the scatter plots). They measure fluorescence coming from the nuclei only, because there is auto-fluorescence in the cytoplasm, but they have shown in a previous paper that this recapitulates fluorescence from both nucleus and cytoplasm. One worrisome aspect of the low throughput of the technique is that by quantifying only 10 animals it may not be possible to capture the coefficient of variation of highly variable genes, by selecting only 10 animals then it may be possible under-sampling. I suggest using less careful quantification of nuclei, quantifying at least 200 animals and using whole animal fluorescence using their statistical framework. In fact, although the microscopy method is very accurate it may not be essential to use this level of accuracy because as shown in figure 2f, changes in fluorescence affect worm expression globally.

Issues with the interpretation. Authors require additional mechanistic insights to make their claims valid.

1. The authors present two experiments to validate the biological relevance of their interpretation. However, I do not think that these experiments provide sufficient evidence. In Figure 4A it is not surprising, *eft-3* is a HK gene and the variability probably reflects global expression state of an

animal, *eft-3* may be a true sensor for global expression capacity and as such it should be predictive of dosage and their consequences. Figure 4B presents evidence to suggest that the levels of *hsp16* are also a sensor for protein expression capacity. However, their experiment is flawed because they used a gain of function allele that has been shown to alter protein expression capacity (PMID: 12024031). Therefore, the function of the allele may be biasing the expression capacity of the reporter in unknown ways. They should present evidence of the effect of HSF1 OE in the penetrance of *let-60(gof)*. In addition, performing predictive studies where expression of *hsp16* is turned on at restrictive temperature, prior to switching animals at permissive temperatures for phenotypic outcomes. Use of other examples, see below.

2. If it is true that sorted animals based on any reporter, represent true protein expression capacity, they should show direct evidence for this. The experimental evidence that is absolutely required to make their claims valid is to measure protein turnover rates using cycloheximide pulse chase experiments. This is an inhibitor of protein biosynthesis due to its prevention in translational elongation. This drug has been effectively used in worms for this purpose in worms. (for example: <https://www.ncbi.nlm.nih.gov/pmc/articles/PMC3882363/>). The expectation is that sorted animals expressing more fluorophore should have lower protein turnover rates. It will be important to use their multiple reporters in their assay to make sure that no assumptions are made. It would be a valuable addition to the field to quantify how protein turnover is modified in *daf-2* and *hsf1oe* animals. Also use *let-60 (gf)* as a positive control, because it is known to alter protein turnover rates.

3. The proposed alternative explanation for the rescue of temperature sensitive (TS) hypomorphic alleles by *hsp16.2* based on protein dosage can be directly tested. Previously it has been shown that only alleles that respond to temperature and to *hsf-1* overactivation can be predicted by *hsp16.2:GFP* dosage. If their interpretation is correct, then, over-expression of hypomorphic TS alleles and not of non-TS alleles should decrease their penetrance. In addition, it implies that overexpression of HSF-1 and DAF16 (in *daf2* animals) should alter protein expression capacity. This can be directly tested by performing Cycloheximide pulse chase experiments in animals over-expressing HSF-1.

4. Present mechanistic evidence that *mml-1* causes variability based on G by performing the appropriate experiments.

Minor points

1. Explain clearly all the processes that could be captured by the N(G) component, as it is explained in supplementary in the main text.

2. Type II experiments, Figure 3 a to d are not specifically called in the main text. They show the extracted variability measurements from all obtained pairwise variations and should be said explicitly to distinguish from e. It is then less clear what is the purpose of figure 3e. I presume it is to show that for the specific pairwise combinations shown, chromosome location or fusion did not have an impact on the overall measurements, and if so, that should be added as a legend to the figure. It should also be specifically mentioned which reporters were measured after induction.

3. The method measures fluorescence in intestinal cells only and conclusions for the whole animal are based on a single tissue. I do not see any possible way to use this method in a high throughput manner, but the conclusions have to be properly framed to avoid undue generalisations.

4. Supplementary Figure 3 clearly explains why the technique does not work by averaging cell.

However, there are clear differences in the correlation values by cell. With the exception of Int3 and Int 4, other cells show higher levels of uncorrelation and therefore possibly, higher N(P). This may be a very interesting finding, because it may show sub-specialisation of intestinal cells. Can the authors show variance values in Int-1 compared to Int 3 as a separate figure?

5. Show the Coefficient of Variation for the reporters used. Figure 3 does not show values in the Y axis and these are probably arbitrary units. But is surprising that daf-21 and vit2, which are known to have very high CV values scale at the same levels than eft-3, which should not.

6. Table S1 and S2 font is too small to be read.

Reviewer #3 (Remarks to the Author):

Differences in protein dosage underlie non-genetic differences in traits

General overview:

In the manuscript, Burnaevskiy et al consider the molecular basis of non-genetic affects on variation in gene expression. The premise of their inquiry is why the expression of a gene not known to be involved in a particular phenotype is nevertheless correlated with that phenotype. This question leads them to a quantitative system utilizing reporter genes in the *C. elegans* intestine to differentiate between three classes of possible variation: intrinsic, pathway, or expression capacity. The authors find that the latter category explains most of the variation in cell-to-cell, and animal-to-animal expression. They extend their findings by correlating high or low expression levels of reporter genes with seemingly unrelated phenotypes like Neomycin resistance and differential penetrance.

I find the question to be of broad interest, and the experimental approach carefully done. Their findings are rather interesting, although not completely satisfying – for 3 reasons:

1. Nuclei in *C. elegans* intestines undergo endoreduplication (Hedgecock, 1985, *Dev. Cell*) experiencing reiterative rounds of DNA replication, which could have important effects on their results. Animals with high reporter expression could simply have higher copy number. The authors don't mention this at all and I am not sure if they have measured endoreduplication of the cells investigated.

2. Their third category ('protein expression capacity') that accounts for the majority of variation is vague and without a molecular mechanism.

3. The authors ignore potential sources of environmental variation, such as the distribution of egg-yolk between young and old mothers (Pereze, *Nature*, 2017). It does not necessarily affect their results, but it would affect their interpretations on the cause of differences in protein expression.

Nevertheless, with a few key experiments I think they can sufficiently satisfy these issues for publication.

*On a less important note, the writing style is imprecise, with several odd grammatical choices, and far too artistic for my taste in a scientific publication. I recommend a native speaker take a second, or third look before re-submission.

Suggestions regarding the three aforementioned comments:

1. The authors should try to verify the DNA content in intestines, perhaps by DAPI normalization or FISH, or at the very least explain why they think it is not an issue.
2. The most interesting interpretation of the manuscript is identifying that a major source of phenotypic variation between animals is a difference in overall protein content. This is also consistent with oxidative stress being correlated with longevity. However, this should be confirmed by a direct experimental method, rather than simply a few reporter genes. For example, the authors could lyse individual worms and obtain a total protein amount perhaps by Bradford or 280 nanodrop, or see Bensaddek et al. Proteomics. 2016 for a single-worm lysis and protein quantification method. Additionally they could measure single worm RNA content, or expression of several genes per worm with RT-qPCR, or globally with spiked-in standards in RNA-seq.
3. The authors should look if their high vs low expression animals correlate with the vitellogenin and thus egg yolk variability observed and published by the Lehner group in Nature last year.

minor suggestions:

1. More details on the methods for image analysis normalization; what was the gain, exposure, how did you normalize fluorescence intensity for the different fluors?
2. The last sentence about the emerin -gfp fusion is not well explained, only by reading the Suppl. figure legend can one interpret the meaning.

Legend: Our text is in Courier New font. Text of the reviewers appears in Geneva font.

General Overview of the Revision

In the process of producing this revised manuscript, we spent almost a year carrying out a number of additional experiments. In the rewritten manuscript, we have toned down our claims and restricted them to the data at hand and previously published work.

In the rewritten manuscript, we now more clearly explain the hypotheses about the physiological mechanisms of variation in gene expression we set out to test. Specifically, we wanted to understand the mechanisms of cell-to-cell variation in gene expression for the *hsp-16.2* biomarker. The mechanisms of cell-to-cell variation in protein expression were not known for any genes in any metazoan tissue before we began these studies; so the results are significant in terms of filling a gap in our knowledge. We now also explain the history of incomplete penetrance, how genes can affect the penetrance of traits controlled by other genes, how chaperones affect the manifestation of traits, and how chaperone reporters have been used to study non-genetic variation in discreet and complex traits. Our intent is to make it clear that natural variation in chaperones has been shown to underlie variation in complex and discrete traits by multiple means in multiple biological systems.

We follow this introduction with clearly stated hypotheses about how variation in biomarker gene expression may manifest at the protein level in cells, based on previous work we now more clearly explain.

We now believe that chaperone-related protein expression capacity can have its greatest effect on penetrance and expressivity in polyploid tissues, because the phenotypes affected by the point mutations here and in the Lehner group's reports all affect polyploid tissues - the hypodermis and intestine. In the same vein, we now also discuss other kinds of variation in gene expression, including intrinsic noise as it relates to monoallelic expression, which seems to be emerging as an extremely important source of phenotypic variation in diploid tissues.

Our studies had revealed a correlation between bright animals/ higher G/ higher penetrance and expressivity for gain of function mutations. The reviewers wanted us to show this effect for larger numbers of animals. To confirm this effect, we tried a large number of ways, including but not limited to microfluidic instruments we built ourselves for this purpose to generate larger populations of animals sorted by the differences in reporter gene expression we use to define G. These experiments definitively revealed technical reasons why we and others will not be using flow or automatic fluidic methods to separate such subpopulations. The failure of these attempts forced us to devise higher throughput non-confocal microscopic methods to identify animals with bright and dim signals from multiple intestinal reporters in animals immobilized under cover slips. We were able to publish the homemade fluidic sorter we had devised for this work as part of another study (Crane et al., 2019).

Our study had revealed a correlation between G, measured by gene expression phenotypes in young adults, and penetrance and expressivity of GoF traits whose phenotypic expression depended on earlier developmental events. In response to the reviewers, we measured G for embryonic and L1 larval gene expression phenotypes directly, and attempted to measure this for L2 L3 and L4 (in those stages, the animals are too motile, have their measurements confounded by anesthesia, and they have a distinct kind of variation in gene expression - cell autonomous bursting). To our surprise, G in embryonic and L1 animals did not correlate with G in young adults. This fact forced us to cast our results with Ras GoF mutations more speculatively.

Response to the reviewers led us to measure effects on expression of a knock-in allele and additional fusion proteins, which showed the same high correlation, and which are now part of the results.

In response to the reviewers, we used a fluorescent timer protein (derived from dsRed) to measure the absolute and relative amounts of old and new proteins in high and low G animals. Those are the same; that is, by this measure, the higher expression in high-G is not simply an effect of decreased proteolysis or increased production - brighter animals have more of both fractions, and in the same ratio.

Summary of reviewer concerns and actions we took to address them.

Reviewer 1 expressed a desire for us to explore and explain the phenomena more completely. This reviewer wished us to examine more alleles to show how general the effects on penetrance were, to examine signal from more reporters to show how general was the relationship between reporter expression and lifespan, and to articulate clear mechanistic model(s) that might explain the effects. Reviewer 2 wanted us to examine larger numbers of animals and to establish, experimentally the mechanistic basis of the effect. Reviewer 3 was concerned that the increased ploidy in the intestinal cells from which we quantified signal might have affected our measurements, that we did not go from our results to a mechanistic model, and we had not considered other kinds of non-genetic variation, such as the fact that eggs from young mothers receive less vitellogenin in their yolk than eggs from older mothers. Reviewer 3 also raised numerous technical concerns.

To summarize, in our revised manuscript, our major response to the major concerns of Reviewer 1 has been to by articulate key results from published work by others that support our conclusions, and by articulating the mechanistic model more clearly. Our response to the concerns raised by Reviewer 2 was to increase the number of animals, a yearlong effort by different means, including construction our own microfluidic sorting instruments, which eventually led us to a successful higher throughput microscopic approach. We also carried out additional experiments to learn establish mechanism, but these were not conclusive. We consider learning the mechanism a priority and one of the foci of our next 5 years of work, but beyond the scope of the present report. Our response to the concerns raised by Reviewer 3 about ploidy, was to attempt to measure it, but the measurements were too noisy to draw any conclusions from due to interindividual variation in loss of fluorescent protein during fixation. Reviewer 3 also wanted us to establish the mechanism of the effect. As with Reviewer 2, although we carried out additional experiments, those were not sufficient to establish a conclusive mechanism, and we must view a five year campaign to establish mechanism as lying outside the scope of this report. Our response to Reviewer 3's concern that we describe one known contributor to non-genetic variation, the fact that older eggs have more vitellogenin, was to perform experiments to directly address this and discuss it. Our point-by-point response to this Reviewer's technical concerns is detailed below.

Reviewers' comments:

Reviewer #1 (Remarks to the Author):

The authors compare single cell expression variation from fluorescent reporter strains to partition expression variance into that between alleles of the same reporter ('intrinsic'), variance uncorrelated between two different genes ('pathway') and correlated between different genes ('G') in *C. elegans*. They conclude for several genes expressed in the intestine that 'G' dominates as a source of variance i.e. it is variation across animals that dominates. The cause of this is unknown. They then suggest that this variance predicts the outcome of incompletely penetrant mutant phenotypes based on the analysis of two gain-of-function mutations and of lifespan based on previous data. Overall I think this is an interesting and novel study. However, I think at the moment the claims of the paper are overstated given that they are based only on the analysis of 1 gain of function mutation and one antibiotic resistance transgene. It is essential that the authors test whether their measurements of 'G' predict the outcome of multiple loss of function mutations and lifespan. I think they have discovered something potentially important, but it may not be as general as they think and/or it may not be the mechanism that they are proposing.

[1] Incomplete penetrance. The idea that a general source of variance affecting expression levels predicts incomplete penetrance is provocative. However the authors need to actually test this idea. At the moment they only do so for two gain-of-function (GOF) mutations (neomycin resistance and a let-60/Ras GOF mutation). However most mutations with phenotypic effects and most disease mutations are loss-of-function. The authors must test whether variation in 'G' predicts the outcome of multiple loss-of-function mutations so that we have an idea of how general the result is that they are proposing.

Results based on two GOF alleles, too few to be said to be general. We agree with the reviewer's point that two different alleles would not be sufficient to establish a general point. The fact that they are depends on previous findings from other investigators. In 2012, Casanueva et al. showed that expression of HSF-1, which activates expression of Hsp90 and other chaperones, decreased penetrance of partial loss of function mutations, but not of null mutations, that heat shock, which increases expression of many Hsp chaperones, and of an hsp-16.2 reporter, had the same effects, and that preexisting organism-to-organism variation in expression of a daf-21/ Hsp90 reporter had the same effect on penetrance of the single partial loss of function allele tests.

Our results are consistent with previous work but extend it to GOF alleles. They also provide a mechanistic explanation (chaperone correlated or dependent) changes in dosage of other proteins for the effects on penetrance.

In response to this critique, we have changed the introduction to explicitly mention the work by Casanueva et al., changed the results to mention that if this idea were true we might observe the same effects for GOF mutations, and introduce possible explanations for differences in expressivity and penetrance.

[2] Lifespan. If animal-to-animal variation in 'G' predicts lifespan, then variation in any reporter that quantifies 'G' should predict lifespan i.e. should account for their previous very interesting demonstration that hsp-16.2::GFP expression predicts lifespan after a heat shock. Is this true? Is this true without a perturbation (heat shock)?

Does the level of any reporter whose expression varies with G predict lifespan? Yes, it does - mostly. This is work by a number of researchers, but conspicuously Sanchez-Blanco et al. in Peter Kim's group). Remarkably, the same Kim group work showed instances in which correlated variation in two seemingly unrelated reporters was about the same predictor, again consistent with the idea that it is a global differences in the production or maintenance of proteins that predicts lifespan. In our work here, we offer a potential explanation for their results. We do note that they did have one reporter protein that did seem to

fluctuate on a distinct axis, and anticipate that there will be more when these types of analyses have been carried out for the whole proteome.

In response to the reviewer comments, we changed the introduction to make more explicit the Sanchez-Bianco finding, and changed the discussion of our results to repeat their significance.

[3] Conceptual model. The model they are proposing is not quite clear. If 'G' represents a general increase in protein production capacity, then one would expect most or all proteins to have their expression level increased. If G' affects the expression level of all or many different genes then the relative expression level of many genes will actually be constant and we would expect little consequence for penetrance. In contrast, if 'G' affects the expression of some genes more than others, why is this and which ones? Which mutations is this relevant for?

It strikes me that they have found something interesting, but the mechanism they are proposing is not quite clear. Indeed it may be the 'G' reflects some general physiological variation that in turn impacts on the outcome of some mutations. Is there any evidence of this for any of the proteins or many proteins? e.g. in Rea et al. (DOI:10.1038/ng1608) sup fig 1 there is a large difference in GFP between high and low hps16.2::GFP animals, but the amount of other proteins appears very similar. i.e. it looks like a specific not a general response. Is this correct if other proteins are quantified individually? Is there actually higher expression of let-60 in 'high' animals? What about for other proteins?

If 'G' represents a general increase in protein production capacity, then one would expect most or all proteins to have their expression level increased.

Yes.

If G' affects the expression level of all or many different genes then the relative expression level of many genes will actually be constant and we would expect little consequence for penetrance.

We are not sure if the reviewer is making a distinction between "protein production capacity" in the first sentence and "gene expression" in the second. Implicit in our description of the system is the older language that many genes are expressed into proteins. In that usage, "gene expression" and "protein production" are synonymous.

Happily, however, our uncertainty about the reviewer's meaning is not relevant here. In many cases, the severity of a phenotype caused by a particular protein might increase if the absolute level of that protein was increased. In other cases, for example if the function of an activating protein kinase is antagonized by particular protein phosphatase, only a change in the relative levels of the two proteins that would affect the severity of the phenotype. This principle has been noted by others (see for example Hart and Alon, 2013). Our own work (Andrews et al. 2016) and the work of our collaborating lab (Bush et al. 2016) have shown that this buffering (we call this "push pull") is a property of some steps in signaling systems and the yeast pheromone response system is the means by which the receptor can transmit fractional receptor occupancy to the G protein despite changes in protein number. So we agree wholeheartedly with the reviewer.

In response to the reviewer, in the discussion we have explicitly detailed that we think that particular chaperone increase the activity, and thus the effective dosage, of many proteins, but it depends on the particular chaperone as to which swath of client proteins are affected.

In contrast, if 'G' affects the expression of some genes more than others, why is this and which ones? Which mutations is this relevant for?

Note that the reviewer has identified areas about which we are also uncertain.

In response, we have changed the manuscript. We now discuss these ideas in the same new paragraph in the discussion about G affecting traits in polyploid tissues more than diploid tissues where intrinsic noise/gamma may dominate.

It strikes me that they have found something interesting, but the mechanism they are proposing is not quite clear. Indeed it may be the 'G' reflects some general physiological variation that in turn impacts on the outcome of some mutations.

Yes.

Is there any evidence of this for any of the proteins or many proteins? e.g. in Rea et al. (DOI:10.1038/ng1608) sup fig 1 there is a large difference in GFP between high and low hps16.2::GFP animals, but the amount of other proteins appears very similar. i.e. it looks like a specific not a general response. Is this correct if other proteins are quantified individually? Is there actually higher expression of let-60 in 'high' animals? What about for other proteins?

We believe that the phenomenon affects many proteins. Here, we have shown that it affects the products of over a dozen transcriptional reporters (more if you consider different colors), a few fusion proteins (including a knockin), and two GoF alleles (Ras and a NeoR gene). We have presented evidence from previous literature (Kim and Casaneuva) that it affects another dozen or so genes, and have argued that the effect on complex traits such as lifespan suggest that it might affect many proteins. Unfortunately, from the standpoint of bearing on our current ignorance, the work on Rea et al we normalized GFP to total protein so doesn't help us here.

[4] Protein dosage. This is a bit misleading as it normally implies alterations in the levels of one protein relative to others e.g. due to amplification/deletion.

We agree that the term protein dosage might be taken to mean changes in relative protein levels.

And yet, we still face the need to come up with a simple English language term to describe the phenomenon we see. We note that others (Lithgow) have faced the same issues.

In response, we have changed the text throughout. We have defined the term "protein expression capacity" and use it consistently throughout. But we used proteome dosage in the title to convey that some large fraction of the proteome covaries because "partial proteome" sounds more confusing. And we don't know the extent... just that it seems to be pretty broad so far.

[5] Figure 4: Shouldn't we expect that dim worms do worse than unselected worms? I wouldn't say anything if that would be one mutant phenotype out of five - but they only looked at two.

Yes, we were surprised by this result for the NeoR animals. Note that the unselected animals are different and behaving as expected in the Ras background. We think this unselected difference in the NeoR has to do with the pleiotropic action of neomycin phosphotransferase (NeoR) gene. While high expression of NeoR confers resistance to neomycin it may also perturb organismic functionality by off-target effects on gene expression - e.g., increases in P-enolpyruvate carboxykinase and tyrosine aminotransferase mRNA. Hence, while NeoR confers the least protection against the drug to dim animals, it also perturbs their function the least compared to an average unselected animal.

[6] As far as I can tell the sorting experiments are all early sorts and looking at traits in adulthood. So the authors should also make some effort to demonstrate that low/high G larvae become low/high G adults (ie that

this state is not very dynamic over development).

We took this objection to heart. We carried out a great deal of additional work to address this. With the standard worm sorter and with a homemade device. WE were able to make some determinations, but need to develop additional technical means to go further, as we explain below. We also clearly define that our original objective was to learn about adult animal gene expression variation because that is when lifespan is predicted and how our history with this biomarker is centered on lifespan prediction.

We have tested to see if there was a relationship between adult G and G in other developmental phases. We tested embryos, starved L1s, and we attempted to measure L2-4 animals. We found the embryonic and L1 states to not persist to adulthood. Thus, G state may change between developmental states. For instance, L1 larval G state may be a result of developmental events (such as maternal loading of ribosomes) which does not determine subsequent proteome dosage in adults. We referenced the recent Fire lab paper showing that animals can develop to L1 without any ribosome genes- demonstrating maternal ribosome load is driving early development. Thus, we suggest that high L1 protein dosage allows bright animals to go through initial development stages better in presence of high neomycin, but does not persist to adulthood. Please see the new section on persistence in the revised manuscript. The section also describes limitations on measuring G in intermediate developmental stages, such as L4.

Our Ras phenotypes data support the idea that larval G may persist or be consequential for adult G, however precise longitudinal microscopic measurement is not currently technically feasible in L2-L4. Hence, we toned down some of our conclusions regarding phenotypes and clearly describe current technical limitations in the revised manuscript.

[7] Discreet or discrete? (!)

Big thanks to the reviewer, we changed this embarrassing typo.

line 217 'that that', line 257 'Activation the'.

We thought we had caught that. Thanks as above. Changed.

[8] The last sentence of the introduction is unusual.

We believe that sentence was ill-advised. In response we eliminated it.

=====

Reviewer #2 (Remarks to the Author):

NCOMMS-18-15137

Title Differences in protein dosage underlie nongenetic differences in traits

The authors present a method that allows them to dissect different sources of variation in expression, based on similar studies performed in unicellular organisms. Worms carrying single-copy reporters at the same locus were analysed using microscopy to capture fluorophore expression in the first 4 pairs of intestinal cells in 10 individual worms per replicate.

They measure the fluorescence coming from the same promoter with two different fluorophores, in order to monitor the “intrinsic noise”, which is explained by allele expression/access. As expected, the intrinsic noise is very low for four different pairs of identical promoters.

They also measure the fluorescence coming from two different promoters driving two fluorophores in heterozygous worms. By subtracting intrinsic noise and estimating uncorrelated variation they can now measure the “pathway/signalling noise” which is surprisingly very low. In addition, from this data they can estimate correlated sources of variation which represents common sources of variation caused by protein expression capacity.

Their conclusion is that most of the variation in gene expression comes from the “protein expression capacity” of the cells. This conclusion is based on several pairs of reporters including stress reporters (hsp-16.1, daf-21mcherry) and others (vit-2, eft-3).

This conclusion challenges the conclusion/assumptions behind their previous work where they showed that variation in hsp-16.2 was a biomarker of lifespan, supposedly because it was linked to the levels of chaperone expression and to variability in the activation of the stress pathways. What they say here is that those worms that express more chaperones just have a higher expression capacity in general (not only for chaperones). They suggest that the variation in protein expression capacity comes from differences in protein turnover rate. They suggest a model where protein turnover is decreased when ILS is activated, and this increases hsp-16.2 expression and also affecting the proteome.

What they say here is that those worms that express more chaperones just have a higher expression capacity in general (not only for chaperones). They suggest that the variation in protein expression capacity comes from differences in protein turnover rate. They suggest a model where protein turnover is decreased when ILS is activated, and this increases hsp-16.2 expression and also affecting the proteome.

Overall, I consider a very positive step to perform careful quantitative experiments to determine the sources of variability in multicellular organisms. The authors present a comprehensive statistical framework to analyse (British spelling, possible wish to mislead, possible former Ben Lehner person) sources of variance, which is an asset for the field. However, although modelling points towards an alternative explanation for previous experiments, two main issues remain, the first is that by studying variability using a very low throughput technique, they may be under-sampling and biasing the results. The second is that there is no mechanistic basis to sustain their claims and therefore more experimental evidence should be provided.

Biggest points.

“...two main issues remain, the first is that by studying variability using a very low throughput technique, they may be under-sampling and biasing the results. The second is that there is no mechanistic basis to sustain their claims and therefore more experimental evidence should be provided.”

We discussed how we addressed the two major concerns, desire for more data and lack of a mechanistic explanation, in our summary response above. In brief, we

tried many different approaches to generating more data, and eventually succeeded with one, and we carried out numerous experiments to try to establish mechanism, and in particular experiments to distinguish between increased synthesis and decreased proteolysis. The revised manuscript shows that these experiments narrowed the set of possible explanations only a little.

Reviewer 2 also listed a large number of specific points of concern,

Specific points.

Major issues:

Issues with the methodology:

"1. It is important to point out that their method specifically captures the behaviour of the reporters under particular experimental conditions."

We take this observation to mean that our method only captures the time averaged steady state expression from these reporters, and does not capture, for example, short term changes such as resulting from transcriptional bursting. This method achieves the main goal we set for this work: to understand better why chaperones predict various gene expression phenotypes. In our previous work, we used expression level of *hsp-16.2* biomarker that we measured 24 hours after heat shock. Here, we follow the same methodology and discover that time-averaged ratios of different proteins and reporters is fixed for a given cell fate and that animal-to-animal differences are largely attributable to overall proteome dosage.

In response, we changed the text to make our assumptions about steady state clear in Supplementary Section 2: Analytical Framework.

(a) Measurement relates to fluorophores (GFP, mcherry) and not protein fusions or endogenous proteins. To make a general claim, correspondence between fluorophore and endogenous proteins has to be presented.

In response to the reviewer's concern, we performed additional experiments with knockins and fusion proteins. We also note that the effects of native gene products in this work (gain of functions) and previous work (loss of functions) covaried with transcriptional reporters of chaperones, which varied with the particular chaperone each reporter was designed to report on. The general claim in the revised manuscript now draws on four lines of work.

- 1) Our previous work in yeast (Colman-Lerner, 2005) showing correlation between reporter expression phenotypes.
- 2) New work presented here, in which we assay a new strain expressing three different fusion proteins: *lmn-1::BFP*, *His2B::GFP* and *Emr-1::mCherry*. Results of the measurement are shown in 3-dimensional plot in Figure 4b.
- 3) Interpretations of previously published work from other groups, Lehner and Kim, consistent with variation in phenotypic expressivity of and penetrance of hypomorphic alleles being due to changes in protein abundance.
- 4) New work presented here about variation in phenotypic expressivity and penetrance of GoF mutations.

(b) All transcriptional reporters contain an unc-54 3'UTR. This is by all means has to be properly controlled, because their measurements are likely biased by it.

We previously showed that differences in locus, copy number, fluorescent proteins and 3'UTRs (e.g., *let-858*) do not affect the amount of worm to worm variation in gene expression. Therefore it is unlikely that these sequence are abnormally

contributing to variation in our measurements. Additionally, many of the reporters or knockins in the revision do not have the unc-54 terminator, but are still highly correlated.

As we described above, our evidence that animals have correlated variation in expressed protein levels and penetrance and expressivity of phenotypes does not simply rely on unc-54 3'UTR reporters, but also on fusion proteins and other means to experimental observations.

We should also point out that we have spent 5+ years learning how to do precise apples-to-apples comparisons of reporter genes that use the unc-54 3'UTR and other 3'UTRs, such as *let-858* or *hsp-16.2*. The results of these studies are reflected in the published papers referenced below:

Sci Rep. 2019 Jun 24;9(1):9192. doi: 10.1038/s41598-019-45517-0.

In vivo measurements reveal a single 5'-intron is sufficient to increase protein expression level in *Caenorhabditis elegans*.

Crane MM1, Sands B1, Battaglia C1, Johnson B1, Yun S1, Kaeberlein M1, Brent R2, Mendenhall A3.

Environmental Canalization of Life Span and Gene Expression in *Caenorhabditis elegans*.

Mendenhall A, Crane MM, Leiser S, Sutphin G, Tedesco PM, Kaeberlein M, Johnson TE, Brent R.

J Gerontol A Biol Sci Med Sci. 2017 Aug 1;72(8):1033-1037. doi: 10.1093/gerona/glx017.

Caenorhabditis elegans Genes Affecting Interindividual Variation in Life-span Biomarker Gene Expression.

Mendenhall A, Crane MM, Tedesco PM, Johnson TE, Brent R.

J Gerontol A Biol Sci Med Sci. 2017 Oct 1;72(10):1305-1310. doi: 10.1093/gerona/glw349.

Single Cell Quantification of Reporter Gene Expression in Live Adult *Caenorhabditis elegans* Reveals Reproducible Cell-Specific Expression Patterns and Underlying Biological Variation.

Mendenhall AR, Tedesco PM, Sands B, Johnson TE, Brent R.

PLoS One. 2015 May 6;10(5):e0124289. doi: 10.1371/journal.pone.0124289.

In response, the rewritten manuscript now emphasizes the multiple lines of evidence for differences in protein abundance as above, and our extensive expertise in precise quantification and comparison of gene expression.

*(c) The reporters used in this study include inducible reporters (*hsp16*) and house-keeping (HK) genes. HK genes are known to be coregulated by the overall system's expression capacity, whereas inducible genes are known to form part of noise regulons that respond to upstream transcriptional reporters (PMID: 22365828).*

Reviewer references an interesting paper by Stewart-Orenstein, Weismann and El-Samad. This paper, from 2012, uses GFP fusions to identify "noise regulons", sets of genes whose expression co-varies with variation in expression of an upstream regulatory gene, and establishes this covariation as a criterion for identifying genes subject to the same regulatory control. Conclusions from this paper are overall consistent with those from our work from 2005, but go beyond that work in their examination of genome wide changes in expression.

Importantly, this paper is in yeast and not *C. elegans*. So, there is no reason a priori why we should expect gene expression variation in single celled yeast to work the same in the metazoan *C. elegans*. In fact, we would expect gene expression variation to be restricted in a multicellular animal that needs to coordinate gene expression among the cells comprising its tissues. And that is

exactly what we saw. Additionally, the invariant cellular lineage of *C. elegans* suggests that variation in gene expression must be constrained; otherwise, how could such phenotypic invariance be possible?

Importantly, Stewart-Orenstein et al. find that *"therefore, noise in S. cerevisiae can be separated into local stochastic components that dominate at low expression, a moderate level of global variation common to all genes due to variations in the overall transcriptional, translational, and metabolic capacity of cells, and substantial extrinsic noise of unknown origin affecting only a subset of genes."* This is essentially G rediscovered in 2012. We simply find this component to be extremely dominant in the polyploid intestine cells of adult *C. elegans*.

Genes involved in Stewart-Orenstein's noise regulons are defined as genes subject to that third source of variation, while our work here measures the effects of the second, global variation. We, (and presumably Stewart-Orenstein et al.) would stress that this effect is "global variation common to all genes", not simply housekeeping genes.

The revised manuscript more explicitly states that we think that some incompletely defined fraction of the proteome, perhaps quite large, covaries with chaperone biomarkers, which means that some fraction of the proteome is covarying with the natural variation in chaperon abundance.

"The authors are however measuring induced GFP 24 hours after transcription and therefore they are likely to be measuring post-transcriptional events, perhaps explaining why hsp16: GFP aligns well with HK genes. "

We thank the reviewer for this comment. We agree that the differences in protein abundance we detect is likely due to post-transcriptional events and transcriptional events. *In response, we explicitly state that we are measuring the dosage of proteins that are the result of production, maintenance (refolding) and turnover.* The goal of this study was to determine the mechanisms of cell-to-cell variation in gene expression for a robust biomarker that predicts differences in both the outcome of point mutations and lifespan. This has not been done before for a metazoan. We think achieved our goal, but did not determine any conclusive mechanism, besides the already apparent one that chaperones affect a diverse array of clients.

They should measure sources of variance for hsp16 and other inducible systems shortly after induction.

We believe that Reviewer 2 asks us to dissect the relative contributions of the different sources of variation at earlier times after induction than 24 hours. We agree that measurement of *hsp-16.2* expression in shorter time frame may yield different results. For instance, our own observations that prevented us from careful analysis of developing larvae argue that bursts of expression can be seen in shorter time frame. But we did not carry out these experiments in order to understand the sources of variation in *hsp-16.2* reporter expression just after heat shock. Rather, we undertook the work here to determine why it was that *hsp-16.2* reporter expression predicted diverse phenotypes, and to understand the mechanisms of cell-to-cell variation in gene expression at the point that the reporter is used to predict the trait we have been focused on for years, lifespan.

In response, we rewrote the introduction to more clearly explain the motivation for the experiments, and we revised the Analytical Framework and the discussion of results to include the limitations of our measurements and what we are able to determine from protein measurement in adult animals in steady state.

They should also present evidence of what is being measured, for example, pre-spliced/spliced ratio of

transcripts by PCR or smFISH to clarify if their measurements represent solely posttranscriptional events.

As we described in our general response, although we undertook some experiments to determine further molecular mechanisms underlying differences in expression capacity, we feel that such a large amount of work simply goes beyond the scope of the current study.

In the future it will be interesting to address different possibilities: difference in transcription, translation, protein turnover etc.

In response, we rewrote the whole manuscript, more explicitly detailing that we are measuring protein.

(d) The heat shock experiments are performed at day 1 but measured at day 2 of adulthood, after the proteostasis collapse has taken place. We do not know if at this stage molecular events are very different and should therefore be re-done at earlier L4-d1 or later d2-d3.

Briefly, to address these concerns, we did measure animals at other time points and their correlations with adult expression.

However, we think the reviewer may be a bit confused or perhaps taking too much stock from a single data point. It is easy to do with the vast and sometimes accidentally confusing literature on proteostasis. The major proteostatic collapse, defined by massive aggregate formation and decreased ability to induce gene expression, happens at day 4 of adulthood at 20C. At day 2 of adulthood, animals are making more yolk (synthesized in the intestine cells we are measuring) and their best eggs. So, while there may be some damage, the animals are at their most functional in terms of producing progeny. This reviewer seems to refer to the fact that you can see some damage with unstable proteins and breaks in myosin rows even in L4 animals. It seems hyperbolic and is at odds with the literature and our own observations, but we think that the reviewer is generally concerned that we are not looking at multiple time points.

We normally take the term, as used by Morimoto, to refer to age-related loss of proteostasis. In this context, we believe Reviewer 2 could also mean that the heat shock itself causes proteins to unfold and misfold, and that after heat shock some proteins might be degraded or formed into aggregates rather than being properly refolded.

While much is known (and more is speculated) about the cellular sequelae for proteins following heat shock, we restate that the point of this study was to better understand the correlation between high expression of the *hsp-16.2* reporter and other phenotypes such as lifespan. It was not to understand the molecular events affecting reporter expression, not the possible developmental differences in cellular physiology in days before and after heat shock.

In the revised paper, our response to this criticism is the same as for the previous point. We rewrote the introduction to make our motivation clear and to the discussion to stress what we did learn and did not learn.

2. The technique is missing quality controls. Can it capture transcriptional noise regulons?

If we understand the first sentence, we point again to our 5+ years of work and four key publications demonstrating our fanatical attention to learning make careful measurements of reporter gene expression in living worms. We think that the reviewer wants us to find distinct signaling pathways. Again, we note that here, we found that, among the dozens of genes we surveyed, we found global covariation. That said, we also found that when we, for example, heat shocked animals, yolk protein production decreased, and translation elongation factor expression increased. We also saw that when we applied a heat shock, heat shock

reporter expression increased. Again, we, (and presumably Stewart-Orenstein et al.) would stress that this effect is "global variation common to all genes", not simply housekeeping genes.

As for whether the method can capture the transcriptional noise regulons defined by Stewart-Orenstein, Weissman, and El-Samad, we first need to state that the noise regulons defined by these investigators were not, strictly speaking, "transcriptional". Rather, they were defined by differences in expression of fluorescent reporter proteins, just as in our work.

That said, in our framework, were we to perform the needed experiments, co-regulation of genes (signaling) would be captured by coordinated changes in the quantity P , and residence in a noise regulon by coordinated changes in the noise (η^2P). As presented in the paper, use of the framework already defines "signaling landscapes" and different cells at different developmental stages show different landscapes. For example, we found transcriptional bursting happening in larvae (Fig S17), but that kind of noise in gene expression, and that particular point in development were beyond the initial scope of the investigation. However, it is important to note that here, in the young adult animal, in which most developmentally important signaling has ceased, such differences here do not loom large, that the differences in gene expression we are studying are gene are dominated by global differences in gene expression capacity, the variable we call G .

In response to this concern, we have changed the introduction to better delineate what we are studying, and explicitly inserted into the discussion a section saying what our study has and has not addressed.

So, the authors need to show high $N(P)$ for genes that are related by means of a common upstream regulator (same pathway) and low $N(P)$ by genes that are not related. For both subsets of genes, provide clear evidence that genes are or not controlled by upstream genes (show causation, epistasis or the lack of it). It will be good practice also to check what happens when GFP is measured as soon as it comes up after transcription, it may be a useful method to show transcriptional and post-transcriptional events in this way.

We understand what Reviewer 2 is saying here, and agree that such a line of experimentation would be appropriate for a study that sought to, for example, identify genes whose expression was regulated by a common regulatory element. Certainly, for all the genes whose expression we quantified here, their expression and differential expression depends on the products of other genes. In many cases the identity of these necessary upstream genes is known to us and their relationship to one another in signaling pathways defined by epistasis is known to us (they are chaperones and part of the protein chaperone network). We do not intend this work to be an investigation or re-investigation of the specific pathways that drive expression of our reporter genes.

Again, in response to this concern, we have changed the introduction to better delineate that we are studying natural variation in the protein chaperone system, and explicitly detailed in the discussion a section saying what kinds of variation in gene expression our study has and has not addressed. We are also clearer that we are extending prior work on chaperones and hypomorphic mutations by the Lehner lab by testing the hypothesis that more chaperones equates to more gene activity.

It will be good practice also to check what happens when GFP is measured as soon as it comes up after transcription, it may be a useful method to show transcriptional and post-transcriptional events in this way.

We agree that deeper understanding of the events, transcriptional and post-transcriptional, translational, etc. underlying these differences in gene expression would be good to have.

Again, in response to this suggestion, we have changed the introduction to better delineate what we are studying, and more clearly describe what our analytical framework and approach captures.

3. *Their method is low throughput. They make inter-individual variability conclusions based on measuring 4 pairs of intestinal cells in 10 individuals per replicate, which seems quite low (I assume that overall, they have 3 biological replicates, that is why they have more than 10 dots in the scatter plots). They measure fluorescence coming from the nuclei only, because there is auto-fluorescence in the cytoplasm, but they have shown in a previous paper that this recapitulates fluorescence from both nucleus and cytoplasm. One worrisome aspect of the low throughput of the technique is that by quantifying only 10 animals it may not be possible to capture the coefficient of variation of highly variable genes, by selecting only 10 animals then it may be possible under-sampling.*

We thank the reviewer for making this point. We measured hundreds of animals in large populations and observed that G was still the dominant axis. Given that we measured over 600 animals in several individual experiments before, this was an expected result, but it is nice to see the same result from many different experiments. It strengthens the paper. We have now revised the text to mention our previous work, and to detail the failings of both the worm sorter and our own microfluidic device, to ensure reader appreciates our new microscopic means of increasing measurement throughput at animal resolution.

I suggest using less careful quantification of nuclei, quantifying at least 200 animals and using whole animal fluorescence using their statistical framework. In fact, although the microscopy method is very accurate it may not be essential to use this level of accuracy because as shown in figure 2f, changes in fluorescence affect worm expression globally.

We addressed this suggestion in our response above.

Issues with the interpretation. Authors require additional mechanistic insights to make their claims valid.

1. *The authors present two experiments to validate the biological relevance of their interpretation. However, I do not think that these experiments provide sufficient evidence. In Figure 4A it is not surprising, *eft-3* is a HK gene and the variability probably reflects global expression [state] of an animal, *eft-3* may be a true sensor for global expression capacity and as such it should be predictive of dosage and their consequences.*

*Yes, *eft-3* does covary with everything else we measured. But so do most other genes. In the introduction of the revised manuscript, we now describe that Stuart Kim's lab was able to predict lifespan with more reporters than not.*

*Figure 4B presents evidence to suggest that the levels of *hsp16* are also a sensor for protein expression capacity. However, their experiment is flawed because they used a gain of function allele that has been shown to alter protein expression capacity (PMID: 12024031). Therefore, the function of the allele may be biasing the expression capacity of the reporter in unknown ways.*

We thank the reviewer for remembering this paper. Yes, there is a claim that GoF Ras can cause protein degradation in muscle. We can even presuppose that Ras GoF might degradation of proteins in all the cells of the organism. And yet, were that true, it would not be relevant for our current results. Consider. We are selecting animals that have a high expression of the reporter and then find that a GoF Ras has greater penetrance. If GoF Ras caused substantial protein degradation, we would expect that higher Ras activity would correlated with lower protein dosage and reduced phenotypic expressivity and penetrance, but it is the opposite that we observe.

Thus, we suggest that the magnitude of any postulated Ras mediated increase in whole animal protein degradation is not sufficient to overcome the opposite effects we observe.

They should present evidence of the effect of HSF1 OE in the penetrance of let-60(gof).

Unfortunately, we don't expect that simple overexpression of HSF-1 will surely increase penetrance of Ras. If increased expression of heat shock chaperones causes increased in global protein expression, or if these merely correlate with increases in global expression, increases in HSF1 activity might increase Ras penetrance. And yet, total penetrance should depend on total Ras activity which results from production and maintenance of Ras proteins. HSF-1 overexpression in the wild type background produces small sized long lived animals with reduced fecundity. Thus, HSF-1 likely decreases metabolic rate and rate of protein synthesis, despite its effect to increase protein stability. We therefore believe that the suggested experiment would be indirect, and its results difficult to interpret.

Due to these complications, we did not carry out this experiment, nor change the manuscript in response to this suggested experiment.

In addition, [experimenters should perform] predictive studies where expression of hsp16 is turned on at restrictive temperature, prior to switching animals at permissive temperatures for phenotypic outcomes. Use of other examples, see below.

If we understand the first sentence, moving animals to a higher inducing temperature and then assaying phenotypic effects on animals at the lower temperature is precisely the experimental setup in Rea et al.

We do not understand the second sentence.

In response to this suggestion, we did not change the manuscript.

2. If it is true that sorted animals based on any reporter, represent true protein expression capacity, they should show direct evidence for this. The experimental evidence that is absolutely required to make their claims valid is to measure protein turnover rates using cycloheximide pulse chase experiments. This is an inhibitor of protein biosynthesis due to its prevention in translational elongation. This drug has been effectively used in worms for this purpose in worms. (for example: <https://www.ncbi.nlm.nih.gov/pmc/articles/PMC3882363/>). The expectation is that sorted animals expressing more fluorophore should have lower protein turnover rates. It will be important to use their multiple reporters in their assay to make sure that no assumptions are made.

We are familiar with the use of cyclohexamide and other inhibitors in kinetic experiments of this type (Gordon et al. 2007). We understand that if the object of this study was to show that the global differences in protein dosage we observe were caused by differences in protein turnover, that such experiments would be helpful in making our case. We certainly agree if we wished to establish the mechanism for the differences in protein dosage, direct experiments of this type could be the means.

In the revised manuscript we have articulated major goals of the study and our major observations better. We were able to determine that high abundance of chaperones correlate with high abundance of other cellular proteins. We suggest that high abundance of chaperones ensures better proteostasis and thus enhanced somatic function. We now state more clearly that we do not yet know all the underlying molecular mechanisms behind non-genetic variation in chaperone abundance. Protein turnover may part the answer, and should be a focus of future

studies, along with other possibilities (translation rate, transcription rate, mRNA export rate etc).

It would be a valuable addition to the field to quantify how protein turnover is modified in daf-2 and hsf1oe animals.

For the reasons described above, we fear that any experiments with HSF-1 OE animals would be inconclusive. Because we believe that protein turnover might be the cause of the global differences in protein expression, we agree that it would be good to understand how it differs in daf-2 animals. In fact, recent work has cast some light on this. What they find is consistent with our hypothesis.

We mention this in the discussion section.

Also use let-60 (gf) as a positive control, because it is known to alter protein turnover rates.

If we wished to affect protein dosage by changing protein turnover, we likely would not do so by changing protein turnover by following up on the single report that Ras GoF might trigger degradation of proteins in worm muscle.

The proposed alternative explanation for the rescue of temperature sensitive (TS) hypomorphic alleles by hsp16.2 based on protein dosage can be directly tested. Previously it has been shown that only [increases in penetrance and expressivity of] alleles that respond to temperature and to hsf-1 overactivation can be predicted by hsp16.2:GFP dosage.

If their interpretation is correct, then, over-expression of hypomorphic TS alleles and not of non-TS alleles should decrease their penetrance.

We are familiar with the results of Casaneuva, Burga, and Lehner 2012. We take from that work a different interpretation, the idea that treatments including HSF1 OE and heat shock elevate penetrance and expressivity of those alleles that cause mutations in protein coding sequences, and not of nulls. In this view, overexpression of hypomorphic point mutations, but not of nulls and deletions, should decrease penetrance.

We have now explicitly detailed this in the revised manuscript.

In addition, [these results imply] that overexpression of HSF-1 and DAF16 (in daf2 animals) should alter protein expression capacity. ([defined so as to include] Global differences in protein dosage, perhaps caused by differences in protein turnover) This [implication] can be directly tested by performing Cycloheximide pulse chase experiments in animals over-expressing HSF-1.

These are interesting suggestions that could be explored as an entire R01, but unfortunately the effects of these genes are pleiotropic. We agree that one could try to address effects of proteolysis on global gene expression by such experiments, and see if altering proteolysis affected global protein dosage. We again note that interpretation of such experiments would be confounded by a number of factors, particularly including the fact that HSF-1 overexpression in the wild type background produces small long lived animals with reduced fecundity. The likely reason being that HSF-1 decreases metabolic rate and rate of protein synthesis, despite its effect to increase protein stability. We therefore believe that the suggested experiment would be indirect, and its results difficult to interpret.

4. [The authors should?/ could?] Present mechanistic evidence that mml-1 causes variability based on G by performing the appropriate experiments.

We submit that we are allowed to propose an alternative hypothesis in the discussion without being required to test it.

We note that most of the experiments we have done for this manuscript have been done to address speculations in the discussion.

We now more clearly state our original intent to discern the mechanisms of cell to cell variation for a chaperone biomarker that was already known to affect several discrete (e.g., penetrance of *lin-31* and *vab-9*) and complex traits (e.g., lifespan and thermotolerance).

Minor points

1. Explain clearly all the processes that could be captured by the N(G) component, as it is explained in supplementary in the main text.

We have changed the main text to reflect this point to the extent we can given space limitations and our goal of keeping readers focused on the most relevant technical details. We also now provide additional details in three additional cartoon diagrams in Figure S1-S3.

Type II experiments, Figure 3 a to d are not specifically called in the main text. [They] show the extracted variability measurements from all obtained pairwise variations and should be said explicitly to distinguish from e. It is then less clear what is the purpose of figure 3e. I presume it is to show that for the specific pairwise combinations shown, chromosome location or fusion did not have an impact on the overall measurements, and if so, that should be added as a legend to the figure. It should also be specifically mentioned which reporters were measured after induction.

We thank the reviewer for noticing this omission.

We revised the manuscript to explicitly detail that locus and whether or not the fluorescent protein was fused to another protein. Knockin and fusion proteins are now split into figure 4.

3. The method measures fluorescence in intestinal cells only and conclusions for the whole animal are based on a single tissue.

We have changed the text to stress that, in published work, (ours and from Rea et al. 2005 on), that high expression of these genes (and, as ample work shows, in this tissue, the intestine operationally defines a physiological state. This effects of this state can be observed in complex whole animal phenotypes: lifespan, thermotolerance, and, in our experiments, resistance to neomycin, and in at least one different tissue, the hypodermis.

I do not see any possible way to use this method in a high throughput manner,

The reviewer has the right to say this. As we stated above, we were unable to use or build sorting devices that used this method, but we did eventually develop a higher throughput microscopic method that allowed us to get larger numbers.

but the conclusions have to be properly framed to avoid undue generalisations.

We believe the revised text avoids such undue generalizations.

*4. Supplementary Figure 3 clearly explains why the technique does not work by averaging cell. However, there are clear differences in the correlation values by cell. With the exception of *Int3* and *Int 4*, other cells show*

higher levels of uncorrelation and therefore possibly, higher N(P). This may be a very interesting finding, because it may show sub-specialisation of intestinal cells. Can the authors show variance values in Int-1 compared to Int-3 as a separate figure?

We are gratified that the reviewer would like to know these things. In 2015, we measured such variation for one reporter in many intestinal cells.

We have now included this information in Supplemental Figure S9 and S10.

5. Show the Coefficient of Variation for the reporters used. Figure 3 does not show values in the Y axis and these are probably arbitrary units. But is surprising that daf-21 and vit2, which are known to have very high CV values scale at the same levels than eft-3, which should not.

We are not aware of reports by researchers other than us that have published CVs for these genes or other measures of normalized variation in reporter genes. We have measured the related η^2 , G,P and Gamma for expression of these reporters and report it for specific cell types in Supplementary Figures S9 and S10. Thus, these two figures provide cell specific measures of more specific components of the CV. We think this is what the reviewer ultimately wants.

6. Table S1 and S2 font is too small to be read.

We thank the reviewer for spotting this. *The font is now readable when zoomed in. The other option would be to rotate the page. So, selecting font that was amicable to zooming in seemed like it would be more amicable than inserting a rotated page. We are open to changing the tables if reviewers and/or the editor strongly advise so.*

Reviewer #3 (Remarks to the Author):

Differences in protein dosage underlie non-genetic differences in traits

General overview:

In the manuscript, Burnaevskiy et al consider the molecular basis of non-genetic affects on variation in gene expression. The premise of their inquiry is why the expression of a gene not known to be involved in a particular phenotype is nevertheless correlated with that phenotype. This question leads them to a quantitative system utilizing reporter genes in the *C. elegans* intestine to differentiate between three classes of possible variation: intrinsic, pathway, or expression capacity. The authors find that the latter category explains most of the variation in cell-to-cell, and animal-to-animal expression. They extend their findings by correlating high or low expression levels of reporter genes with seemingly unrelated phenotypes like Neomycin resistance and differential penetrance.

I find the question to be of broad interest, and the experimental approach carefully done. Their findings are rather interesting, although not completely satisfying – for 3 reasons:

1. Nuclei in *C. elegans* intestines undergo endoreduplication (Hedgecock, 1985, *Dev. Cell*) experiencing reiterative rounds of DNA replication, which could have important effects on their results. Animals with high reporter expression could simply have higher copy number. The authors don't mention this at all and I am not sure if they have measured endoreduplication of the cells investigated.

Appreciated. Please see detailed response below.

2. Their third category ('protein expression capacity') that accounts for the majority of variation is vague and without a molecular mechanism.

Yes.

Indeed we do not yet fully understand the mechanisms behind variation in 'expression capacity'. We believe it has something to do with natural variation in chaperones. In the revised manuscript we have better articulated our findings and interpretation. We find that abundance of chaperons is correlated with abundance of other proteins (maybe not all, but presumably many) in intestine cells of *C. elegans*. Given the known role of chaperones in protein production and maintenance, we propose that chaperones are least partially determine this 'expression capacity' or 'protein dosage'. Please see the revised manuscript for detailed interpretation of our results.

3. The authors ignore potential sources of environmental variation, such as the distribution of egg-yolk between young and old mothers (Pereze, *Nature*, 2017). It does not necessarily affect their results, but it would affect their interpretations on the cause of differences in protein expression.

We have directly tested this possibility. Please see the detailed response below.

Nevertheless, with a few key experiments I think they can sufficiently satisfy these issues for publication.

*On a less important note, the writing style is imprecise, with several odd grammatical choices, and far too artistic for my taste in a scientific publication. I recommend a native speaker take a second, or third look before re-submission.

We rewrote the manuscript substantially and we now believe that the revised

manuscript is more succinct and precise.

Suggestions regarding the three aforementioned comments:

1. The authors should try to verify the DNA content in intestines, perhaps by DAPI normalization or FISH, or at the very least explain why they think it is not an issue.

We cannot yet rule this out but we do not think it is likely. We attempted to perform experiments to address it. We do not currently trust the fixation methods for quantifying the relationship between ploidy and gene expression because there is too much uncontrolled loss of fluorescent protein from the nuclei during fixation. Additionally, in related work that will be published elsewhere, we found approximately the same magnitude extrinsic component in diploid muscle cells in a study focused on cis control of intrinsic noise. Thus, if the same amount of variation exists in another diploid tissue, the cause is not likely ploidy. We now state this in the revised manuscript discussion and refer to this biorxiv work.

In addition, we have previously found that loss of AFD thermosensory neurons firing (in *tax-4* and *gcy-8* mutant animals) reduces *hsp-16.2* biomarker expression as well as animal-to-animal variation in biomarker expression. Importantly, worms mutant are morphologically normal and display no body size abnormalities. If animal-to-animal variation in gene expression was due to ploidy variation, body size abnormalities would be expected given prior findings regarding body size and tissues ploidy in *C. elegans* (e.g. *cye-1* mutants have reduced ploidy and small body size).

Furthermore, copy number of the reporter gene had nothing to do with the amount of worm-to-worm variation in gene expression (Mendenhall et al 2015). If ploidy was affecting gene expression levels, than we might expect that high copy number genes would tend to get looped out during endoreduplications; this would mean that higher copy number strains would tend to have a relatively lower range of differences between bright and dim animals. In fact, whether worms are heterozygous for a single copy *hsp-16.2* reporter, or diploid for a 1000 copy dsRed reporter, they still have the same worm to worm variation in gene expression (Mendenhall et al 2015). They could all still have the same losses of ploidy too. It just doesn't seem likely with the high copy number strains covarying with the single copy strains, in terms of the distributions of expression levels in populations.

Finally, in haploid yeast, they also found big differences in G, which cannot be underlain by differences in ploidy, because there were none (as far as we know).

With that being said, we measured ploidy using DAPI staining. Please see the

figure. We don't see significant difference between bright and dim animals. However, technical differences in staining resulted in high animal-to-animal variation that precluded us from drawing firm conclusions regarding positive or negative results. For this reason, we show this result here, but we do not include it into the manuscript in order to not publish potentially misleading

data. Additionally, we could not perform a simple correlation between ploidy and expression level because of large differences in how many genomes we quantified in known diploid nuclei, and because of interindividual differences in how much fluorescent protein leaked out of each animal's cells.

2. The most interesting interpretation of the manuscript is identifying that a major source of phenotypic variation between animals is a difference in overall protein content. This is also consistent with oxidative stress being correlated with longevity. However, this should be confirmed by a direct experimental method, rather than simply a few reporter genes. For example, the authors could lyse individual worms and obtain a total protein amount perhaps by Bradford or 280 nanodrop, or see Bensaddek et al. Proteomics. 2016 for a single-worm lysis and protein quantification method. Additionally they could measure single worm RNA content, or expression of several genes per worm with RT-qPCR, or globally with spiked-in standards in RNA-seq.

We have tested this possibility and we did not see significant difference between bright and dim animals. Please see the figure. We have used sterile *glp-1* animals to only measure protein content in somatic tissues and we have obtained values similar to those obtained by recent David Gems publication (Ezcurra et al, 2018). However, we find again that sample-to-sample variation is quite high and we cannot rule out the possibility that there may be differences. Nor can we confidently say that there are not differences because of the technical variability we observe in our

quantitative protein assay (BCA assay). For this reason, we show this result here, but we do not include it in the manuscript. In the revised manuscript we now clearly introduce our hypotheses on the mechanisms of cell to cell variation in gene expression we are trying to test, and describe that it is known that these differences are underlain by natural variation in chaperone abundance. This natural variation in chaperone abundance may or may not affect the total protein; it may just affect the fraction of properly folded protein. We simply do not know what underlies variation in chaperone abundance, besides what we have already published (neurons and insulin signaling), and we are now clear about this.

What we now know, is that animals with higher G are better able to both produce and maintain protein. To learn this, we used a reporter protein that allowed us to quantify the fraction of new and old protein with or without heat shock in bright and dim animals. It is the same. Thus, animals that make more reporter protein are better able to both produce and maintain proteins, but do not do one better than the other, relative to dimmer animals.

3. The authors should look if their high vs low expression animals correlate with the vitellogenin and thus egg yolk variability observed and published by the Lehner group in Nature last year.

We appreciate the suggestion. We obtained *vit-2* knock-in strain from Lehner lab and we directly tested the hypothesis. We performed longitudinal examination of animals at 2-cell embryonic stage and at second day of adulthood and we found no correlation between measures. Thus, in the revised manuscript we report that variation in yolk loading is not a mechanism for variation in expression capacity of adult animals. See new Supplementary Figure S16.

minor suggestions:

1. More details on the methods for image analysis normalization; what was the gain, exposure, how did you normalize fluorescence intensity for the different fluores?

Yes. We have reworded the text to put all that in.

2. The last sentence about the emerin -gfp fusion is not well explained, only by reading the Suppl. figure legend can one interpret the meaning.

With thanks to the reviewer for catching this. We corrected this in the revised manuscript.

Reviewers' comments:

Reviewer #1 (Remarks to the Author):

The referees have addressed most of my concerns, as well as those of referee 3. I think this is an interesting and important study that should be published without delay.

Suggestions to improve the presentation:

[1] The last sentence of the abstract is very speculative. There is little evidence that variation in protein dosage or (endogenous) chaperone activity is inherited in most animals, particularly in humans. So whilst it is an interesting idea that this may contribute to the heritability of traits, this isn't the question addressed in this manuscript and I think this is best left to the discussion rather than the abstract.

[2] line 42 and elsewhere - this sentence is incorrect. In yeast (and worms), variation in chaperone activity (including hsp90) can modify the outcome of null alleles. e.g. Zhao et al. Cell 2005 120:715. (one assumes by an indirect mechanism).

[3] line 95 'Here we adapted an approach we used in yeast¹⁸'. I understand that this is correct, but the authors' approach is an extension of the intrinsic - extrinsic noise distinction pioneered by other labs so I think broader citations would be fairer here.

[4] It would be useful to have a figure that more explicitly quantifies how much of the variation is accounted for by variation between the cell types (rings) and how well correlated the between individual variation is comparing different rings across the same individuals.

[5] The working models section is discussion and so should be moved from the results to the discussion section.

[6] I don't really understand why the authors seem to favour a model in which variation in chaperone activity causes variation in protein production rather than being simply part of it /a consequence of it.

[7] 'These differences can arise from epigenetic differences²⁸ and seemingly stochastic environmental perception differences^{14,15}.' It would be good to unpack this rather vague sentence.

[8] 'Furthermore, understanding this can kind of noise in gene expression may have implications for understanding how people can live with and escape the consequences of dominant negative mutations, like some oncogenes⁴⁰.' Again, I can guess at what the authors mean but it would be better to be more explicit.

[9] 'And we know that chaperone levels can be epigenetically heritable²⁸.' Most readers will not be familiar with what was shown in ref 28 and how general this result is. It would be helpful to be more precise and to state whether such conclusions have been replicated by other labs or observed in other species.

Reviewer #2 (Remarks to the Author):

The authors present a very useful statistical framework to study different sources of variability in a metazoan model organism. They also make some interesting observations regarding protein expression capacity and its relevance for health span. The authors addressed some of the issues raised by reviewers and included a few new figures to give more substance to their claims. In

particular, they have used an alternative method to measure variability across worms, which allows them to increase the number of observations, a sticking point for any study that claims to be studying variability. The authors have added additional reporters and a TIMER based reporter, to define the behaviour of the reporters over time.

Generally, these results should be enough to grant them an acceptance. However, I still have issues with the interpretation of the results and as written, I do not think it should be published. The authors claim that the findings have been more accurately presented. I would say that the main text of the article does read better than the previous version; however, there are so many inaccuracies and claims that are not evidence-based, which I really do not think it should grant an acceptance in the current form. However, if the over-interpretations is changed by a more accurate description of their findings, then I believe it could be a good contribution to the field. Second, the results section and figure legends should be properly edited. There are very pervasive inaccuracies throughout the article (in figures and figure legends), to an extent that makes reading and interpreting some of the results impossible. I have painstakingly gone line by line to help the authors communicate their ideas clearly.

To re-state the general issues that I have with this article:

1. Title. Authors have not shown that hsp16 tracks physiological states of proteome dosage. I do not think this is a paper about the heat shock response per se; however, the article presents very interesting insights into the prevalence and significance on the variation of protein expression capacity. The title should convey the prevalence of protein dosage (/protein expression capacity) and its relevance to health span in post reproductive animals. Generally, I do not believe that there has been substantial work in this article to disprove the entire notion of signalling noise in multicellular organisms, nor within the heat shock response. This would represent a very welcomed paradigm shift in the field. Their results are relevant to a very specific post-developmental time-point and using a period after heat shock where signalling noise may no longer be detectable. The authors did re-phrase their article in response to reviewers but from my point of view, their interpretations of their results are still misleading.

2. The article is not necessarily about the heat shock response. The conclusions tend to be inaccurate, as to the behaviours of hsp16 and the heat shock response in general. The conclusions are based on the behaviour of intestinal cells 24 hours after heat shock. Heat shock changes the ability of cells to produce and eliminate proteins, so the proteostasis will likely be different 24 hours after heat shock. In addition, the work does not show correlation between GFP measurements and native protein (i.e. Western blots). The remaining GFP expression may be more aligned with protein expression capacity than with the heat shock response itself at that point. It does not invalidate their conclusions on protein expression capacity, I simply would not write a paper about the chaperone system, I would write a paper about protein expression capacity.

3. Interpretation of the results using large number of observations. I appreciate that the authors obtained data from many animals and that this was challenging. I do not fully agree with the conclusion that G was the principal axis, as shown in Figure 4c, second panel, it is now possible to see dispersion of data that seems to be consistent with signalling noise. Line 185 of the main text, onwards. I don't understand why measurements of whole intestine taken from hsp16 do not carry bleed through technical issues as much as eft-3. The argument used to downplay the wider "P" axis is misleading, as hsp16 promoter should drive expression in many tissues, as much as eft-3.

Their favoured interpretation is that the increase in dispersion in Figure 4C second panel, is caused by technical inadequacies. However, it may be also telling us that expression may be dominated by different sources of variance depending on the tissue type. I do not believe that this interpretation causes their arguments on G necessarily incorrect. I suggest that when mechanisms are not clear, then it is more correct to keep interpretations open. I think that authors should write an article about G as a dominating source of noise in polyploidy tissues in post reproductive

animals. It still is meaningful, because at least one of the reporters that show this behaviour is a good predictor of lifespan.

4. What exactly is the "G" value that they are measuring?

The study was based on a handful of reporters, however, the authors did not provide evidence to support that they are measuring genes that are regulated independently (i.e., that they do not respond to common signalling pathways). In some cases, they observed co-regulation, for example, vit-2 is downregulated by heat shock (Figure S7B); however, their methodology does not allow them to capture this interaction, as the two genes show a positive correlation of hsp-16.1, in Figure 3b, driven by G and not P. Generally, can the methodology and protocol detect Pathway related variability? This is not clear, and therefore, it should be stated throughout the article that they are able to measure G and that G seems pervasive among genes tested, but that they cannot rule out that they are simply not able to measure Pathway-related variability using their protocol.

Unclear how does G relates to pathways that influence protein expression capacity and lifespan? If any reporter that varies with G predicts lifespan, then how do the authors link these claims with the broader literature that links protein synthesis and lifespan. Inhibiting growth related pathways, such as TOR signalling or Insulin signalling prolongs lifespan. If it were true that G positively correlates with extended lifespan, then either G is capturing opposite protein expression capacity than TOR. It would be ideal to determine how TOR does and ILS signalling alters their measurements of G. The full mechanism is a long-term project, but the article should acknowledge the complex implications of their findings. Very simple experiments with their TIMER-based reporter to provide further evidence.

I wonder if what they are observing has more to do with the overall epigenome status in aged animals, in line with <http://genesdev.cshlp.org/content/28/4/396.full> or with an ageing clock related to the overall health of animals.

5. This article is almost unreadable in its current form. The figures are not properly labelled and some of the legends are very lacking in details; I had trouble following some of the arguments because of this. I have gone figure by figure, explaining what my problems with them are:

Figure 2.

1. Bar-plots misrepresent data and scatter plots should be provided instead. This article explains why: <https://journals.plos.org/plosbiology/article?id=10.1371/journal.pbio.1002128>
2. Pearson correlations should only be used only if the distributions are normal. The normality of the distributions could be tested in whole worms. At least some of the reporters presented here do not show a normal distribution.
3. Do the reporters faithfully represent the endogenous genes? If so, state it.

Figure 3.

Eliminate lines 423 and 424, the results cannot be generalised to all genes in the genome. Unclear why two types of results are presented. In the upper graphs, variables were extracted from individual promoters and lower graphs, the variability is combined for two promoters. It is probably more insightful to present variability per promoter, unless a good rationale for figures e and f are presented.

In the main text, why is figure 4e discussed before all of figure 4? Probably figure 4e needs to be brought forward to figure 3.

Figure 4.

- (A) Is it possible to show values separated by promoter? Why is the statistics not shown in the graph? Label axis.
- (B) I find the 3D plots unclear, as the dots look highly dispersed. It would be better to show data similar to 4a, using scatter plots (and not bar plots).
- C and main text line 185: add Figure 4 c and d. It is very unclear what method was used to generate this data, from page 20 in material and methods, it appears to have been obtained from

full animals using a Leica dissecting microscope. Write the method used in the figure and clarify in the text.

Text associated with Figure 4.

179-187. Some Sorters allow accurate measurements of co-variance and the elimination of samples that are nonlinear or that preclude proper analysis. I would eliminate this comment from the main text, perhaps add it to materials and methods to explain why it was not used.

Line 185 onwards. The argument used is misleading, as hsp16 promoter should drive expression in many tissues, as much as eft-3.

A dissecting microscope should not be used to gather data from the intestine only, because, as explained, it is true that signal from other tissues can contribute to it. However, if the whole animal is measured then all pixels from the image can be gathered. The experiment should have been done with two promoters that are expressed ubiquitously or even using whole animal pixel intensity from eft-3P: GFP. For example, eft-3 and hsp-16.2.

What this experiment would tell is interesting information. If it is true that the signalling noise increases if pixels from the entire animal are used, then expression may be dominated by different sources of variance depending on the tissue type.

Figure 5. line 490. Be precise with statements. "Interindividual variation of hsp16.2 reporter" add: after heat shock.

In the figure, Mac-1 has not been introduced in the main text.

The section of "working models" would fit better if it were merged with the discussion.

Line 228, "In our working model, animals with higher concentration of hsp16.2" should read: "animals that express GFP for longer time, driven by the promoter of hsp6.2"

Line 231 therein. The model of how chaperones can alter G is not clear and should be spelled out clearly. If the authors suggesting that the heat shock response may be altering protein dosage, I agree with them. I would also add that it might be the heat shock response more broadly. This however, does not seem consistent with previous statements in the article.

Figure S5.

Every graph should be a scatter plot, showing individual experiment values and the axis properly labelled.

Figure S6.

Define ratio set point of expression. Label axis.

Figure S7. Change title of the figure, what does "proper" stand for.

A. What are the colours red and blue standing for, not explained anywhere?

B. What was the stats used, what are the two panels showing? If they represent two biological replicates, then a third one is necessary and the data represented as a scatter plot with statistic only performed on data with at least three biological replicates.

Figure S12 and main text from line 194.

Is it editorially correct to base main text on figures that are not presented in the main figure section? Probably Figure S12 has to be brought forward to the main figure section.

Line 204, brighter for what?

Line 206. There is a mistake as Figure S8 shows a completely different set of results.

I cannot fully gather the significance of Figure S12 because the figure is so poorly labelled and explained. Materials and methods has no information on the TIMER protein used.

However, from what I can take out of the figure it appears as if animals that produce more of new EFT-3 protein, also keep it for longer. This suggests that there are animals that have an increased

capacity to produce and keep proteins. This is a very interesting result.

I would eliminate panels b and c (unless they say something that I cannot fathom the importance of) and add a few extra experiments:

Two very good and easy experiments that can be done are:

1. Are animals with high EFT: timer-red/green good predictors of thermotolerance and/or hsp16p: gfp induction, lifespan, etc.
2. EFT: timer seems to suggest a broad variability across worms. Is this correlated with the expression of other reporters such as VIT-2?
3. Can VIT-2 expression predict hsp16 induction and/or EFT-timer?

Figure S12 title has a misspelling it should say proteins and not protein.

A. What do colours correspond to? My understanding of this figure is that if a worm produces more young protein it has also produced and/or maintained more protein than other worms for the last 2 days. It is not possible with this sort of experiment to distinguish the source of the old protein.

B. This figure is very difficult to understand. How was the experiment done, have animals been sorted by green/red and then heat shocked and tested for green/red accumulation? What are the asterisks representing? The bars do not appear different. Again, scatter plots should interchange bar charts. If the asterisks represent statistical significance, present the raw data as a table justifying the choice of statistical test.

C. This experiment does not show that the timer works as expected. The authors should include the reference where the protein turnover has been shown to decrease in older animals. One experiment to test the validity of the timer, would be to cross it with a temperature sensitive ama-1 allele, to determine if KO of transcription results in accumulation of older protein.

Additional Corroborating Data.

I am not familiar with the journal's policies on this sort of supplementary sections. Unless part of the results is presented in the main text, the section should be removed from the article.

Comments on supplementary material page 32.

On the results related to Figures S13 and S14.

These are interesting observations and they support their model, therefore should be brought (at least parts of it) to the main text. The fact that RAS GOF phenotype anti-correlate with chaperones later point than when the phenotype arises can be interpreted in two ways:

First chaperones can aid in the folding of a hypermorphic allele.

Second, those differences in protein dosage may explain the dose dependency of RAS (GOF).

Based on results from figure S12 eft-3: TIMER there seems to be animals with larger capacity to make and keep proteins than others. Their RAS-GOF results would argue that G is quite stable in time and can retroactively predict an earlier phenotype.

I am less keen to believe that this has to do with chaperoning activity. The heat shock driven fluorophores are measured 1 day after the heat shock, and it is unclear if chaperone levels remain high in those animals, but the fact that they express more GFP has to do with the fact that that animal had a large G capacity to begin with.

Supplementary Section 4.

Supplementary figure S17. It is unclear what is the point that is been made.

Supplementary S18. E and f, these are two independent experiments, though in the legend it says that three independent experiments had been done. Where is the third one? If there are not 3x then no statistic is possible.

What is exactly the point that is been made here? Given that daf2 and daf16 score similarly, it goes against the idea that insulin signalling modulates fecundity differently. I am not sure this

figure is necessary within the overall context of the article.

Comments to response to reviewer 2. I have indented these comments in the rebuttal letter as well [ED: attached].

Page 10, point 1. I appreciate that the authors have made the steady state point more clear in the methods and in the manuscript, however, as stated above, it is not how the message is read throughout the article, beginning with the title.

Page 10, point 1 a. I appreciate that the authors have added substantial amount of work to determine that their measurements are technically sound. I have no further comments on this aspect. However, I do have a problem with point 1a, 3, on page 11. "Interpretations of previously published work", and also in response to Reviewer 1, and further in the introduction.

In response to reviewer 1 Page 1. Paragraph 3 and throughout the introduction.

It is unclear how variability in protein expression capacity would correlate with phenotypes driven by loss of function mutations. The article cited 2012 Lehner article, which showed that hsp16 is a good predictor of l-o-f phenotypes, used a very different protocol. In that article, the authors showed that heat shocked animals during L3 larval stage could predict adult LOF phenotypes. In this article, the authors show here G is not stable over time, that G obtained from early (L1) larval stages does not predict G in adults. If that is the case, then it is not clear if generalisations are appropriate.

In response to reviewer 1 Page 4, Paragraph 3 and throughout the introduction.

The authors are again, inaccurate in their reading of the cited Lehner 2012 article showed that hsf-1 and hormesis rescued the effect of temperature sensitive alleles, but both partial and complete loss of function alleles, but not a cold-sensitive mutation. The same inaccuracy is repeated in lines 41 and 42 of the first page of the introduction. Again, that article cannot be used to provide evidence to support their model because the cited article used variable expression of hsp-16 in larval stages (as pointed above).

In response to reviewer 1 Page 5. Paragraph 1 and throughout the article.

If any reporter that varies with G predicts lifespan, then how do the authors link these claims with the broader literature that links protein synthesis and lifespan. Inhibiting growth related pathways, such as TOR signalling or Insulin signalling, prolongs lifespan. If it were true that G positively correlates with extended lifespan, then either G is capturing opposite protein expression capacity than TOR. If G was totally independent from TOR signalling, mutants in this pathway should not alter their main observations. I wonder if what they are observing has more to do with the overall epigenome status in aged animals, in line with <http://genesdev.cshlp.org/content/28/4/396.full> or with an ageing clock related to the overall health of animals. It would be ideal to determine how does TOR and ILS signalling alters their measurements of G. The full mechanism is a long-term project, but the article should acknowledge the complex implications of their findings.

Page 12, Response to concepts put forward in the reference PMID 22365828.

I disagree with the authors in that I do not believe that their manuscript clearly states what they are measuring and how that is different from the above reference. First, I think the reference should be included to this article. Second, it should be explained that their experimental set up allows them to directly measure G, but the fact that they do not measure pathway-related variability may be related to the type of reporters used and the timings used to approach the heat shock response. I am not convinced that " a considerable fraction of the proteome co-varies with natural variation of chaperones" They use GFP expression 24 hours after heat shock and the expression of Hsp-90, which as shown by Klosin and Lehner, 2017 is a reporter that aligns with piRNAs and not the heat shock response.

In reality, it does not really matter that they may not be measuring the heat shock response per se, because they care about it is that it is a biomarker that predicts lifespan. I have no problem with this observation, but I do struggle with generalisations with respect to the heat shock response and with signalling noise.

Page 14. I urge the authors to update their knowledge in the proteostasis field. I was referring to the proteostasis collapse described by Labaddia and Morimoto, 2014. A clear summary of the field of proteostasis during ageing can be seen in this figure:
<https://www.ncbi.nlm.nih.gov/pmc/articles/PMC3914504/figure/fig-001/>

I do believe that again, the collapse of the HSR may be relevant to the timing of their measurements. And this should be acknowledged.

Page 14, point 2. The authors point out that they do see interactions among different reporters. However, G still dominates, but I cannot make sense of their data. For example, why if vit-2 is downregulated by heat shock (Figure S7B), still presents a positive correlation of hsp-16.1, in Figure 3b? That interaction must be related to pathway related variability and yet in figure 3b, the dispersion is non-existent and the correlation is highly positive. It is difficult as a reader to make sense of this.

Page 15, 5th paragraph. If this work is not about the investigation of pathways, then it should be stated, that you were seeking to measure common sources of gene expression variance and did not design a study to capture Pathway variance.

Page 15, last paragraph. I appreciate that the authors obtained data from many animals and that this was challenging. I do not fully agree with the conclusion that G was the principal axis, as shown in Figure 4c, second panel, it is now possible to see dispersion of data that seems to be consistent with signalling noise.

Line 185 of the main text, onwards. I don't understand why measurements of whole intestine taken from hsp16 do not carry bleed through technical issues as much as eft-3. The argument used to downplay the wider "P" axis is misleading, as hsp16 promoter should drive expression in many tissues, as much as eft-3.

A dissecting microscope should not be used to gather data from the intestine only (cut through by using FIJI), because, as they are well aware, it is true that signal from other tissues can contribute to it. However, if the whole animal is measured then all pixels from the image can be gathered. The experiment should have been done with two promoters that are expressed ubiquitously or even using whole animal pixel intensity from eft-3P:GFP. For example, eft-3 and hsp-16.2.

Their favoured interpretation is that the increase in dispersion in Figure 4C second panel, is caused by technical inadequacies. However, it may be also telling us that expression may be dominated by different sources of variance depending on the tissue type. I do not believe that this interpretation causes their arguments on G necessarily incorrect. I suggest that when mechanisms are not clear, then it is more correct to keep interpretations open. I think that authors should write an article about G as a dominating source of noise in polyploidy tissues in post reproductive animals. It still is meaningful, because at least one of the reporters that show this behaviour is a good predictor of lifespan.

Page 16, comment on Figure 4B.

I take the point with regards to Ras GOF and its effect on the reporter (Do they see that the reporter expression goes down in expression compared to controls?, perhaps be worth making a

note in figure legends). I think that their incompletely penetrant data should go back to main results. The authors should tone down conclusions, because they have not been able to show that G is constant over developmental time (rather the opposite if one takes earlier stages). But since they cannot base their conclusions on previously published data, and they did not provide fresh data on LOF incomplete penetrance, then the GOF is what they can use.

Page 17. I can agree with the publication of an article without proper mechanistic understanding of what G really is about. However, the article must be accurate. In the last paragraph of page 17 and throughout their paper, they continue to say that "high abundance of chaperones correlate with high abundance of other cellular proteins". If this is what they want to claim, then they have to show that

1. Their measurements of GFP from a hsp16 reporter 24 hours after heat shock accurately represents measurements of chaperones. This should be done using western blots that GFP corresponds with the levels of endogenous chaperones.

2. They should include HSF1 over-expression experiments, where they pump up the levels of chaperones to determine if RAS GOF depends on Hsf1 activity.

If the authors do not wish to continue doing experiments, then they should be careful with each and every statement in the article. I would say that they should treat hsp16:GFP as a biomarker that correlates with lifespan. They have not shown properly that in their protocol, they are dealing with chaperones.

Page 18, on their re-interpretation of the Science 2012, Lehner article.

That article does not show molecular evidence that the chaperone refractive alleles are genetic nulls. A careful reading of supplementary figures shows that the alleles that are refractive to Hsf-1 are cold sensitive. There is no genetic evidence for any of such alleles to be true nulls. Second, the authors argument that elevated heat shock increases the penetrance of such alleles, is opposite to what was shown in that article.

Reviewer comments are highlighted by gray. Author responses are in Courier New font. Revised text from the manuscript shown here in this reviewer response is in Arial font and contains references in explicit named format in order to make references easier to find and review. Corresponding text in manuscript is appropriately formatted for Nature numerical reference style.

Reviewers' comments:

Reviewer #1 (Remarks to the Author):

The referees have addressed most of my concerns, as well as those of referee 3. I think this is an interesting and important study that should be published without delay.

Suggestions to improve the presentation:

[1] The last sentence of the abstract is very speculative. There is little evidence that variation in protein dosage or (endogenous) chaperone activity is inherited in most animals, particularly in humans. So whilst it is an interesting idea that this may contribute to the heritability of traits, this isn't the question addressed in this manuscript and I think this is best left to the discussion rather than the abstract.

We thank the reviewer for pointing this out. We concur. We have removed it from the revised abstract.

[2] line 42 and elsewhere - this sentence is incorrect. In yeast (and worms), variation in chaperone activity (including hsp90) can modify the outcome of null alleles. e.g. Zhao et al. Cell 2005 120:715. (one assumes by an indirect mechanism).

We thank the reviewer. We deleted this. As the reviewer points out, the words "but not null mutants" are incorrect, and were actually an editing error on our part, moved from a part of the manuscript referring to *hsf-1* overexpression.

There is more to be said on this topic. We are aware of *hsp82* alleles affecting the growth phenotypes of deletions in *cerevisiae*, but we are unwilling to equate those effects with the reported covariation of natural hsp90 variation (indicated by a reporter) and the penetrance of discrete traits. We are also of course aware that *hsp-90 /daf-21* reporters can predict the penetrance and expressivity of *tbx-9* null mutants, but we ascribe that to their effect on expression of the redundant gene *tbx-8*, essentially a duplication. We view our interpretation of the *daf-21* expression data as an important point, but are saving further discussion of it for a detailed review, beyond the scope of this report.

[3] line 95 'Here we adapted an approach we used in yeast18'. I understand that this is

correct, but the authors' approach is an extension of the intrinsic - extrinsic noise distinction pioneered by other labs so I think broader citations would be fairer here.

We thank the reviewer for this refining suggestion. In the previous version of the manuscript, we cited Elowitz and Swain, and Raser and O'Shea later in the introduction, but we now cite them more appropriately at the first mention of intrinsic noise.

Revised text now reads:

" Here, we adapted and extended an experimental design and analytical framework we developed in yeast (COLMAN-LERNER *et al.* 2005), to quantify sources of variation in gene expression in a metazoan. This analytical framework is an expansion of the intrinsic/extrinsic noise framework pioneered in *E. coli* (ELOWITZ *et al.* 2002). The concept of intrinsic noise in gene expression arose from measuring the correlation between two differently colored but otherwise identical reporter genes in work in prokaryotic bacteria by Elowitz and Swain (ELOWITZ *et al.* 2002). This approach for quantifying intrinsic noise was then adapted by Raser and O'Shea to quantify variation in the expression of differently colored alleles in eukaryotic single celled yeast (RASER AND O'SHEA 2004). Our expanded experimental and analytical framework allowed us to distinguish among three hypotheses for how differences in *hsp-16.2* reporter expression might arise. The three hypotheses were that the differences in biomarker expression level arose from intrinsic noise, signaling noise or differences in general protein expression capacity. The first hypothesis was that differences in the *hsp-16.2* lifespan biomarker might arise from differences in intrinsic noise in gene expression. Previous work with human autosomal genes (GIMELBRANT *et al.* 2007) showed that individual cells may only express much less of, or only one, of their two distinct copies of each allele. Therefore, animals might express more or less of a gene by expressing different amounts of each allele – anywhere from full expression of both alleles to no expression of either allele."

[4] It would be useful to have a figure that more explicitly quantifies how much of the variation is accounted for by variation between the cell types (rings) and how well correlated the between individual variation is comparing different rings across the same individuals.

We added figure 4f to show the quantified average expression of cells in particular (the first three) intestine rings of individual animals' intestines in a scatter plot. The plot shows contribution of cells in individual rings from individual animals in the image from 4e.

[5] The working models section is discussion and so should be moved from the results to the discussion section.

We thank the reviewer and have moved this section to the discussion.

[6] I don't really understand why the authors seem to favour a model in which variation in chaperone activity causes variation in protein production rather than being simply part of it /a consequence of it.

We think that an initial transcriptional difference in chaperone expression is some part of the model because we detect transcriptional differences when animals are expressing more biomarker, and because neurons can regulate the proteostatic machinery of intestines. We think that the chaperone activity increase is because of the transcriptional difference and that it plays out at the protein level as a change in the abundance of at least properly folded, biologically active proteins or maybe total protein.

[7] ‘These differences can arise from epigenetic differences²⁸ and seemingly stochastic environmental perception differences^{14,15}.’ It would be good to unpack this rather vague sentence.

We thank the reviewer for this clarifying suggestion.

The new text reads “**How intestines may end up in high protein dosage states after heat shock.** The natural variation in the *C. elegans* chaperone system is a property that is consequential, and likely selected for over geological time, if we subscribe to the prior conclusions of Klaus Gartner when he was pondering the origins of nongenetic variation in murine systems (GARTNER 1990); it seems reasonable to do so. In our working model, animals with higher concentrations of the *hsp-16.2* lifespan/penetrance biomarker in their intestine cells have higher concentrations of other proteins in those cells.

In this model, differences might arise from epigenetic differences (CYPSEY *et al.* 2013) and seemingly stochastic environmental perception differences (MENDENHALL *et al.* 2017a; MENDENHALL *et al.* 2017b). In support of this idea, we previously found that animals whose parents had been heat shocked express more chaperone biomarker (CYPSEY *et al.* 2013). Moreover, we also previously found that thermosensory neuron function is required for generating biomarker differences between individuals after heat shock¹⁴. In addition, we previously found that animals at different temperatures have different amounts of lifespan biomarker variation and lifespan variation¹⁵. Finally, in the same report in which we examined heritability, we also found that whole animals that make more biomarker have significantly elevated transcripts for five chaperones including *hsp-16.2* (CYPSEY *et al.* 2013). These previous studies led us to believe that differences in the life history of the parents and differences in the perceived or real environment can lead to differences in heat shock protein biomarker expression. In particular, the fact that we previously detected transcriptional differences between animals that had more and less of the *hsp-16.2* biomarker 24 hours after heat shock, suggests that there was an initial thermosensory neuron signaling difference that manifested at the transcript level. But, because chaperones generally affect protein production and turnover, when an intestine produces more chaperones, that increased abundance of chaperones causes a generally greater abundance of other proteins in its cells. Our results suggest that this is from both increased production and decreased turnover. That is, in this view, the initial signaling difference we think comes from neurons results in a physiological state of high protein expression capacity or high G in the peripheral intestine tissue, and possibly other tissues.”

[8] ‘Furthermore, understanding this ... kind of noise in gene expression may have implications for understanding how people can live with and escape the consequences of

dominant negative mutations, like some oncogenes⁴⁰.’ Again, I can guess at what the authors mean but it would be better to be more explicit.

We thank the reviewer. We have revised to try to make clearer. The new text reads: “Furthermore, understanding this kind of noise in gene expression may have implications for understanding how people can live with and escape the consequences of dominant negative mutations, like some oncogenes (MARTINCORENA *et al.* 2018). Specifically, silencing a dominant negative allele can prevent the manifestation of the associated negative trait(s).”

[9] ‘And we know that chaperone levels can be epigenetically heritable²⁸.’ Most readers will not be familiar with what was shown in ref 28 and how general this result is. It would be helpful to be more precise and to state whether such conclusions have been replicated by other labs or observed in other species.

We thank the reviewer for spotting this. We now cite the Brunet lab study of the grandchildren and subsequent generations descended from *wdr-5* mutants. These F4/F5 animals inherited differences in chaperone expression level. And, locusts can transmit chaperone level differences to their progeny too.

The revised text now reads... “And we know that chaperone levels can be epigenetically heritable, at least in *C. elegans* (GREER *et al.* 2011; CYPSEY *et al.* 2013) and locusts (CHEN *et al.* 2015).”

Reviewer #2 (Remarks to the Author):

The authors present a very useful statistical framework to study different sources of variability in a metazoan model organism. They also make some interesting observations regarding protein expression capacity and its relevance for health span. The authors addressed some of the issues raised by reviewers and included a few new figures to give more substance to their claims. In particular, they have used an alternative method to measure variability across worms, which allows them to increase the number of observations, a sticking point for any study that claims to be studying variability. The authors have added additional reporters and a TIMER based reporter, to define the behaviour of the reporters over time.

Generally, these results should be enough to grant them an acceptance. However, I still have issues with the interpretation of the results and as written, I do not think [this paper] should be published. The authors claim that the findings have been more accurately presented. I would say that the main text of the article does read better than the previous version; however, [first] there are so many inaccuracies and claims that are not evidence-based, which I really do not think it should grant an acceptance [sic] in the current form. However, if the over-interpretations is changed by a more accurate description of their findings, then I believe it could be a good contribution to the field. Second, the results section and figure legends should be properly edited. There are very pervasive

inaccuracies throughout the article (in figures and figure legends), to an extent that makes reading and interpreting some of the results impossible. I have painstakingly gone line by line to help the authors communicate their ideas clearly.

We are grateful for the time and effort Reviewer 2 committed to our revised manuscript. We are of course saddened by the fact that after this effort, Reviewer 2 still found issues with the manuscript, including: our statements about previous work, our results, and our conclusions. In many of the criticisms, Reviewer 2 asserts that our stated facts and conclusions are inaccurate, or that they not supported by the evidence, or both. Perhaps understandably, we believe our knowledge of the relevant literature, our own experimental results, and our inferences from our results to be sound. We believe that this reviewer's objections are best dealt with in our point-by-point response below.

In the process of thinking through Reviewer 2's comments, we have also made what we consider to be a good guess for technical reasons involving different means of collecting data that might explain why Reviewer 2's results might not match ours. If our guess is correct, it might in turn explain why Reviewer 2 seems to reject our main conclusion, that global differences in protein expression capacity account for most of the difference in expression of the $P_{hsp-16.2}$ biomarker. We will present that possible explanation below.

To re-state the general issues that I have with this article:

1. Title. Authors have not shown that hsp16 tracks physiological states of proteome dosage. I do not think this is a paper about the heat shock response per se; however, the article presents very interesting insights into the prevalence and significance on the variation of protein expression capacity. The title should convey the prevalence of protein dosage (/protein expression capacity) and its relevance to health span in post reproductive animals.

We thank the reviewer. In response, changed the title to "hsp-16.2 biomarkers track proteome dosage", which we believe better reflects our data and addresses this reviewer's concern.

Actually, these are not post-reproductive adults. They are at the peak of reproduction (self-fertility) for hermaphrodites at 20°. We revised the abstract to more clearly state the point in the lifecycle of the worm at which we focused our measurements (young adult animals).

We enthusiastically agree that the relationship of protein and proteome dosage with healthspan is an important aspect to consider. We are also excited about this possibility. However, because our experiments did not directly measure healthspan, and because it is a complex topic

supported by many different publications consisting of many different kinds of experiments, we prefer to address this in another more focused review.

Generally, I do not believe that there has been substantial work in this article to disprove the entire notion of signalling noise in multicellular organisms, nor within the heat shock response. This would represent a very welcomed paradigm shift in the field.

We believe we understand this reviewer's comment, and if we do, we agree with it of course. That is, this work does not disprove the entire notion of signaling noise.

We start by pointing out that the study of non-genetic variation has been ill-served by some of its terminology, including the early introduction of the term "noise" to describe variation among genetically identical cells, and by the imprecise use of the term "extrinsic noise" to refer to different kinds of cell-cell variation. The insight in the analytical framework we are using used here (Colman-Lerner et al, Nature 2005) is that it provides a way to define "extrinsic noise", and to decompose and quantify the contributions to that variation from two distinct components, variation in signaling, $\eta^2(P)$, and variation in general gene expression power $\eta^2(G)$. In young adult animals, we can and do observe variation (Figure 3) in signaling, and we can and do quantify it (S). Our results are unambiguous, and they clearly indicate that for the genes we studied, in the cells of the young adult animals we study, variation in signaling is indeed low.

That said, again, we agree with the reviewer that we have not disproven the entire notion of signaling noise in multicellular organisms, nor even within the heat shock response. In our model, we detail that we think signaling variation happens in the neurons.

We show that we do see some signaling variation, just that it is not dominant, in revised Figures 3 and 4 and in the revised results section.

For additional examples of, and methods to detect, signaling noise, please also refer to Figure 1 in this reviewer response, and Figure 4g (a global view), Figs S8-10 (some noisy in ring 1 cells) and Supplementary Information section "**S2.5 Two Color Variants Driven by Different Promoters and Pathway Variation**" in the revised manuscript.

In the revised maintext, we now state explicitly that we might detect more signaling noise with additional reporter genes and that we do not know the fraction of the proteome that covaries with all the reporters we examined in this report, just that the variation is significant and consequential.

The major relevant text in the revised discussion reads:

“The high correlation of chaperone reporters with phenotypes (Figs. S13-15 and (REA *et al.* 2005; YANG AND TOWER 2009; CASANUEVA *et al.* 2011; MENDENHALL *et al.* 2012)), distinctly regulated transcriptional reporters (Fig. 3), fusion proteins (Fig. 4) and knockins (Fig. 4) suggests that at least some significant fraction of cellular proteome covaries in some circumstances. This is worth considering in the larger context of any biological scenario. However, we do not yet know what fraction of the cellular proteome for which protein dosage correlates with chaperone abundance. Our experimental system did detect gene expression levels changing in response to external signals (Fig. S7), and instances of clearly detectable cell autonomous intrinsic noise and signaling noise (for intrinsic noise see Fig. S4 *hsp-90* ring 4; for signaling noise see Fig. S9 *hsp-16.2* & *mtl-2* ring 1, and *hsp-17* & *vit-2* ring 1). However, for the most part, the small fraction of the genome we examined covaried fairly well (e.g., Figs. 3&4). Presumably, additional work exploring variation in the expression or activity of the myriad of remaining genes will identify additional axes of variation that will help define specific fractions/modules of the proteome that covary in different scenarios. For example, Sanchez-Blanco *et al.* found another, uncorrelated axes of variation using a fluorescent protein reporters (SANCHEZ-BLANCO AND KIM 2011). While most reporters they looked at in eight day old adult worm intestines were highly correlated (e.g, $r = 0.4-0.87$), one single reporter, controlled by the promoter for *C26B9.5*, which encodes a serine protease, did not correlate well(SANCHEZ-BLANCO AND KIM 2011). While we did not see additional major axes of uncorrelated variation, it may be possible to find other axes of G, or significant differences in P, with additional fluorescent reporter genes or fluorescent knockins.”

Their results are relevant to a very specific post-developmental time-point and using a period after heat shock where signaling noise may no longer be detectable.

Yes. Again, this is so. In the text we are explicitly state at multiple points that we are examining at specific intestine cells at a specific point in the lifecycle of the animal, the two day old adult animal.

For examples:

The new abstract text reads...” Here, we used an *in vivo* microscopy approach to dissect the mechanisms of covariation in *hsp-16.2* biomarker, focusing on the intestines of adult animals, which generate the most signal when predicting lifespan. ”

And the last paragraph of the results now reads: “In Supplementary Fig. S17 we show that intestine cells in larvae show differences in G, but also have seemingly autonomous bursts of protein expression, which we do not see in two-day old adults, and will require further study to understand how these bursts fit in the context of developmental progression and other kinds of noise in gene expression.”

For another example, in the discussion we also note the age of the animals we are measuring, which now reads, “**Intestinal protein expression capacity during development and aging.** We examined sources of variation in gene expression in the intestine cells of two-day old adult *C. elegans* that had or had not been heat shocked. For the non-heat shocked animals, this was the peak of reproductive output (MCCARTER *et al.* 1999).”

Importantly, we also note that, at this timepoint, we do detect and quantify signaling variation. Figures 3 and 4 show that we do detect and quantify some signaling noise, but that it is not the major source of cell-to-cell variation in gene expression. Figure 3 shows that in most measurements, signaling noise is about an order of magnitude lower. Figure 4 shows that at animal or cell resolution signaling noise is also relatively low, using both bar graphs and scatter plots, including a new scatter plot showing all cells from all experiments.

It is worth noting that we did detect some instances of fairly high signaling noise for some gene pairs in some cells. This is seen in scatter plots Figs S8-10 (some noisy cells in ring 1). In addition, Figure 1 in this reviewer response, and Figure 4g (a global view) show images and a global view of signaling noise, respectively.

(Again, we describe how we calculate signaling noise from experimental measurements in Supplementary Information section **“S2.5 Two Color Variants Driven by Different Promoters and Pathway Variation”** in the revised manuscript)

The revised main text notes these instances of higher signaling noise.

For example, a relevant passage in the maintext now reads:

Results: “Supplementary Figs. S8-S10 show individual scatter plots for Type II experiments and bar graphs of individual cell types. Supplementary Fig. S9 also shows that some cells have relatively more or less noise through some pathways, such as higher signaling noise observed in the intestine cells of ring 1 (for signaling noise see Fig. S9 *hsp-16.2* & *mtl-2* ring 1, and *hsp-17* & *vit-2* ring 1). Thus, this experimental system is capable of detecting differences in signaling at the cell and population levels, we just did not detect a large degree of signaling noise in young adult intestine cells.”

And another passage reads

Discussion: “Supplementary Figs. S8-S10 show individual scatter plots for Type II experiments and bar graphs of individual cell types. Supplementary Fig. S9 also shows that some cells have relatively more or less noise through some pathways, such as higher signaling noise observed in the intestine cells of ring 1 (for signaling noise see Fig. S9 *hsp-16.2* & *mtl-2* ring 1, and *hsp-17* & *vit-2* ring 1). Thus, this experimental system is capable of detecting differences in signaling at the cell and population levels, we just did not detect a large degree of signaling noise in young adult intestine cells.”

The authors did re-phrase their article in response to reviewers but from my point of view, their interpretations of their results are still misleading.

We are not sure what the general statement about our interpretation of our results being misleading refers to but we respond to individual points below.

[REDACTED]

Figure 1. G as a major axis of variation, it is the dominant axis of variation for cells in two-day old adult worm intestine tissue, and we detect and quantify cell autonomous signaling noise in worms. Scatter plot in top left panel shows the major result from Elowitz and Swain in 2002. The intrinsic component is significant in bacteria (left panel), but in yeast – the top row right panel, the extrinsic component is relatively greater – from Raser and O’Shea, 2004. This is shown by the data points for allele expression values clustering along the 45° diagonal. The second row, left and right panels show that later, in 2005, we decomposed the extrinsic component into signaling noise and gene expression capacity. Gene expression capacity is the contribution to the extrinsic component shown by correlated variation, evident in these scatter plots of two distinctly regulated reporters. In these plots this source dominates cell-to-cell variation in yeast. It also (next row) dominates cell-to-cell variation in worm intestine cells, shown here in the 3rd row with and without major axes labeled; this plot shows almost 4,000 hand measured cells from over 500 animals. Bottom row shows images of worms where arrows point to cells redder or greener than the yellow composite of the two reporters. These deviations from yellow indicate these cells show significant variation for type I (left for *hsp-90*) and type II (right, for *hsp90* and *vit-2*) variation in gene expression.

2. The article is not necessarily about the heat shock response. The conclusions tend to be inaccurate, as to the behaviours of hsp16 and the heat shock response in general.

We agree with the reviewer's first point. We don't view the heat shock response as the object of our study. For that reason, we don't mention the heat shock response in the title or make it a major object of the paper.

To the second point, that **"The conclusions tend to be inaccurate, as to the behaviours of hsp16 and the heat shock response in general"**

To respond, we will begin by repeating that we have examined reporter genes activated by different signaling pathways in numerous experiments, and observe in all cases that in cells of the tissue of interest (*C. elegans* intestine) both intrinsic noise $\eta^2(\gamma)$ and signaling noise $\eta^2(P)$ are low. Quantifying these contributions to variation requires careful microscopic measurements of fluorescence from cells in individual animals. We note that some of the methods we devised and used here are actually higher throughput than our previous confocal methods (Mendenhall et al. 2015), but acknowledge that they are labor intensive. The requirement for this careful microscopic quantification from individual cells in individual animals to get accurate measurements limited the number of cells we can examine to a few thousand. We admit this technical limitation in the manuscript. But these numbers are sufficient to give us statistically sound results for the reporters examined, which span diverse modes of regulation, and of the diverse phenotypes examined. We assert that the breadth of the ten transcriptional reporter and protein fusion genes that we have examined does allow us to make generalizing conclusions about the relative contributions of different sources of cell-cell variation, even if those conclusions are limited to the genes and tissues and developmental times we examined, and that they certainly also enable us to speculate as to the relative contributions of different sources of variation in other tissues and at other developmental times.

We give specific responses to the statement that "[our] conclusions tend to be inaccurate" in the context of specific articulated objections by the reviewer below.

The conclusions are based on the behaviour of intestinal cells 24 hours after heat shock. Heat shock changes the ability of cells to produce and eliminate proteins, so the proteostasis will may likely be different 24 hours after heat shock.

We take this statement by the reviewer to mean that the GFP signal might be affected by changes that the heat shock response causes in cells of the organism-- that, for example, that heat shock might change protein degradation-- and of course we agree.

In what is possibly the first important relevant paper, Link et al (1999) showed that expression of *native* HSP-16.2 protein peaks 24 hours after heat shock.

In *C. elegans*, the use of GFP transcriptional reporters and protein fusions to quantify aspects of the heat shock response has a long history Rea et al. (2005), and is widely used Casaneuva et al. (2012), Link et al (1999), Mendenhall et al (2012).

Here, we used this particular measure of the heat shock response-- GFP signal from a *Phsp-16.2::GFP* reporter 24 hours after heat shock-- precisely because in much

previous work (Link et al 1999) showed that this is when HSP-16.2 expression peaks, and we have shown that the differences in signal from *hsp-16.2* reporters at this time can predict phenotypes (see for example (Rea et al. 2005, Mendenhall et al 2012). Among the phenotypes predicted by this signal, this biomarker, is the particularly interesting phenotype of organismic lifespan. We measure it in adults in this work because we wish to understand the relation between differences in expression of this biomarker and its ability to predict other interesting phenotypes like resistance to lethal thermal stress.

Of course, we agree that heat shock changes proteostasis by inducing the expression of chaperones. We also note that we examined just as many non heat shocked animals and found the same kinds of minor cell autonomous deviations attributable to intrinsic noise and signaling noise, but that the major axis was still G.

In addition, the work does not show correlation between GFP measurements and native protein (i.e. Western blots).

Above, we have laid out and referenced the established use of GFP signal from this reporter gene at this time point as a means to quantify the heat shock response. We think the reviewer means that we have not shown the reporter gene correlates with the gene it is reporting on.

To address the first hypothesis about the reviewer's objection, we have changed the text to reference previous work showing that both the *hsp-16.2* and *hsp-90* reporters correlate with the chaperones they intend to report on.

The revised text cites papers that established these facts. The revised text now reads... ``As expected, but important to note, both *hsp-90*(BURGA *et al.* 2011) and *hsp-16.2*(REA *et al.* 2005) reporters properly correlate with the expression levels of the chaperones on which they are reporting (*hsp-16.2* & *hsp-90*).''

The remaining GFP expression may be more aligned with protein expression capacity than with the heat shock response itself at that point.

[REDACTED]

Figure 2. And another reason Reviewer 2 seems generally confused about the timing of the heat shock response and GFP. Reviewer 2 suggests that we are measuring gene expression at an irrelevant time relative for the heat shock, when actually we are measuring gene expression at the peak of *hsp-16.2* chaperone expression. Image from Chris Link's lab's Direct Observation of Stress Response in *C. elegans* using a reporter transgene, 1999 in *Cell Stress and Chaperones*.

We are not sure what the reviewer means by "remaining GFP expression". We take it to mean "GFP signal 24 hours after heat shock", in which case the comment means that such GFP signal might be affected by global changes in proteostasis in cells, changes caused by the heat shock. We agree.

We should say again that we measure GFP signal after 24 hours because it is an established measure of heat shock, and because ample evidence establishes a correlation between expression of *hsp-16.2* reporters and expression of the native protein, and because *this specific measure* has been shown multiple times to predict phenotypes of interest to us and others (stress resistance and lifespan).

It does not invalidate their conclusions on protein expression capacity, I simply would not write a paper about the chaperone system, I would write a paper about protein expression capacity.

Again, by title and abstract, we did not intend to convey that this paper was a study of either the heat shock response system or the chaperone system per se. However, our study links expression of this heat shock reporter to cells' protein dosage (expression capacity). We wrote the paper we meant to write, and its title, "*hsp-16.2* chaperone biomarkers track proteome dosage", reflects the content of our work.

3. Interpretation of the results using large number of observations. I appreciate that the authors obtained data from many animals and that this was challenging.

We thank the reviewer. We found this to be hard work but work eminently worth doing.

I do not fully agree with the conclusion that G was the principal axis, as shown in Figure 4c, second panel, it is now possible to see dispersion of data that seems to be consistent with signalling noise.

We show here in Figure 3 in this reviewer response, which relates to Figures 4c&d in the revised manuscript, that coefficients of correlation 0.811 (for *vit-2* and *hsp-16.2*) and 0.703 (for *eft-3* and *vit-2*) clearly allow us to speak of G as the main axis of variation.

Line 185 of the main text, onwards. I don't understand why measurements of whole intestine taken from *hsp16* do not carry bleed through technical issues as much as *eft-3*.

We thank the reviewer for this comment. Regarding the

Figure 3. In scatter plots of large numbers of animals, most animal's epifluorescently collected signals line up near the trendline, and at cell resolution, the correlation is even more evident. Our original manuscript only included the cell resolution data, which lacks the contaminating signals from the other tissues, and we thus consider it a better measure. Figure 1 in this reviewer response and 4g and S8-10 in the revised manuscript demonstrate that we in fact do detect some signaling noise, just that it is not dominant. We have included the lower resolution data non cell resolution data, even though it is confounded with extraneous signal, at the request of reviewer 2's comments on the previous manuscript, who suggested we look at more animals at lower resolution.

bleedthroughs, here, and in all experiments where we quantify signal at the whole animal level via the epifluorescence microscope methods we detail here (as opposed to the more laborious confocal microscope used for cell resolution measures (Mendenhall et al. 2015)) the intestine always makes the largest contribution to overall *hsp-16.2* signal (Seewald et al 2010). Using the same whole animal imaging methods, the intestinal *eft-3* signal is significantly more convolved with signal from hypodermis and muscles. Hence, our lower correlation between *eft-3* and *vit-2* signal obtained using whole animal (segmented intestine) epifluorescence microscopy is at least partially due to higher contribution of other out of plane tissues to *eft-3* signal. Reviewer response Figure 3 shows these results. In Figure 3, we show cell resolution correlations for this reporter gene pair that are much greater, consistent with the idea that the lower correlation at animal resolution is from out of plane signals.

The argument used to downplay the wider “P” axis is misleading, as *hsp16* promoter should drive expression in many tissues, as much as *eft-3*.

Although it is true that *hsp-16.2* promoter drives expression in many tissues, it is not expressed to the same degree in all tissues. In fact, the overwhelming bulk of *hsp-16.2* reporter gene expression ($\approx 90\%$) comes from the intestine. We quantified this and established this fact in 2010 in Seewald et al. We quantified it again in 2012 with single copy reporters, in Mendenhall et al, and again in 2015 in Mendenhall et al. Other investigators have found the same thing. For example, in the images of larval animals expressing *hsp-16.2* reporters in Casanueva et al, the dominance of the intestinal signal is also apparent.

We revised the text to make this point clearer. The new text reads:

“At Cell and Animal Resolution, Most Variation in Intestine Cell Gene Expression is Due to Variation in Protein Expression Capacity. To confirm that the differences in general protein dosage we saw at cell resolution constitute the major axis of variation in gene expression, we examined large populations of animals. This was not a straightforward experiment.

When we tried to quantify covariation between signals in unconstrained animals in the COPAS Biosort, or in a microfluidic, image-based quantification device we developed (CRANE *et al.* 2019), we found little covariation between genes we know to be highly correlated inside individual intestine cells. This fact was due to the rotational freedom and variable angular positioning of the animals along their long axes with respect to the detectors. This artefact allowed the higher throughput devices to get the right answer for the average expression level of the two different reporters, but not to detect covariation within a tissue. To circumvent these problems, we developed means to mount large numbers of animals in soft agarose pads, anesthetizing them and then imaging them, constrained in the agarose, using an objective with a large field of view.

When we examined distinctly regulated reporters in hundreds of animals, we found that animal-to-animal variation in G was dominant. Figs. 4c&d, the same trend for differences in G can be seen, even at animal resolution, provided animals are constrained and a tissue specific marker is used to extract relatively purer signal from a focused region of interest (the intestine tissue). Signals from other tissues differentially contributed to the intestine signal we measured using epifluorescent scope method to examine larger populations of animals. We believe the deviations from the trendline with the *eft-3* and *vit-2* reporters are from out of plane non-intestinal signals contributing to the intestine signal, and remaining rotational differences in mounting. We think the *vit-2* and *hsp-16.2* reporters are more representative of what we see at the cell level because both reporters are mostly expressed in the intestine (90% for *hsp-16.2* reporters in adults (SEEWALD *et al.* 2010; MENDENHALL *et al.* 2015) and exclusively in the intestine for *vit-2*), whereas *eft-3* reporter signal is also relatively stronger in muscle and hypodermal tissues surrounding the intestine, compared to *hsp-16.2* reporter signal. To test the idea that

the uncorrelated variation was attributable to out of plane signal from other tissue, we analyzed the cell resolution expression data from the 193 cells we measured from animals expressing the same *vit-2* and *eft-3* reporters and found the correlation to be $r = 0.927$.”

Their favoured interpretation is that the increase in dispersion in Figure 4C second panel, is caused by technical inadequacies.

We believe the reviewer means Figure 4d. And we believe that the cause of apparent increased non-G variation in 4d is due to the rotational artefact described above, and the intestine specific expression of *vit-2* vs the general expression of *eft-3* as above. Between them, these factors explain the [slightly] lower correlation between *vit-2* and *eft-3* ($r = 0.703$). This is consistent with the observed higher correlation from a smaller number of precise measurements of individual cells, shown in the scatter plot of the same two genes, measured at single cell resolution, giving an r of 0.927, meaning 86% of the expression level of one reporter is explained by the expression level of the other in a given cell.

Again, see reviewer response Figure 3 and the text changes we made in response above.

However, it may be also telling us that expression may be dominated by different sources of variance depending on the tissue type.

Indeed, expression may be dominated by different sources of variation depending on cell and tissue type. In fact, in unpublished experiments, we have evidence that intrinsic noise may well be dominant (in striated muscle, Sands et al). But, we stress again, that in the cells we intended to study, in the animals we studied, in all cases we find G, correlated differences in expression of reporter genes and thus a measurement of general gene expression capacity, to be the dominant source of cell-cell variation. To repeat, cell-cell differences in expression are dominated by G in the intestine cells. We discuss explicitly that other kinds of cell-cell variation, including signaling noise, may make a greater contribution in diploid tissues.

\
In the revised text we discuss this possibility with the following text:

“Mechanisms of cell-to-cell variation in gene expression in other tissues. Other reports have shown that intrinsic noise can be a significant component of cell-to-cell variation in gene expression. In yeast, intrinsic noise was an order of magnitude higher than in worm intestine cells (RASER AND O'SHEA 2004).”

I do not believe that this interpretation causes their arguments on G necessarily [to become] incorrect. I suggest that when mechanisms are not clear, then it is more correct to keep interpretations open.

We are not altogether sure what the reviewer means by this comment. We should emphasize here that we know very little about mechanisms that account for differences in G. All our work shows is that these differences in G are the preponderant source of differences for reporter genes in the tissues studied at the time studies.

We now wrote the discussion to emphasize this and our own ignorance as to the fraction of the proteome that covaries with G.

Revised text reads:

“The high correlation of chaperone reporters with phenotypes (Figs. S13-15 and (REA *et al.* 2005; YANG AND TOWER 2009; CASANUEVA *et al.* 2011; MENDENHALL *et al.* 2012)), distinctly regulated transcriptional reporters (Fig. 3), fusion proteins (Fig. 4) and knockins (Fig. 4) suggests that at least some significant fraction of cellular proteome covaries with chaperone biomarkers in some circumstances. This is worth considering in the larger context of any biological scenario. However, we do not yet know what fraction of the cellular proteome for which protein dosage correlates with chaperone abundance.”

I think that authors should write an article about G as a dominating source of noise in polyploidy tissues in post reproductive animals. It still is meaningful, because at least one of the reporters that show this behaviour is a good predictor of lifespan.

In this article, we state that from our experiments, we see that G is the dominating source of cell-cell variation in the cells in the young adult animals that we observed. It is the dominant contribution to variation in the polyploid tissues (intestine and X) that we measure. So we agree with the reviewer that G is the dominating source of cell-cell variation.

We agree that these differences in G are meaningful, one reason being that differences in expression of the *Phsp-16.2::GFP* reporter correlate with differences in lifespan. This correlation sparked our original interest in these issues, and we write about it in our introduction.

We must also note again that the reviewer seems to be suggesting that we write up our work differently, to make different points. At this point in the process of publication, we thank the reviewer for these ideas, but must respectfully state that it is our names that are on this work. Our names are guarantors that the results and conclusions are true and correct. Given that responsibility, the decision as to what to study, what results to publish, and what results are to be emphasized, needs to be ours to make.

4. What exactly is the “G” value that they are measuring?

We wish the reviewer might have raised this question earlier...As described, G is essentially a measure of the correlated variation between two distinctly regulated genes. Please refer to **Supplementary Information Section 2: Analytical Framework, subsections S1-S3** for a detailed description of exactly what G is and how we calculate it.

In yeast, in Colman-Lerner *et al.* (Nature, 2005) the co-first author, Andrew Gordon (a high energy physicist working with Colman-Lerner), developed an analytical framework that had the great virtue of suggesting experimental paths to quantify different, hitherto unquantifiable, contributions to cell-cell variation in expression of genes in the *S. cerevisiae* pheromone response system (PRS).

And we adapted that framework for use here in *C. elegans*.

We also calculate signaling noise from the same Type II experiments.

For signaling noise, please also refer to Figure 1 in this reviewer response, and Figure 4g (a global view), Figs S8-10 (some noisy in ring 1 cells) and Supplementary Information section **“S2.5 Two Color Variants Driven by Different Promoters and Pathway Variation”** in the revised manuscript.

We changed text to make these quantities and how they are calculated clearer.

New text reads:

“Type II Experiments Quantify Signaling Noise ($\eta^2(P)$) and Differences in Protein Expression Capacity ($\eta^2(G)$). Type II experiments compare the expression levels from two distinct genes to tell us about cell-to-cell differences in signaling through different pathways ($\eta^2(P)$). In the scatterplot from a Type II experiment shown in Fig. 1, the uncorrelated variation (dispersion across the major axis) is a measure of cell-to-cell differences in signaling through distinct pathways. We subtract the average intrinsic noise ($\eta^2(\gamma)$) for the gene pairs we are measuring, so we know that the remaining dispersion is due to signaling/pathway noise ($\eta^2(P)$) (shown in Fig. 1). For additional details, please also refer to Supplementary Figure S2 and Supplementary Information Section S2.5 Two Color Variants Driven by Different Promoters and Pathway Variation. Type II experiments also quantify how much cell-to-cell variation is attributable to differences in protein expression capacity ($\eta^2(G)$), which we define as the ability to express and maintain proteins. If a cell is able to produce relatively more protein from two or more distinctly regulated genes, compared to other cells in which the same proteins are measured, then we observe that cell as having relatively high protein expression capacity. In the scatter plot of a Type II experiment on the bottom right of Fig. 1, the correlated variation (dispersion along the major axis) is a measure of ($\eta^2(G)$). For additional details, please also refer to Supplementary Information Section S2, Analytical Framework, Sections S1-S3.”

The study was based on a handful of reporters, however, the authors did not provide evidence to support that they are measuring genes that are regulated independently (i.e., that they do not respond to common signalling pathways).

In the 2005 experiments in yeast that are the basis for the present work, we took as a given that our pheromone responsive gene (P_{PRM1} -YFP) was induced differently than main housekeeping gene (P_{ACT1} -CFP) we used as a control controls. Had that not been true, we would not have been able to measure signaling noise specific to the pheromone response system. Similarly, we do assert that the 10 genes used here (*hsp-16.2*, *daf-21*, *eft-3*, *vit-2*, *hsp-17*, *mtl-2*, *His2B::GFP*, *Emerin::mCherry*, *Lamin::BFP*, *Emerin::GFP*) encode very different proteins carrying out very different biological functions, expressed under very different circumstances) are regulated differently. Ample evidence, not limited to the following statements, supports the point. *Hsp-16.2* reporters are not expressed until heat shocked, and *vit-2* reporters only turn on in the intestine after reproductive maturity, as they are supposed to. *Lamin* and *Emerin* only localize to the nuclear periphery as they are supposed to. *His2B* is expressed in the nuclei of all cells as it is supposed to—metaphase plates can be seen in movies of developing embryos on our lab website. They work like they are supposed to as most single copy reporters, knockins and fusion proteins do. Furthermore there are different DNA elements in each promoter required for expression such as the VPE1&2 elements in *vit-2* and the HSE elements in *hsp-16.2*. HSF-1 is the major trans activating factor for chaperones. ELT-2 is the major trans acting factor for *vit-2* and *mtl-2*, which become unrepressed under the right conditions (sexual maturation and exposure to heavy metals (for induction beyond the constitutive baseline of *mtl-2* expression)). We thus are comfortable that genes are activated by different pathways.

In some cases, they observed co-regulation, for example, *vit-2* is downregulated by heat shock (Figure S7B); however, their methodology does not allow them to capture this interaction, as the two genes show a positive correlation of *hsp-16.1*, in Figure 3b, driven by G and not P.

In the revised manuscript, Figure S7B shows lower expression of *vit-2* in heat shocked animals, but, in both groups of animals, those heat shocked and those not, expression of *vit-2* reporter is correlated with that of *hsp-16.2* reporter, defining, operationally, that the differences in expression in these cells are dominated by differences in protein expression capacity. In fact, this figure demonstrates that we do see a signaling change to decrease yolk production and increase *eft-3* production, and that the major axis in both cases is G, despite a shift in signal (P) in the whole population. The figure demonstrates that the population reacted to the P signal from heat shock coherently, but still retains G as the major axis of variation.

We revised the figure legend to more clearly illustrate this point.

Revised text now reads:

Supplementary Figure S7. Evidence for proper signaling. Scatterplot of $P_{eft-3}::GFP$ and $P_{vit-2}::mCherry$ expression at whole animal level with and without heat shock. For heat shock (red dots), day 1 adults were incubated for 1hr at 35°C and imaged 24 hours later together with age-matched non heat shocked animals (blue dots). While the correlated variation is still the dominant source of variation among individuals in each population, the shifts in expression caused by heat shock is clearly visible, and directionally indicated by black arrows annotated with the name of the gene for which they are indicating an expression level shift. **a)** A scatter plot of signaling changes in *vit-2* and *eft-3* reporter genes in response to heat shock from an individual experiment is shown. **b)** Bar graphs quantifying average expression levels from individual experiments are shown; error bars are standard error of the mean. Animals were measured with an epifluorescent microscope at animal resolution, mounted on cover slips as described in methods. At least 30 animals were used per group per experiment."

Generally, can the methodology and protocol detect Pathway related variability?

Yes, it can, as described above.

This is not clear, and therefore, it should be stated throughout the article that they are able to measure G and that G seems pervasive among genes tested, but that they cannot rule out that they are simply not able to measure Pathway-related variability using their protocol.

Again, we can and do measure, quantify, pathway related variability (we explain this in detail in "S2.5 Two Color Variants Driven by Different Promoters and Pathway Variation" in the revised manuscript) We find it to be low. This finding in *C. elegans* is by contrast to our findings about cell-cell variation in gene expression in the *S. cerevisiae* yeast pheromone response system. Being able to quantify these components of variation in *C. elegans* requires precise genome editing to emplace the reporter genes, time consuming microscopic image capture and quantification, and, for this paper, a brave ultimately unsuccessful effort to design and build and use a custom built microfluidic instrument for data collection. Again, in this work, we found that, in cells in the young adult animal, the contribution to variation due to differences in gene expression power was high, and the contribution due to "signaling noise", differences in signaling through different pathways, was low. See for example this scatter plot and images in our reviewer response Figure 1. For additional instances of relatively high signaling noise, please also refer Figs S8-10 (some noisy in ring 1 cells).

At first, we found the low pathway variation to be surprising, but we can speculate that in the young adult, signaling itself might be less important than

in earlier phases of development. Alternatively, we might also attribute it to the fact that *C. elegans* animals have lineage-invariant precisely timed pattern of development, and that such a stereotypical developmental program demands a well-coordinated program of gene expression and that, compared with single cell organisms, variation in signaling for critical genes might be relatively constrained.

It seems to us possible that the reviewer may not be disliking our methods so much as disliking our conclusion, that the main source of variation in gene expression in the tissues studied is variation in G, variation in protein expression, and that variation in signaling is, in these tissues, at these developmental times, relatively unimportant. We believe we need to hew to our conclusion. Other researchers are now welcome to repeat our work, ask the same questions in other tissues at other developmental times, etc.

But at this point we need to get the results we have in front of the community. If publication sparks additional experimentation to disprove the generality of the results, we would view that as a victory.

Unclear how does G relates to pathways that influence protein expression capacity and lifespan? If any reporter that varies with G predicts lifespan, then how do the authors link these claims with the broader literature that links protein synthesis and lifespan. Inhibiting growth related pathways, such as TOR signalling or Insulin signalling prolongs lifespan. If it were true that G positively correlates with extended lifespan, then either G is capturing opposite protein expression capacity than TOR.

We believe that this question will spark wider discussion within the aging community, and we look forward to such discussion. There are many ways to increase lifespan in a worm. We have our ideas on the possible resolution to this possible paradox, but would prefer to save our thinking on these issues for a subsequent papers and reviews. There are over 800 mutations known to affect lifespan, perhaps more than for any other trait.

It would be ideal to determine how TOR does and ILS signalling alters their measurements of G. The full mechanism is a long-term project, but the article should acknowledge the complex implications of their findings.

We discuss how decreased insulin signaling affects protein turnover in the revised text.

The revised text reads:

“It is interesting to notice in this regard, that we have previously found that animals expressing high levels of *hsp-16.2* biomarker have lower levels of *mac-1/rix7p* ribosome export factor (GADAL *et al.* 2001). It is therefore intriguing to speculate that bright animals synthesize less protein and simply maintain mature proteins better. Consistent with this speculation, decreased insulin signaling is associated with decreased global protein turnover (DEPUYDT *et al.* 2016; DHONDT *et al.* 2016; VISSCHER *et al.* 2016). Yet, high biomarker expression resulting from decreased protein turnover is also not supported by our current results, as animals that made more timer protein under *eft-3* promoter control had the same ratio of relatively younger versus relatively older protein (Fig. S12). Moreover, these alternative trade-off models are not supported by our current data (CYPSEY *et al.* 2013). Supplementary Material Section 5: Trade-offs and Supplementary Fig. S18 shows that we can detect trade-offs between lifespan and progeny production in other stressful biological scenarios, such as damaging UV radiation treatment. However, we do not detect such trade-offs after adult heat shock (CYPSEY *et al.* 2013).”

However, we find anything more ambitious to be out of scope. Consider the following set of ideas. It is our hypothesis and working model that chaperones are at least partially responsible for variation in proteome dosage. For instance, it is possible that bright animals do not synthesize more polypeptide chains per unit of time, but rather synthesize them more efficiently with a higher success rate, as measured by percent of properly folded mature proteins. By such means, higher chaperone abundance might result in higher efficiency of protein synthesis and subsequent maintenance. In this scenario it is even possible that bright animals need less ongoing translation at any given time) to fulfill cellular needs for functional proteins. If this was so, the apparent paradox might be reconciled.

However, as mentioned, we would find such a complex speculation out of scope for the discussion section of this paper.

We revised the discussion to reflect this complexity and open the door to future research as shown above in the revised discussion text.

[They could carry out] Very simple experiments with their TIMER-based reporter to provide further evidence.

We concur that such experiments might shed light on whether G was an effect on synthesis or chaperone mediated increased folding and maintenance, as above.

We find such experiments out of scope for this paper. We just spent a year doing experiments suggested by reviewers and feel that we have a sufficient body of work to justify publication.

I wonder if what they are observing has more to do with the overall epigenome status in aged animals, in line with <http://genesdev.cshlp.org/content/28/4/396.full> or with an ageing clock related to the overall health of animals.

We believe that by "what we are observing" the reviewer means the correlation between differences in gene expression and lifespan. As before, we find this correlation fascinating and are grateful for the reviewer's interest. Then, if we understand the reviewer properly, the first question is whether that correlation might have to do with the diminution in nucleosome occupancy and dysregulation of nucleosome positioning in the cited paper (Tyler and coworkers call this "fuzziness" of positioning) in aging animals. The next question is whether this correlation might have to do with [some effects on] an "aging clock" that [by some means] affects or is related to the overall health or lifespan of the animals.

We find both points interesting, and are grateful for the reviewer's interest. We find these out of scope for our current discussion, but (as for the points about TOR and ILS above), may want to fit them into a discussion in a subsequent paper or a review.

5. This article is almost unreadable in its current form.

We thank the reviewer for having plowed through previous versions. In this revision, we've worked very hard to make it more readable.

The figures are not properly labelled and some of the legends are very lacking in details; I had trouble following some of the arguments because of this. I have gone figure by figure, explaining what my problems with them are:

Figure 2.

1. Bar-plots misrepresent data and scatter plots should be provided instead. This article explains why: <https://journals.plos.org/plosbiology/article?id=10.1371/journal.pbio.1002128>

We are aware of the limitations of bar plots and have never shied from using scatter plots to represent data from individual cells (see for example Colman-Lerner et al. 2005). We wish the reviewer had raised this issue in the initial review as it might have weighed in our decision-making for figures in the maintext. But here, for a number of reasons we consider good, mainly need to communicate the important results, we have used barplots in the maintext figures, to provide a concise summary of measurements from many cells of many animals, but have now provided 6 pages of raw scatterplots in supplemental Figures 4&8.

2. Pearson correlations should only be used only if the distributions are normal. The normality of the distributions could be tested in whole worms. At least some of the reporters presented here do not show a normal distribution.

We thank the reviewer for this helpful suggestion. We were somewhat surprised by the assertion that some of the distributions of signal from the different reporters we measured are not normal, as we ourselves are unable to detect deviation from normality by eye- we appreciate this observation. We were not aware of any published results showing that *vit-2* is normally or non-normally distributed.

We also point out that use of Pearson does not require an assumption of normality (Zar, Biostatistical Analysis).

That said, since Spearman does not require normality, and can operate on ranks on non-normal data, and we agree with the reviewer that use of it might be better. So, we, in the revised manuscript, we repeated the analysis using the Spearman correlation, which only slightly changed *r* values.

We found that use of Spearman gave a slightly better correlation for *eft-3* with *vit-2*, but the use of Spearman otherwise did not otherwise change the overall finding, of great correlation of expression of unrelated reporter genes.

3. Do the reporters faithfully represent the endogenous genes? If so, state it.

This work has shown and quantified cell-cell and animal-animal differences in expression of *hsp-16.2*, *daf-21*, *eft-3*, *vit-2*, *hsp-17*, *mtl-2*, *His2B::GFP*, *vit-2::GFP*, *Emerin::mCherry*, *Lamin::BFP*, *Emerin::GFP* reporters. 49 years ago, the use of reporters as proxies for expression driven by different promoters was pioneered in the phage λ and its relatives, with the development and use of $\phi 80trp-lac$ fusions to explore genetics of regulation of tryptophan synthesis in *E. coli* by Jeffery Miller. The popularization of the use of *lacZ* reporters in *C. elegans* as proxies for expression of the native gene is probably most associated with Andrew Fire (1990) and of course the use of promoter GFP fusions and protein-GFP fusions as measures of gene expression and protein levels is most associated with Marty Chalfie's work in the middle 1990s. So a half-century of scientific work in

practice justifies our general approach to using promoter and protein fusions to measure gene expression.

A great deal of evidence and accumulated scientific experience supports the use of the *specific* fusions used here for the same purpose. The three protein fusions used contain the entire coding sequences of the proteins, have all been shown to show correct subcellular localization, and have been used in studies where the intensity of the fluorescent signal has been interpreted as a faithful measure of the amount of the native protein (Haithcock et al 2005; Dickinson et al 2013). The use of single-copy promoter fusion reporters integrated into well behaved chromosomal loci as an approximation of endogenous gene expression is a mainstay of our own work, and we have argued that use of such reporters provides a technical advantage/ freedom from certain artifacts by comparison with earlier published work that used signals from multicopy reporters as lifespan and phenotype predictors. The use of these particular *hsp-16.2*, *daf-21* (Mendenhall et al. 2015) and *vit-2* (Mendenhall et al. 2017) promoter fusions as quantitative indications of gene expression have been mainstays of our own work during the past decade. For the $P_{hsp-16.2}::GFP$ reporters used here, the relationship between GFP signal and mRNA and protein level is well established (Link et al 1999, Rea et al 2005, Cypser et al 2013). The idea that these reporters faithfully mirror expression of the native endogenous genes is supported by correct pattern and timing of expression, e.g. *vit-2* is only expressed in adults and only in intestine, heat shock reporter *hsp-16.2* is only expressed after heat shock, etc.

Figure 3. Eliminate lines 423 and 424, the results cannot be generalised to all genes in the genome.

We are not sure which lines in the revised manuscript the reviewer refers to. We explicitly indicate that not all of the proteome may covary with that of our 10 reporters, but interpret previous data and provide corroborating data that expression of many other genes does as well.

The key passage reads as follows...

“The high correlation of chaperone reporters with phenotypes (Figs. S13-15 and (REA *et al.* 2005; YANG AND TOWER 2009; CASANUEVA *et al.* 2011; MENDENHALL *et al.* 2012)), distinctly regulated transcriptional reporters (Fig. 3), fusion proteins (Fig. 4) and knockins (Fig. 4) suggests that at least some significant portion of the cellular proteome covaries with chaperone biomarkers in some circumstances. This is worth considering in the larger context of any biological scenario. However, we do not yet know what fraction of the cellular proteome for which protein dosage correlates with chaperone abundance.”

Unclear why two types of results are presented. In the upper graphs, variables were extracted from individual promoters and lower graphs, the variability is combined for two promoters. It is probably more insightful to present variability per promoter, unless a good rationale for figures e and f are presented.

The difference is that we do not have an explicit gamma (intrinsic noise) measured for the two of the genes, so, for those, we could not distinguish how much of the uncorrelated variation is due to signaling noise as opposed to intrinsic noise, thus the bins are combined.

In the main text, why is figure 4e discussed before all of figure 4? Probably figure 4e needs to be brought forward to figure 3.

We agree. Deleted. We prematurely mentioned figure 4e. We now wait to discuss figure 4e until later in the manuscript, now with figures 4f and 4g.

Figure

4.

(A) Is it possible to show values separated by promoter? Why is the statistics not shown in the graph? Label axis.

We have added information to these figures. For what is now Figure 4a, the graphed information shows values measured for two different reporters. The left hand bar shows $\eta^2(\gamma) + \eta^2(P)$ for the two reporters, while the right hand bar shows $\eta^2(G)$ for the same two reporters. Those are the numbers graphed; they can't be further decomposed. Although $\eta^2(\gamma)$ is a value that can be measured for two instances of a single promoter, or two different colors of the same protein reporter in the same cell, we did not carry out such measurements.

(B) I find the 3D plots unclear, as the dots look highly dispersed. It would be better to show data similar to 4a, using scatter plots (and not bar plots).

We have attempted explain our use of different kinds of plots (eg. bar graphs in maintext, scatter plots in Supp) above. In this case, our use of a 3D plot allows us to show that we observe substantial covariation of proteins, even we measure three proteins at once, and so we wish to leave it the main text. But, to agree with the reviewer, we have now added separate two-dimensional projections of the same data in separate supplemental images.

We now show the 2d versions of each pair from each of the 3 experiments in a series of nine scatterplots in Figure S11.

C and main text line 185: add Figure 4 c and d. It is very unclear what method was used to generate this data, from page 20 in material and methods, it appears to have been obtained from full animals using a Leica dissecting microscope. Write the method used in the figure and clarify in the text.

We revised the figure legend to refer to the supplementary text.

The new text reads as follows....

c. Scatterplot of $P_{hsp-16.2}::GFP$ and $P_{vit-2}::mCherry$ expression at whole animal level pooled from three experiments. For pooling, values from each experiment were normalized for mean values. **d.** Scatterplot of $P_{eft-3}::mTagBFP2$ and $P_{vit-2}::mCherry$ expression at whole animal level pooled from three experiments. For pooling, values from each experiment were normalized for mean values. Additional details in Supplementary Text "Whole Animal and Whole Intestine Slide-based Epifluorescent Image Cytometry."

[in supplements]

Whole Animal and Whole Intestine Slide-based Epifluorescent Image Cytometry.

For whole animal/intestine analysis animals were anesthetized with 0.2%Tricaine/0.02%Tetramisole and then mounted on 1% agarose gel pads and covered with cover slips. Animals were imaged on Leica Fluorescent Dissecting scope using MicroManager program for image acquisition. For whole intestine analysis of fluorescence in $P_{vit-2} \times P_{hsp-16.2}$ and $P_{vit-2} \times P_{eft-3}$ animals we segmented intestines with thresholding in ImageJ using P_{vit-2} fluorescent channel and measured average signals from both reporters in the segmented area. For whole body analysis of fluorescence in $P_{eft-3}::mTagBFP2$ animals we segmented entire animals with thresholding in ImageJ and measured average signals in each channel in the segmented area."

Text associated with Figure 4.

179-187. Some Sorters allow accurate measurements of co-variance and the elimination of samples that are nonlinear or that preclude proper analysis. I would eliminate this comment from the main text, perhaps add it to materials and methods to explain why it was not used.

The reviewer raises an important point. As veteran users of the instrument, we have long been aware of limitations of the COPAS Biosort for measurements of this type in seeking accurate measurements of the amount of reporter expression in individual cells. One large limitation arises from the fact that the animals are oriented in many different angles along their long axes as they pass through the excitatory beam and its emission is collected. This presentation at different angles blurs signal from single cells or any set of cells that is asymmetrically disposed around the long axis. In a previously reviewed version of this manuscript, the reviewer required that we take more measurements, and in fact insisted that we collect data using a sorter. Our knowledge of the limitations of the COPAS sorter made us design our own microfluidic-based high throughput animal-resolution measurement device. The advantages of this custom built microfluidic device (Crane et al, 2019, Scientific Reports) included that it generates an image of every analyzed animal, which is available for re-examination and re-analysis. Yet, this custom device still lacked enough precision when dealing with two colors, because animals were still positioned too variably along their long axes when in front of the camera. Restated, in both the worm sorter and our microfluidic device, the worms had so much rotational freedom that they did not present the same orientation to the detector. By contrast immobilization of animals on soft agar pads allowed more consistent positioning. We described this new method and our use of it in the manuscript in detail.

Given these outstanding issues with collection of data using the COPAS worm sorter, and particularly given their recurrence even with us using a custom-made device we built in the hopes that it would help us respond to this reviewer's initial review, we believe it's important for us to mention these difficulties in the main text.

Line 185 onwards. The argument used is misleading, as hsp16 promoter should drive expression in many tissues, as much as eft-3.

We have discussed the issue of the vast intestine specificity of hsp-16.2 and the nearly uniform distribution of eft-3 and the effect on measured correlation in gene expression when whole animals are analyzed (in a Biosort or under an epifluorescent microscope) previously in this document, and have included supporting Figure.

A dissecting microscope should not be used to gather data from the intestine only, because, as explained, it is true that signal from other tissues can contribute to it.

This comment prompts us to point out that we carried out this quantification in response to a requirement laid down by this same reviewer, in her/his review of a previous version of this manuscript.

Here's the key request: "I suggest using less careful quantification of nuclei, quantifying at least 200 animals and using whole animal fluorescence using their statistical framework. In fact, although the microscopy method is very accurate it may not be essential to use this level of accuracy because as shown in figure 2f, changes in fluorescence affect worm expression globally."

And we complied with this request. In response to it, we obtained readings from hundreds of additional animals using a newly developed, [somewhat] less labor intensive method that uses a dissecting scope/ which gives [somewhat] less precise measurements. We were comfortable developing this method, having established previously (Mendenhall et al, 2015) that 90% of *hsp-16.2* signal comes from intestine. Since the goal of our current study was to understand the nature of interindividual variation in expression of *hsp-16.2*, which correlates with interesting phenotypes including lifespan, we focused when using the microscope on expression of *hsp-16.2* and other reporters specifically in intestine.

Importantly, the results we obtained largely agree with the results we had previously obtained with our confocal microscopic measurements, despite the fact that other tissues besides intestine contribute to the overall signal (particularly the *eft-3* signal). If, for any reason, the results we obtained with the dissecting scope differed too greatly from those we obtained previously with the confocal, we might be inclined to concede that the reviewer had a point in her/ his comment here.

However, given that, in a response to the reviewer's previous review, and cognizant of the limitations of data gathered by the COPAS sorter alone, we had already gone to the extent of building a new instrument and trying to use it, and then devising a new, higher throughput microscopic method, and given that our results using the new method utterly confirm our previous results using the confocal method, we are comfortable at stating that, for these measurements, we rest our case.

However, if the whole animal is measured then all pixels from the image can be gathered.

This is true, but, on the other hand, whole animal measurements in the worm sorter, or in our custom device, are subject to other sources of error, described above.

The experiment should have been done with two promoters that are expressed ubiquitously or even using whole animal pixel intensity from *eft-3P: GFP*. For example, *eft-3* and *hsp-16.2*.

The promoters for *eft-3* and *hsp-16.2* are not active uniformly or ubiquitously to the same extent; they do not have matching expression patterns, nor is there data anywhere to determine that. It is in fact, incorrect. Given that fact, and the fact that animals in flow are rotated around their long axes and so present different angles to the detectors in the flow instruments that can measure whole animal pixel intensity, and so introduce errors in the measurements as described above, we are not able to use whole animal pixel intensity for the experiments above.

We note that in other work by other researchers, the above artifact (call it the "rotating whole animal pixel intensity artifact") could give rise to data that could be analysed to show differences in gene expression that could be interpreted (erroneously) as animal-animal signaling noise.

We changed the text to make this point explicit.

New text reads:

“When we examined distinctly regulated reporters in hundreds of animals, we found that animal-to-animal variation in G was dominant. Figs. 4c&d, the same trend for differences in G can be seen, even at animal

resolution, provided animals are constrained and a tissue specific marker is used to extract relatively purer signal from a focused region of interest (the intestine tissue). Signals from other tissues differentially contributed to the intestine signal we measured using epifluorescent scope method to examine larger populations of animals. We believe the deviations from the trendline with the *eft-3* and *vit-2* reporters are from out of plane non-intestinal signals contributing to the intestine signal, and remaining rotational differences in mounting. We think the *vit-2* and *hsp-16.2* reporters are more representative of what we see at the cell level because both reporters are mostly expressed in the intestine (90% for *hsp-16.2* reporters in adults(SEEWALD *et al.* 2010; MENDENHALL *et al.* 2015) and exclusively in the intestine for *vit-2*), whereas *eft-3* reporter signal is also relatively stronger in muscle and hypodermal tissues surrounding the intestine, compared to *hsp-16.2* reporter signal. To test the idea that the uncorrelated variation was attributable to out of plane signal from other tissue, we analyzed the cell resolution expression data from the 193 cells we measured from animals expressing the same *vit-2* and *eft-3* reporters and found the correlation to be $r = 0.927$."

What this experiment would tell is interesting information. If it is true that the signalling noise increases if pixels from the entire animal are used, then expression may be dominated by different sources of variance depending on the tissue type.

Yes, this is interesting! And the reviewer seems to agree with the out of plane signals observation we made at this point in this response to review. But it is not the point of our paper. To repeat an earlier general point. As we indicate in the discussion, different sources of noise may indeed dominate in other tissues, such as diploid muscle cells. However, in the cells we examined, at the developmental stage we examined (young adult animals), the predominant source of differences in gene expression, including those particularly interesting differences in gene expression of the lifespan predictive *hsp-16.2* biomarker, is global intestinal differences in G, gene expression capacity. This is what this paper establishes. We believe that this point, and the work that supports it, is ready to publish, and we would like to publish it.

Figure 5. line 490. Be precise with statements. "Interindividual variation of *hsp16.2* reporter" add: after heat shock.

In *C. elegans*, there is no detectable expression of the reporter for the heat shock induced gene *hsp-16.2*, in the absence of heat shock. Since the dependence of heat shock induced genes on heat shock for their induction is widely appreciated, we do not see a need in this case, add the words "after heat shock".

However, to better set this up in the context of the paper, since we've already introduced the idea that expression after heat shock of the *hsp-16.2* reporter is a biomarker, that predicts lifespan, we have changed the abstract wording to read

"Many traits vary among isogenic individuals in homogeneous environments. In microbes, plants and animals, variation in the chaperone system affects a wide variety of traits. In *C. elegans*, the expression level of *hsp-16.2* chaperone biomarkers correlates with or predicts the penetrance of mutations and lifespan after heat shock. However, the reasons that cells express different amounts of the biomarker are unknown. Here, we used an *in vivo* microscopy approach to dissect the sources of cell-to-cell variation in *hsp-16.2* biomarkers, focusing on the intestines of adult animals, which generate the most signal when predicting lifespan. For *hsp-16.2* and a diverse set of distinctly regulated genes, the contribution of intrinsic noise and signaling or pathway noise was relatively small. The major source of cell-to-cell variation in gene expression was due to general differences in gene expression capacity. These *hsp-16.2* biomarkers operationally define states of high or low effective dosages for many proteins. "

In the figure, Mac-1 has not been introduced in the main text.

We thank the reviewer. The revised manuscript reads as follows... "It is interesting to notice in this regard, that we have previously found that animals expressing high levels of *hsp-16.2* biomarker have lower levels of *mac-1/rix7p* ribosome export factor (GADAL *et al.* 2001)."

The section of "working models" would fit better if it were merged with the discussion.

Yes. Again, we have no great justification for so defying perfectly reasonable genre conventions. We moved it to discussion.

Line 228, "In our working model, animals with higher concentration of *hsp16.2*" should read: "animals that express GFP for longer time, driven by the promoter of *hsp-16.2*"

With respect, we want to point out that our working model is the working model of the authors, not the working model of Reviewer 2. And, we measure the *hsp-16.2* reporter to be more concentrated. And, we previously established that more *hsp-16.2* biomarker means more *hsp-16.2*.

But in more detail, we find the Reviewer's [different] interpretation of our results to be unsupported by existing data. Certainly none of the numerous previous publications on phenotype predictors *hsp-16.2* and *hsp-90* / *daf-21* claimed that bright animals expressed reporters for longer (though there is no data against this possibility either). Certainly we have no evidence that bright animals express GFP for longer time. Higher FP signal might result from differences increased production (rate of production and/or duration of production) and/or better folding, and/or better maintenance/ diminished degradation. What we do know is that, as in previously published work, all animals were heat shocked and imaged at the same time, as much as speed of image acquisition allowed, and some animals showed brighter GFP signal than others.

Line 231 therein. The model of how chaperones can alter G is not clear and should be spelled out clearly. If the authors suggesting that the heat shock response may be altering protein dosage, I agree with them. I would also add that it might be the heat shock response more broadly. This however, does not seem consistent with previous statements in the article.

We believe that increased chaperone activity is related to increased protein activity/dosage and have revised the text to make this point.

Figure S5.
Every graph should be a scatter plot, showing individual experiment values and the axis properly labelled.

Insofar as we understand this comment, we have shown scatter plots for all the experiments in the supplementary material. Scatter plots for these graphs we have shown in S4.

Figure S6.
Define ratio set point of expression. Label axis.

We revised the figure legend to define the ratio set point as the cell-type specific slope. We define x and y axes as constants for this figure in the

figure legend. We also expand upon this concept in figures S2&3 in a similar fashion to ensure that the point is clear.

New text in S6 legend reads:

“Scatterplots of expression of two reporters from the different reporter genes, P_{hsp-17} (x axis) and P_{mtl-2} (y axis), grouped by ring (one through four) or combined in one plot (right panel). Cell fate determines ratiometric setpoint for expression of two distinct genes. Specifically, because each gene has a gene-specific stereotypical anterior-posterior expression pattern in the intestine, every gene pair measured has its own particular slope in each cell type (ring), which we refer to as a ratiometric setpoint. When cells are not split by fate, correlation of expression is quite low. Correlation of expression is much higher when cells are grouped by fate. Data shown is from an independent experiment quantifying the aforementioned reporters in intestine cells located in intestine rings one through four.”

Figure S7. Change title of the figure, what does “proper” stand for.

We thank the reviewer for noticing this. We mean correct. The new figure title reads as follows:

“Supplementary Figure S7. Evidence for correct regulation of *eft-3* and *vit-2* expression after heat shock.”

A. What are the colours red and blue standing for, not explained anywhere?

They stand for heat shocked and not heat shocked. We revised the legend.

The new legend reads:

“Supplementary Figure S7. Evidence for correct regulation of *eft-3* and *vit-2* expression after heat shock.

In response to heat shock, animals decrease progeny production and increase expression of chaperones and *eft-3*. Here, we heat shocked animals to determine if the reporter genes would respond to the heat shock as expected. The scatterplot shows $P_{eft-3}::GFP$ and $P_{vit-2}::mCherry$ expression at whole animal level with (red dots) and without heat shock (blue dots). For heat shock (red dots), day 1 adults were incubated for 1hr at 35°C and imaged 24 hours later together with age-matched non heat shocked animals (blue dots). While the correlated variation is still the dominant source of variation among individuals in each population, the shifts in expression caused by heat shock are clearly visible. The directionality of the population expression shifts caused by the heat shock are indicated by black arrows annotated with the name of the gene for which they are indicating an expression level shift. **a)** A scatter plot of signaling changes in *vit-2* and *eft-3* reporter genes in response to heat shock from an individual experiment is shown. **b)** Bar graphs quantifying average expression levels from individual experiments are shown; error bars are standard error of the mean. Animals were measured with an epifluorescent microscope at animal resolution, mounted on cover slips as described in methods. At least 30 animals were used per group per experiment. A one tail.....”

B. What was the stats used, what are the two panels showing? If they represent two biological replicates, then a third one is necessary and the data represented as a scatter plot with statistic only performed on data with at least three biological replicates.

We added information about the statistics to the supplementary figure legend. The two different panels show results from two separate sets of experiments. We changed the Supplementary Figure legend to state this fact.

If they represent two biological replicates, then a third one is necessary and the data represented as a scatter plot with statistic only performed on data with at least three biological replicates.

We have included details showing all three replicates for figure S7, now showing a third bar plot.

Figure S12 and main text from line 194.

Is it editorially correct to base main text on figures that are not presented in the main figure section? Probably Figure S12 has to be brought forward to the main figure section.

Again, this was an additional experiment done after the first review. We do not want to distract from the main focus of the paper, which was determining the mechanisms of cell to cell variation in a metazoan tissue for the first time. We do understand that this seems to be editorially permissible.

Line 204, brighter for what?

We thank the reviewer. We changed "animals that are brighter" to "animals that express more reporter"

Line 206. There is a mistake as Figure S8 shows a completely different set of results.

We are grateful to the reviewer for catching this. Corrected to Fig S12c with thanks.

I cannot fully gather the significance of Figure S12 because the figure is so poorly labelled and explained. Materials and methods has no information on the TIMER protein used.

We imaged these animals with whole animal fluorescence as in Figs. 4c&d.

We revised the text to say:

"We quantified the age of protein *in vivo* using a fluorescent timer protein that matures from green to red in about 48 hours (TERSKIKH *et al.* 2000). We used whole animal image cytometry to quantify the fractions of relatively young and old protein in individual animals, detailed in Supplementary Materials and Methods."

However, from what I can take out of the figure it appears as if animals that produce more o[ld] new EFT-3 protein, also keep it for longer.

Yes.

This suggests that there are animals that have an increased capacity to produce and keep proteins. This is a very interesting result.

I would eliminate panels b and c (unless they say something that I cannot fathom [fathom?] the importance of)

We moved panel C showing that the TIMER protein works to become Figure S12a. We kept the replicate experimental data on display.

and add a few extra experiments:

Two very good and easy experiments that can be done are:

1. Are animals with high EFT: timer-red/green good predictors of thermotolerance and/or hsp16p: gfp induction, lifespan, etc.
2. EFT:timer seems to suggest a broad variability across worms. Is this correlated with the expression of other reporters such as VIT-2?
3. Can VIT-2 expression predict hsp16 induction and/or EFT-timer?

We performed this experiment (Figure S12) for the revised manuscript with the goal to better understand the observed phenomenon mechanistically. Importantly, even this experiment went beyond original scope of the manuscript, in which we aimed to report that proteome dosage (as quantified by the G component of extrinsic noise) and not stochastic or signaling noise governed interindividual variation in hsp-16.2 biomarker expression. Nevertheless, we did this work because a) because it offered experimental means to address what we deemed an interesting and important question (in higher G animals, is the greater protein dosage due to increased protein production or greater maintenance of older protein?) and b) the reviewer was also curious about this/ requested it.

We are grateful that the reviewer has engaged with this finding, and has suggested additional experiments. With respect, at this point in the review process, we do not wish to carry these out, but these and other questions will inform our future work.

Figure S12 title has a misspelling it should say proteins and not protein.

We are grateful to the reviewer for catching this imprecision. Corrected.

The new text reads:

Supplementary Figure S12. Animals that respond better to heat shock produce and maintain timer protein better.

[S12] A. What do colours correspond to?

We thank the reviewer. The blue and red dots correspond to animals that were or were not heat shocked. We have changed S12 Figure legend to explain that fact.

The new text reads:

Supplementary Figure S12. Animals that respond better to heat shock produce and maintain timer protein better. To test the function of the timer protein in **a**, we quantified whole animal fluorescent signals in the green (younger protein) and red (older protein) channels in each individual animal in populations of L1 larvae or two-day old adults and then plotted the results in a box plot. To measure the fractions of younger and older protein in animals in **b** and **c**, we heat shocked adult animals (red dots) on day one of adulthood and measured them on day two of adulthood, or, for non heat shocked animals (blue dots), we simply measured them on day two of adulthood, consistent with all other experimental measurements of adults in this report. We then quantified whole animal fluorescent signals in the green (younger protein) and red (older protein) channels in each individual animal in populations of two-day old adult animals that had or had not been heat shocked in four independent experiments. **a**. Timer protein works as reported. To ensure that timer protein reports on average protein age, we examined ratio of old to new timer fraction in approximately ten L1 larvae and in approximately ten, two-day old adults. As expected, slowdown of protein turnover with age increased the ratio of old to new timer protein. The boundary of the box closest to zero indicates the 25th percentile, a line within the box marks the median, a dash within the box marks the average, and the boundary of the box farthest from zero indicates

the 75th percentile. Whiskers above and below the box indicate the 90th and 10th percentiles. **b.** Scatterplot of whole animal average values fluorescent timer protein expressed from *eft-3* promoter ($P_{eft-3::timer}$). Linear dependence of young and old protein fractions indicates that bright animals are better at both protein production and maintenance. **c.** Whole animal average abundance of young and old fraction of timer protein with and without heat shock in arbitrary units (tiff counts). The three plots correspond to three independent biological replicates quantifying expression from at least 30 whole animals per group per experiment. Error bars are standard error of the mean.”

My understanding of this figure is that if a worm produces more young protein it has also produced and/or maintained more protein than other worms for the last 2 days.

That is our understanding of our results as well.

It is not possible with this sort of experiment to distinguish the source of the old protein.

We concur with the reviewer. This experiment does not allow us to distinguish the source of the old protein. We do not assert that it does.

B. This figure is very difficult to understand. How was the experiment done, have animals been sorted by green/red and then heat shocked and tested for green/red accumulation? What are the asterisks representing? The bars do not appear different. Again, scatter plots should interchange bar charts. If the asterisks represent statistical significance, present the raw data as a table justifying the choice of statistical test.

We respectfully disagree with the reviewer regarding data presentation. Figure S12a demonstrates a representative scatterplot from one experiment. Bargraphs in Figure S12b represent statistical analysis of three other independent experiments. Combination of scatterplots and bargraphs provides maximum information for readers. We therefore did not make any changes to the figure.

We have updated S12 figure legend and methods to explain.

The new text reads:

Supplementary Figure S12. Animals that respond better to heat shock produce and maintain timer protein better. To test the function of the timer protein in **a**, we quantified whole animal fluorescent signals in the green (younger protein) and red (older protein) channels in each individual animal in populations of L1 larvae or two-day old adults and then plotted the results in a box plot. To measure the fractions of younger and older protein in animals in **b** and **c**, we heat shocked adult animals (red dots) on day one of adulthood and measured them on day two of adulthood, or, for non heat shocked animals (blue dots), we simply measured them on day two of adulthood, consistent with all other experimental measurements of adults in this report. We then quantified whole animal fluorescent signals in the green (younger protein) and red (older protein) channels in each individual animal in populations of two-day old adult animals that had or had not been heat shocked in four independent experiments. **a.** Timer protein works as reported. To ensure that timer protein reports on average protein age, we examined ratio of old to new timer fraction in approximately ten L1 larvae and in approximately ten, two-day old adults. As expected, slowdown of protein turnover with age increased the ratio of old to new timer protein. The boundary of the box closest to zero indicates the 25th percentile, a line within the box marks the median, a dash within the box marks the average, and the boundary of the box farthest from zero indicates the 75th percentile. Whiskers above and below the box indicate the 90th and 10th percentiles. **b.** Scatterplot of whole animal average values fluorescent timer protein expressed from *eft-3* promoter ($P_{eft-3::timer}$). Linear dependence of young and old protein fractions indicates that bright animals are better at both protein production and maintenance. **c.** Whole animal average abundance of young and old fraction of timer protein with and without heat shock in arbitrary units (tiff counts). The three plots correspond to three independent biological replicates

quantifying expression from at least 30 whole animals per group per experiment. Error bars are standard error of the mean.”

C. This experiment does not show that the timer works as expected.

By our legend, we meant that the TIMER protein was working *in our hands*. By our legend, we meant to convey that the ratio of green to red TIMER protein was low in larval animals, where there had been less time to accumulate. Perhaps we should have used the words "works as advertised".

We have changed the title as follows...

Supplementary Figure S12. Animals that respond better to heat shock produce and maintain timer protein better..

We do note that the fact that old worms on average have older TIMER than young larvae, is consistent with proteomic studies that showed that protein turnover slows down with age.

We have changed the legend to reflect this.

The revised legend reads:

Supplementary Figure S12. Animals that respond better to heat shock produce and maintain timer protein better. To test the function of the timer protein in **a**, we quantified whole animal fluorescent signals in the green (younger protein) and red (older protein) channels in each individual animal in populations of L1 larvae or two-day old adults and then plotted the results in a box plot. To measure the fractions of younger and older protein in animals in **b** and **c**, we heat shocked adult animals (red dots) on day one of adulthood and measured them on day two of adulthood, or, for non heat shocked animals (blue dots), we simply measured them on day two of adulthood, consistent with all other experimental measurements of adults in this report. We then quantified whole animal fluorescent signals in the green (younger protein) and red (older protein) channels in each individual animal in populations of two-day old adult animals that had or had not been heat shocked in four independent experiments. **a.** Timer protein works as reported. To ensure that timer protein reports on average protein age, we examined ratio of old to new timer fraction in approximately ten L1 larvae and in approximately ten, two-day old adults. As expected, slowdown of protein turnover with age increased the ratio of old to new timer protein. The boundary of the box closest to zero indicates the 25th percentile, a line within the box marks the median, a dash within the box marks the average, and the boundary of the box farthest from zero indicates the 75th percentile. Whiskers above and below the box indicate the 90th and 10th percentiles. **b.** Scatterplot of whole animal average values fluorescent timer protein expressed from *eft-3* promoter ($P_{eft-3::timer}$). Linear dependence of young and old protein fractions indicates that bright animals are better at both protein production and maintenance. **c.** Whole animal average abundance of young and old fraction of timer protein with and without heat shock in arbitrary units (tiff counts). The three plots correspond to three independent biological replicates quantifying expression from at least 30 whole animals per group per experiment. Error bars are standard error of the mean.”

The authors should include the reference where the protein turnover has been shown to decrease in older animals. One experiment to test the validity of the timer, would be to cross it with a temperature sensitive *ama-1* allele, to determine if KO of transcription results in accumulation of older protein.

See above. We already discussed decreased insulin signaling and decreased protein turnover. We believe that the fact that the reviewer was moved to suggest experiments might have been based on an ambiguity in the meaning of the title we gave the figure, for which we apologize. We thank the reviewer for suggesting these experiments, which would help explore whether the reported decrease in protein

degradation in older animals resulted in a change in the ratio of new to old TIMER in those animals.

Additional Corroborating Data.

I am not familiar with the journal's policies on this sort of supplementary sections. Unless part of the results is presented in the main text, the section should be removed from the article.

We refer to this material in the main text, and so normally would need it here.

Comments on supplementary material page 32.

On the results related to Figures S13 and S14.

These are interesting observations and they support their model, therefore should be brought (at least parts of it) to the main text.

The results presented in these supplementary figures were criticized in previous reviews. As they are not the main point of our paper, we prefer they remain in the supplements.

The fact that RAS GOF phenotype anti-correlate with chaperones later point than when the phenotype arises can be interpreted in two ways:

First chaperones can aid in the folding of a hypermorphic allele. Second, those differences in protein dosage may explain the dose dependency of RAS (GOF).

Yes, we agree, and we stated that we think that more chaperone abundance equates to more protein activity, be it hypomorph or hypermorph.

The revised text reads:

“Prior work showed that higher expression of chaperones is correlated with the effective activities of loss of function mutations (CASANUEVA *et al.* 2011), because these hypomorphic alleles were less penetrant in animals that expressed more *hsp-16.2* chaperone biomarker reporter gene. Expressing more *hsp-16.2* chaperone biomarker indicates these high biomarker expressing animals have more of some types of actual chaperones (REA *et al.* 2005; CYPSEY *et al.* 2013). This suggests that gain of function mutations would also sometimes have higher activities with higher abundance of some chaperones. This work thus extends previous work with loss of function mutations to include alleles that conferred gains of function.”

Based on results from figure S12 eft-3: TIMER there seems to be animals with larger capacity to make and keep proteins than others. Their RAS-GOF results would argue that G is quite stable in time and can retroactively predict an earlier phenotype.

Yes, we agree. But, given that our L1 results on the persistence of G showed a lack of persistence into adulthood, and given that the bursts of protein expression we saw in larvae do not yet fit into our analytical framework, we are more cautious about these results and would like to leave it in the supplements.

I am less keen to believe that this has to do with chaperoning activity. The heat shock driven fluorophores are measured 1 day after the heat shock, and it is unclear if chaperone levels remain high in those animals,

24 hours after heat shock is when the *hsp-16.2* chaperone levels peak at the protein level, determined via polyclonal antibody staining by Link et al in 1999.

Supplementary Section 4.

Supplementary figure S17. It is unclear what is the point that is been made.

We have reworded this to be more precise.

The revised text now reads:

“In Supplementary Fig. S17 we show that intestine cells in larvae show differences in G, but also have seemingly autonomous bursts of protein expression, which we do not see in two-day old adults, and will require further study to understand how these bursts fit in the context of developmental progression and other kinds of noise in gene expression.”

Supplementary S18. E and f, these are two independent experiments, though in the legend it says that three independent experiments had been done. Where is the third one? If there are not 3x then no statistic is possible.

There are three independent experiments for lifespan and five independent experiments measuring the fecundity of individuals of different genotypes.

We revised the figure legend to make this explicit.

New text reads:

“We measured 50 animals in three independent experiments and we measured the self-fertile fecundity of five individual animals from each genotype in five independent experiments.”

Comments to response to reviewer 2. I have indented these comments in the rebuttal letter as well [ED:attached].

Page 10, point1. I appreciate that the authors have made the steady state point more clear in the methods and in the manuscript, however, as stated above, it is not how the message is read throughout the article, beginning with the title.

This and some of the comments below are by Reviewer 2 on remarks made by another reviewer of the previous version of the manuscript and our responses to that other reviewer. Our title uses the term "biomarker expression" and we measure the same biomarker and its expression in same way we and others have been doing it for more than 10 years. We are not sure we have the context to know exactly what this comment means.

However, working from that best guess. As we understand it: the bulk of this paper and much of our work over the previous two decades involves quantification of GFP or other fluorescent protein signal from different transcriptional reporters and protein fusions. Our use, and use by many 1000s of other researchers, for the amount of fluorescent signals of this type as proxies for gene expression is well established. Amount of signal by no means relates to instantaneous rate of gene expression, but rather reflects maturation time of the fluorophore (Gordon et al. 2007), and fluorophore degradation (Gordon et al., Nature Methods 2007, and of

course many other others). In 2005, we demonstrated (Colman-Lerner et al., Nature, 2005) the use of these signals to quantify the different contributions of signaling noise and gene expression capacity in *Saccharomyces cerevisiae*, as we have done here.

Page 10, point 1 a. I appreciate that the authors have added substantial amount of work to determine that their measurements are technically sound. I have no further comments on this aspect.

We worked hard on this, and are grateful that the reviewer (Reviewer 1?) is now satisfied.

However, I do have a problem with point 1a, 3, on page 11. "Interpretations of previously published work", and also in response to Reviewer 1, and further in the introduction.

We will deal with these objections point by point, below.

In response to reviewer 1 Page 1. Paragraph 3 and throughout the introduction. It is unclear how variability in protein expression capacity would correlate with phenotypes driven by loss of function mutations. The article cited 2012 Lehner article, which showed that *hsp16* is a good predictor of l-o-f phenotypes, used a very different protocol. In that article, the authors showed that heat shocked animals during L3 larval stage could predict adult LOF phenotypes. In this article, the authors show here G is not stable over time, that G obtained from early (L1) larval stages does not predict G in adults. If that is the case, then it is not clear if generalisations are appropriate.

First, as we understand it, the reviewer making the above comments (Reviewer 2) is asserting that Casaneuva et al. 2012 showed that the response of animals heat shocked in L3 could predict diminution of function phenotypes in adult animals. We differ from this assertion about the published work on two points.

First, we believe the heat shock was during L1. We quote from the article "Animals received a 2-hour 35°C heat shock as L1 larvae and were sorted 1 day later into "high" (right worm) and "low" (left worm) populations, according to the induction of an *hsp-16.2* chaperone promoter reporter." As we now re-read the paper in more detail, we understand that researchers might have heat shocked the animals at the L1/L2 transition stage (which occurs 12 hours post hatching) and sorted them at the L3/L4 stage. After sorting, the researchers scored the following phenotypes from named alleles in the following genes alleles: multivulval phenotypes in *lin-31* (*n1053*), male

[REDACTED]

Figure 4. L1 heat shock used by Lehner and coworkers to determine the correlation of chaperone reporter levels with the penetrance of discrete traits. The researchers in Casaneuva et al tested consequences of heat shocks at two times during development: during L1, for penetrance of mutations, and during L3, to examine tradeoffs between heat shock response and adult fertility. Image is from Figure 2, Casaneuva et al. (2012).

gonad migration defect in *lin-29* (*ga94*), male tail ray defect in *mab-19* (*bx83*), dumpy in *vab-9* (*ju6*).

Our second point is that all these phenotypes are developmental phenotypes, eg, vulva formation occurs during L3-L4 stages and not during adulthood. Hence, the phenotypes of interest manifested either with or just after the time of sorting, and, once manifest persisted into adulthood (e.g. extra vulval formations did not disappear). We thus view the statement that "In that article, the authors showed that heat shocked animals during L3 larval stage could predict adult LOF phenotypes." as incorrect. As before, the heat shock was earlier, in L1. And as important, although the phenotypes were scored in adulthood, they were likely already manifest at or just after the time of sorting.

We note that such an interpretation of correlation of L1-induced *hsp-16.2* expression with greater expressivity and penetrance of mutations in L3/L4 larvae is also consistent with our results on neomycin resistance. We imagine that higher proteome dosage at L1 stage should allow some larvae to go through their initial developmental stages in the presence of neomycin better than larvae with low protein dosage. Our work in this paper seems to indicate that this difference in initial protein dosage may not persist through all developmental stages, but higher levels of drug resistance may have been sufficient to get the animals through L1. In our revised manuscript, we cited the recent paper by Andy Fire's lab and coworkers showing that animals with no ribosome genes survive to L1. We interpret that to mean that sometime before or during L1, there is a switch from translation using maternal ribosomes to translation using zygotically encoded ribosomes.

We stand by our statement about results from the Casaneuva et al. 2012 paper. In response to the comments of Reviewer 2 on the review of Reviewer 1, we have made no

Figure 5. Why we believe alleles used in Casaneuva et al 2012 are null. These are large deletions (highlighted in yellow) that take out most of the exons and leave only one remaining exon for *efn-2* (top panel) and *lin-31* (bottom panel). Miller et al 2000 found that anything that deleted the DNA binding domain contained in the first three exons of *lin-31* would be a null. Thus, the deletions of most exons in these genes lead us to believe that they are null; the remaining sequence in both mutants seems unlikely to transcribe anything and almost certainly untranslatable. The point about nulls is not necessary for the paper and will be removed from the introduction as it distracts from the main points and is not discussed further until the discussion section, where we note that we think that these *hsf-1* refractive alleles are probably null. We note that the authors did not definitively list them as cold sensitive but also stated non-ts. It is worth noting and considering that genetic requirements for growth and lifespan are also different at different temperatures for poikilotherms.

changes in revised text.

In response to reviewer 1 Page 4, Paragraph 3 and throughout the introduction. The authors are again, inaccurate in their reading of the cited Lehner 2012 article [This article] showed that hsf-1 and hormesis rescued the effect of temperature sensitive alleles, but both partial and complete loss of function alleles, but not a cold-sensitive mutation. The same inaccuracy is repeated in lines 41 and 42 of the first page of the introduction. Again, that article [Casaneuva et al., 2012] cannot be used to provide evidence to support their model because the cited article used variable expression of hsp-16 in larval stages (as pointed above).

In our revised manuscript, we do not mention this in the introduction as reviewer 1 was also confused about nulls. We now mention this point in the discussion.

The sentence in question reads: "While we do not know exactly what kind of variation is happening in larval cells, it is worth reconsidering some of the results from Casanueva et al. on penetrance. In 2012, Casanueva et al. showed that overexpression of HSF-1, which activates expression of chaperones, decreased penetrance of hypomorphic mutations, but not of null mutations. The mutations we believe to be null were listed as "non-TS/Cold-sensitive"¹⁰; we believe they are almost certainly null because these alleles (*lin-31(gk569)* & *efn-2(ev568)*) delete most of the exons, leaving only the final exon for each gene (www.wormbase.org)."

This is an important point, we think this is an important lesson from Casaneuva et al. 2012, and we stand by this statement.

The reviewer asserts that HSF-1 overexpression rescues both partial and complete loss of function alleles.

By contrast, our understanding of the alleles not rescued by HSF-1 overexpression, and those not rescued, is the following. In the published work, two alleles were refractory to HSF-1 overexpression: *efn-2(ev568)* and *lin-31(gk569)*. *efn-2(ev568)* lacks the first four exons of *efn-2*, and *lin-31(gk569)* lacks the first three exons (see figure showing these alleles in wormbase) of *lin-31*. We also refer to work by Miller et al from 2000, showing that then existing alleles of *lin-31* were all null or virtually null, and that (significantly) any deletion of more than half of the coding sequence-- which the *gk569* allele is-- all three of the first exons containing the DNA binding domain are gone-- is null. For this reason, we strongly believe these alleles are nulls. By contrast, the alleles that were successfully rescued by HSF-1 overexpression, were either substitutions or deletion within introns, making it highly likely that those are reduction of function alleles.

We thank Reviewer 2 for giving us the impetus to make this argument more explicit, and, we hope, more powerful.

In response to reviewer 1 Page 5. Paragraph 1 and throughout the article. If any reporter that varies with G predicts lifespan, then how do the authors link these claims with the broader literature that links protein synthesis and lifespan. Inhibiting growth related pathways, such as TOR signalling or Insulin signalling, prolongs lifespan. If it were true that G positively correlates with extended lifespan, then either G is capturing opposite protein expression capacity than TOR. If G was totally independent from TOR signalling, mutants in this pathway should not alter their main observations. I wonder if what they are observing has more to do with the overall epigenome status in aged animals, in line with <http://genesdev.cshlp.org/content/28/4/396.full> or with an ageing clock related to the overall health of animals. It would be ideal to determine how does TOR and ILS signalling

alters their measurements of G. The full mechanism is a long-term project, but the article should acknowledge the complex implications of their findings.

We have addressed these questions in our response to remarks by Reviewer 2 above.

Page 12, Response to concepts put forward in the reference PMID 22365828. I disagree with the authors in that I do not believe that their manuscript clearly states what they are measuring and how that is different from the above reference. First, I think the reference should be included to this article.

We wonder if this comment may represent a misunderstanding. PMID 22365828 is a nice paper from Jonathon Weisman's and Hana El-Samad's labs showing work done by [one] coworker, a shared researcher named Stewart-Orenstein. In it, these workers monitor by flow cytometry expression of 400+ FP fused proteins in diploid strains. In one set of strains, one copy of the gene is fused to GFP, and, in the cognate set of strains, both copies are fused. Comparison of GFP signal from large numbers of a given strain in each of the two sets allowed the researchers to use a method pioneered by Hasty and coworkers (Wolfson et al., 2006) to calculate a measure of intrinsic noise, equivalent to that measure we call $\eta^2(\gamma)$, and a different measure of total extrinsic noise. By modeling based on the experiments, Stewart-Orenstein et al. make plausible two points, show that genes that show high extrinsic noise are enriched in certain well known transcription factor binding sites, and that the amount of cell-cell variation ("noise") expression covaries under different conditions, and that different sets of co-regulated genes define noise regulons.

Again, theirs is fun work. But we do not find it applicable here. The reason is that the method of Hasty and coworkers is indirect, and (to us) highly non-intuitive, and it does not get us G. Moreover, and by contrast with our framework, the method of Wolfson et al. relies only on comparison of expression from identical genes present in different copy number. Expression of the gene of interest is not compared with that of any other control gene and so the method of Wolfson et al cannot detect cell-cell differences in signal reaching the promoter $\eta^2(P)$.

For that reason, we find the work from Weisman and El-Samad, although interesting, not relevant to our own work, in which we set out to quantify different contributions to cell-cell variation in *hsp-16.2* biomarker expression.

We do take the point that Reviewer 2 is somewhat perplexed by our framework.

In the revised manuscript we now include graphical illustrations of the kinds of noise we can measure and what cell autonomous and cell non autonomous variation in each type of noise would look like in Figs. S1-S3. We also stress the importance of reading the details of our analytical framework when comparing it to other contemporary studies of cell-to-cell variation in gene expression.

Second, it should be explained that their experimental set up allows them to directly measure G, but the fact that they do not measure pathway-related variability may be related to the type of reporters used and the timings used to approach the heat shock response.

As stated, this is not correct. Indeed, as stated, we do observe and quantify uncorrelated variation, indicating either intrinsic noise or signaling noise depending on the type of experiment (Type I or Type II) and magnitude of deviation. Our framework permits measurement and quantification of this variability.

Figure 1 in this reviewer response, and Figure 4g (a global view), Figs S8-10 (some noisy in ring 1 cells) and Supplementary Information section **“S2.5 Two Color Variants Driven by Different Promoters and Pathway Variation”** in the revised manuscript, all contain information relevant to this point.

Again, as published many years ago (2005), in the *S. cerevisiae* PRS, $\eta^2(P)$ is a significant source of cell-cell difference. Indeed, the abstract of that paper states variation is dominated by differences in the capacity of individual cells to transmit signals through the pathway (‘pathway capacity’) and to express proteins from genes (‘expression capacity’) and the heart of the paper is a table (see inserted) which quantifies $\eta^2(P)$ in various strains. We were and are surprised to have found that this is not a major source of variation in expression of the *hsp-16.2* biomarker and other genes we studied in the young adult, and wish to share our surprise with the world via the medium of a published paper.

I am not convinced that “ a considerable fraction of the proteome co-varies with natural variation of chaperones.

Reviewer is not convinced.

... apparently because...

They use GFP expression 24 hours after heat shock and the expression of Hsp-90, which as shown by Klosin and Lehner, 2017 is a reporter that aligns with piRNAs and not the heat shock response.

Lehner and colleagues showed that, as expected, animals that make more hsp-90 reporter, also make more hsp-90 protein, in Burga et al 2012.

We have discussed the appropriateness of our methodology elsewhere in this very lengthy response. To avoid even the appearance of overreach, we have deleted this assertion from the revised manuscript. We now use the following terms:

“some significant portion of”, or “the small fraction of the genome we examined”

In reality, it does not really matter that they may not be measuring the heat shock response per se, because they care about it is that it is a biomarker that predicts lifespan. I have no problem with this observation, but I do struggle with generalisations with respect to the heat shock response and with signalling noise.

We have discussed at sufficient length the appropriateness and very long history of use of GFP reporters as a measure of the heat shock response. We believe we have more than sufficiently demonstrated our ability to detect and quantify cell-cell differences in signaling $\eta^2(P)$, and are quite comfortable making sure our generalizations from our findings are appropriately constrained. What we do not wish to further countenance any dispute that we can measure $\eta^2(P)$, that it is low for the genes and cells and developmental stages we measured, and the other component, $\eta^2(G)$, is high and that we should be able to publish that result.

For signaling noise, please also refer to Figure 1 in this reviewer response, and Figure 4g (a global view), Figs S8-10 (some noisy in ring 1 cells) and Supplementary Information section **“S2.5 Two Color Variants Driven by Different Promoters and Pathway Variation”** in the revised manuscript.

Page 14. I urge the authors to update their knowledge in the proteostasis field. I was referring to the proteostasis collapse described by Labaddia and Morimoto, 2014. A clear summary of the field of proteostasis during ageing can be seen in this figure: <https://www.ncbi.nlm.nih.gov/pmc/articles/PMC3914504/figure/fig-001/>

I do believe that again, the collapse of the HSR may be relevant to the timing of their measurements. And this should be acknowledged.

Although we are by our temperaments never satisfied with our relative ignorance of any field, for this topic, we consider our knowledge of proteostasis and the collapse posited by Labaddia and Morimoto in their review article to be up to the demands we placed on it. Our response is that don't believe it's relevant for the proteins we measure when we measure them. We studied variation in two-day old adults. This is a peak of reproductive activity, which involves large amount of protein synthesis. Although Labaddia and Morimoto propose that their posited collapse may begin around this time point, the only effects that can be seen on cellular proteins at this time point can only be seen with mutated unstable proteins; if it were otherwise, animals would not be at the peak of their reproduction and ability to synthesize and accumulate new proteins.

Although it is possible that "collapse" or other phenomena related to proteostasis might challenge or at least affect or slightly modify quantification based on decades of work using *hsp-16.2::GFP* and other reporter genes to quantify the heat shock response, we do not believe this paper is now the place to launch any such discussion, and it is certainly not the time or the place to litigate this well-established experimental methodology.

Page 14, point 2. The authors point out that they do see interactions among different reporters. However, G still dominates, but I cannot make sense of their data. For example, why if *vit-2* is downregulated by heat shock (Figure S7B), still presents a positive correlation of *hsp-16.1*, in Figure 3b? That interaction must be related to pathway related variability and yet in figure 3b, the dispersion is non-existent and the correlation is highly positive. It is difficult as a reader to make sense of this.

The reviewer notes that correlated differences, differences in G dominates cell-cell differences in expression. This observation holds for all genes tested in this paper, and is its point. Reviewer asks why, if *vit-2* is diminished by heat shock, expression of *hsp-16.1* [*hsp-16.2*] and *vit-2* are still correlated? The answer to that question is that even though expression of one gene or the other might change under some specific condition, cells that produce large amounts of one reporter product tend to produce large amounts of other gene products. This is the essence of the positive correlation. We believe that this point made by the reviewer may illustrate a significant misunderstanding of our methods and conclusions. We hope that the revised text makes this clearer.

For signaling noise, please also refer to Figure 1 in this reviewer response, and Figure 4g (a global view), Figs S8-10 (some noisy in ring 1 cells) and Supplementary Information section **''S2.5 Two Color Variants Driven by Different Promoters and Pathway Variation''** in the revised manuscript.

The revised text reads:

Supplementary Figure S7. Evidence for correct regulation of *eft-3* and *vit-2* expression after heat shock. In response to heat shock, animals decrease progeny production and increase expression of chaperones and *eft-3*. Here, we heat shocked animals to determine if the reporter genes would respond to the heat shock as expected. The scatterplot shows $P_{eft-3::GFP}$ and $P_{vit-2::mCherry}$ expression at whole animal level with (red dots) and without heat shock (blue dots). For heat shock (red dots), day 1 adults were incubated for 1hr at 35°C and imaged 24 hours later together with age-matched non heat shocked animals (blue dots). While the correlated variation is still the dominant source of variation among individuals in each population, the shifts in expression caused by heat shock are clearly visible. The directionality of the population expression shifts caused by the heat shock are indicated by black arrows annotated with the name of the gene for which they are indicating an expression level shift. **a)** A scatter plot of signaling changes in *vit-2* and *eft-3* reporter genes in response to heat shock from an individual experiment is shown. **b)** Bar graphs quantifying average expression levels from individual experiments are shown; error bars are standard error of the mean. Animals were measured with an epifluorescent microscope at animal resolution, mounted on cover slips as described in methods. At least 30 animals were used per group per experiment.”

Page 15, 5th paragraph. If this work is not about the investigation of pathways, then it should be stated, that you were seeking to measure common sources of gene expression variance and did not design a study to capture Pathway variance.

In fact, our study did detect and quantify pathway variance.

In support of this, we ask the reader to refer to Figure 1 in this reviewer response, and Figure 4g (a global view), Figs S8-10 (some noisy in ring 1 cells) and Supplementary Information section **“S2.5 Two Color Variants Driven by Different Promoters and Pathway Variation”** in the revised manuscript.

As we have mentioned elsewhere in this response, we carried out a study to capture and quantify 3 major contributors to cell-cell variation in gene expression: "intrinsic noise", $\eta^2(\gamma)$, cell-cell differences in global gene expression, $\eta^2(G)$, and cell-cell differences in signaling to specific promoters (pathway), or $\eta^2(P)$. Our results quantified $\eta^2(P)$ but revealed, to our surprise and by contrast with our previous observations in *S. cerevisiae*, that, in young adult animals in the cells studied, pathway variation is low. To repeat: just as the reviewer was surprised, we were surprised by how much gene expression variation is determined by G and not stochastic or signaling noise. However, after measuring many hundreds of animals, at single cell resolution, we are confident in these results. Our data show that variation in G and not pathway variation is the major axis of protein expression variation in our system in the cells we measured at the time we measured them.

For signaling noise, again, please also refer to Figure 1 in this reviewer response, and Figure 4g (a global view), Figs S8-10 (some noisy in ring 1 cells) and Supplementary Information section **“S2.5 Two Color Variants Driven by Different Promoters and Pathway Variation”** in the revised manuscript.

Page 15, last paragraph. I appreciate that the authors obtained data from many animals and that this was challenging. I do not fully agree with the conclusion that G was the principal axis, as shown in Figure 4c, second panel, it is now possible to see dispersion of data that seems to be consistent with signalling noise.

We have tried to address this same comment earlier in our response to Reviewer 2.

Line 185 of the main text, onwards. I don't understand why measurements of whole intestine taken from *hsp16* do not carry bleed through technical issues as much as *eft-3*. The argument used to downplay the wider "P" axis is misleading, as *hsp16* promoter should drive expression in many tissues, as much as *eft-3*.

Again, this is due to differences in the percent contribution to total signal from different tissues. For *hsp-16.2*, in adults, and seemingly in images of larvae from Cassanueva et al, 90% of the signal comes from the intestine, and this is not the case for *eft-3*, which has more expression in the hypodermis and muscles. We know precisely where the *hsp-16.2* and *eft-3* reporters are expressed because we have measured them with a confocal microscope, obviating any assertions based on observations with a stereoscope or other relatively lower resolution measurements.

For more explanation on this topic, the reader is directed to reviewer response Figure 3.

A dissecting microscope should not be used to gather data from the intestine only (cut through by using FIJI), because, as they are well aware, it is true that signal from other tissues can contribute to it. However, if the whole animal is measured then all pixels from the image can be gathered. The experiment should have been done with two promoters that are expressed ubiquitously or even using whole animal pixel intensity from *eft-3P:GFP*. For example, *eft-3* and *hsp-16.2*.

Again, we've addressed this in our lengthy response to Reviewer 2.

Their favoured interpretation is that the increase in dispersion in Figure 4C second panel, is caused by technical inadequacies. However, it may be also telling us that expression may be dominated by different sources of variance depending on the tissue type. I do not believe that this interpretation causes their arguments on G necessarily incorrect. I suggest that when mechanisms are not clear, then it is more correct to keep interpretations open. I think that authors should write an article about G as a dominating source of noise in polyploidy tissues in post reproductive animals. It still is meaningful, because at least one of the reporters that show this behaviour is a good predictor of lifespan.

As above, we've attempted to address this comment in a previous part of our comprehensive response to Reviewer 2.

For signaling noise, please also refer to Figure 1 in this reviewer response, and Figure 4g (a global view), Figs S8-10 (some noisy in ring 1 cells) and Supplementary Information section "**S2.5 Two Color Variants Driven by Different Promoters and Pathway Variation**" in the revised manuscript.

Page 16, comment on Figure 4B.

I take the point with regards to Ras GOF and its effect on the reporter (Do they see that the reporter expression goes down in expression compared to controls?, perhaps be worth making a note in figure legends). I think that their incompletely penetrant data should go back to main results.

We are happy that the reviewer appreciates our correlation between G and the expressivity and penetrance of the Ras GOF we examined. One reason we moved this finding to Supplementary Information is because we learned that G during L1 does not always match G measured after heat shock in young adults, and, given that the multivulval phenotype manifests during larval development, were unwilling to make too much of the correlation. Correlations between reporters and phenotypes after

the phenotype has already manifested may suggest that the difference in reporter expression was a reaction and not necessarily causal.

The authors should tone down conclusions, because they have not been able to show that G is constant over developmental time (rather the opposite if one takes earlier stages).

The fact that G may not be consistent over developmental time is the point we just made above. And it is why we toned down the result on Ras and neomycin, moving them to the supplements.

But since they cannot base their conclusions on previously published data,

As we described earlier in our response to Reviewer 2, we are able to interpret previously published data on nulls and hypomorphs. There is no reason we cannot interpret previous scientific results to guide our own studies and interpretations.

and they did not provide fresh data on LOF incomplete penetrance,

As we described earlier in our response to Reviewer 2, we are able to interpret previously published data on nulls and hypomorphs.

then the GOF is what they can use.

Our response is as before. We want to be careful in our interpretation of the RAS GOF data, and leave it in the Supplementary Information where it is now. Doing so freed up space for a new Figures 4e&f in the maintext, whose addition we believe better helps make the point of the paper.

Page 17. I can agree with the publication of an article without proper mechanistic understanding of what G really is about. However, the article must be accurate. In the last paragraph of page 17 and throughout their paper, the continue to say that “high abundance of chaperones correlate with high abundance of other cellular proteins”. If this is what they want to claim, then they have to show that 1. Their measurements of GFP from a hsp16 reporter 24 hours after heat shock accurately represents measurements of chaperones. This should be done using western blots that GFP corresponds with the levels of endogenous chaperones.

See above response to the similar comments about heat shock response with Link et al and measurements of chaperone levels in bright and dim animals from Rea et al.

Previously, within the text of this response to Reviewer 2, we have reviewed the 10+ year history of use of fluorescent signal from chaperone reporters signal as a proxy for actual chaperone levels, and referenced the pre-existing studies that experimentally confirm that relationship. Link et al 1999 showed that HSP-16.2 protein levels peaked at 24 hours post heat shock. Rea et al 2005 showed that more *hsp-16.2* reporter gene equates to more HSP-16.2 protein. Burga et al 2012 showed that more *hsp-90/daf-21* reporter gene equates to more HSP-90. For the most part, reporter genes correlate with the gene they are reporting on, otherwise no one would use reporter genes. While there can be deviations in post transcriptional regulation between native genes and fluorescent proteins, the vast majority of published literature examining the correlation between reporter genes and their proteins to be positive. Again, otherwise, what would be the point. We assert that there are decades of scientific finding that continue to rest firmly on conclusions drawn from reporter genes, including Casanueva et al 2012.

2. They should include HSF1 over-expression experiments, where they pump up the levels of chaperones to determine if RAS GOF depends on Hsf1 activity.

We have not carried out these experiments. One reason we have not is that we are not convinced that they would admit of clean interpretation, that HSF-1 overexpression would result in increased penetrance of Ras by the mechanism described.

In more detail, if increased expression of heat shock chaperones causes increased in global protein expression, or if these merely correlate with increases in global expression, increases in HSF1 activity might increase Ras GOF penetrance. And yet, total penetrance should depend on total Ras activity which results from production and maintenance of Ras proteins. The issue with the HSF-1 overexpression experiments is that in the wild type background, HSF-1 overexpression produces small sized long lived animals with reduced fecundity. Thus, we believe that HSF-1 overexpression likely decreases metabolic rate and rate of protein synthesis, at the same time as it increases protein stability. We therefore feared that the suggested experiment would be indirect, and its results difficult to interpret.

If the authors do not wish to continue doing experiments...

With respect, the authors have performed more than a year of additional experiments to address concerns raised by reviews of a previous manuscript. We believe we have addressed these concerns. Authors now wish to publish these results.

then they should be careful with each and every statement in the article.

Authors believe that, in the revised article, every statement is in fact careful. Certainly, we have tried our best.

I would say that they should treat hsp16:GFP as a biomarker that correlates with lifespan.

Authors believe that they have done so.

They have not shown properly that in their protocol, they are dealing with chaperones.

Insofar as we take this statement to express concern that the GFP chaperone reporters reflect changes in chaperone activity, we have now in this response covered the evidence and this argument, multiple times.

Page 18, on their re-interpretation of the Science 2012, Lehner article. That article does not show molecular evidence that the chaperone refractive alleles are genetic nulls. A careful reading of supplementary figures shows that the alleles that are refractive to Hsf-1 are cold sensitive. There is no genetic evidence for any of such alleles to be true nulls. Second, the authors argument that elevated heat shock increases the penetrance of such alleles, is opposite to what was shown in that article.

Again, please also see the response to the same concern already raised by reviewer 2, listed above, and shown graphically in reviewer response Figure 5.

We disagree, respectfully, on two counts. First, as mentioned previously, for the genetic evidence presented our reasoning supporting the nullitude of the chaperone-refractory alleles, and including the sequences from Wormbase, the *C. elegans* genome sequence database, showing their missing exons. *efn-2(ev568)* lacks the first four exons, and *lin-31(gk569)* lacks the first three. Moreover, we noted that Miller et al 2000, showed that existing alleles of *lin-31* were all null or virtually null, and that anything deleting more than half of the protein (which the *gk569* allele does) would be null. Hence our argument that these alleles are nulls. Second, we note we stated that heat shock increased the activity (and thus decreased the phenotypic effect), not of these mutations, but of other hypomorphic alleles used in the 2012 study.

Revised text now reads:

''While we do not know exactly what kind of variation is happening in larval cells, it is worth reconsidering some of the results from Casanueva et al. on penetrance. In 2012, Casanueva et al. showed that overexpression of HSF-1, which activates expression of chaperones, decreased penetrance of hypomorphic mutations, but not of null mutations. The mutations we believe to be null were listed as ''non-TS/Cold-sensitive''¹⁰; we believe they are almost certainly null because these alleles (*lin-31(gk569)* & *efn-2 (ev568)*) delete most of the exons, leaving only the final exon for each gene (www.wormbase.org).''

References:

- Burga, A., M. O. Casanueva and B. Lehner, 2011 Predicting mutation outcome from early stochastic variation in genetic interaction partners. *Nature* 480: 250-253.
- Casanueva, M. O., A. Burga and B. Lehner, 2011 Fitness trade-offs and environmentally induced mutation buffering in isogenic *C. elegans*. *Science* 335: 82-85.
- Chen, B., S. Li, Q. Ren, X. Tong, X. Zhang and L. Kang, 2015 Paternal epigenetic effects of population density on locust phase-related characteristics associated with heat-shock protein expression. *Mol Ecol* 24: 851-862.
- Colman-Lerner, A., A. Gordon, E. Serra, T. Chin, O. Resnekov, D. Endy, C. G. Pesce and R. Brent, 2005 Regulated cell-to-cell variation in a cell-fate decision system. *Nature* 437: 699-706.
- Crane, M. M., B. Sands, C. Battaglia, B. Johnson, S. Yun, M. Kaeberlein, R. Brent and A. Mendenhall, 2019 In vivo measurements reveal a single 5'-intron is sufficient to increase protein expression level in *Caenorhabditis elegans*. *Scientific Reports* 9: 9192.
- Cypser, J. R., D. Wu, S. K. Park, T. Ishii, P. M. Tedesco, A. R. Mendenhall and T. E. Johnson, 2013 Predicting longevity in *C. elegans*: fertility, mobility and gene expression. *Mech Ageing Dev* 134: 291-297.
- Depuydt, G., N. Shanmugam, M. Rasuloova, I. Dhondt and B. P. Braeckman, 2016 Increased Protein Stability and Decreased Protein Turnover in the *Caenorhabditis elegans* *Ins/IGF-1 daf-2* Mutant. *J Gerontol A Biol Sci Med Sci* 71: 1553-1559.
- Dhondt, I., V. A. Petyuk, H. Cai, L. Vandemeulebroucke, A. Vierstraete, R. D. Smith, G. Depuydt and B. P. Braeckman, 2016 FOXO/DAF-16 Activation Slows Down Turnover of the Majority of Proteins in *C. elegans*. *Cell Rep* 16: 3028-3040.
- Elowitz, M. B., A. J. Levine, E. D. Siggia and P. S. Swain, 2002 Stochastic gene expression in a single cell. *Science* 297: 1183-1186.
- Gadal, O., D. Strauss, J. Braspenning, D. Hoepfner, E. Petfalski, P. Philippsen, D. Tollervey and E. Hurt, 2001 A nuclear AAA-type ATPase (Rix7p) is required for biogenesis and nuclear export of 60S ribosomal subunits. *EMBO J* 20: 3695-3704.
- Gartner, K., 1990 A third component causing random variability beside environment and genotype. A reason for the limited success of a 30 year long effort to standardize laboratory animals? *Lab Anim* 24: 71-77.
- Gimelbrant, A., J. N. Hutchinson, B. R. Thompson and A. Chess, 2007 Widespread monoallelic expression on human autosomes. *Science* 318: 1136-1140.
- Greer, E. L., T. J. Maures, D. Ucar, A. G. Hauswirth, E. Mancini, J. P. Lim, B. A. Benayoun, Y. Shi and A. Brunet, 2011 Transgenerational epigenetic inheritance of longevity in *Caenorhabditis elegans*. *Nature* 479: 365-371.
- Martincorena, I., J. C. Fowler, A. Wabik, A. R. J. Lawson, F. Abascal, M. W. J. Hall, A. Cagan, K. Murai, K. Mahbubani, M. R. Stratton, R. C. Fitzgerald, P. A. Handford, P. J. Campbell, K. Saeb-Parsy and P. H. Jones, 2018 Somatic mutant clones colonize the human esophagus with age. *Science* 362: 911-917.
- McCarter, J., B. Bartlett, T. Dang and T. Schedl, 1999 On the control of oocyte meiotic maturation and ovulation in *Caenorhabditis elegans*. *Dev Biol* 205: 111-128.
- Mendenhall, A., M. M. Crane, S. Leiser, G. Sutphin, P. M. Tedesco, M. Kaeberlein, T. E. Johnson and R. Brent, 2017a Environmental Canalization of Life Span and Gene Expression in *Caenorhabditis elegans*. *J Gerontol A Biol Sci Med Sci*.
- Mendenhall, A., M. M. Crane, P. M. Tedesco, T. E. Johnson and R. Brent, 2017b *Caenorhabditis elegans* Genes Affecting Interindividual Variation in Life-span Biomarker Gene Expression. *J Gerontol A Biol Sci Med Sci*.
- Mendenhall, A. R., P. M. Tedesco, B. Sands, T. E. Johnson and R. Brent, 2015 Single Cell Quantification of Reporter Gene Expression in Live Adult *Caenorhabditis elegans* Reveals Reproducible Cell-Specific Expression Patterns and Underlying Biological Variation. *PLoS One* 10: e0124289.
- Mendenhall, A. R., P. M. Tedesco, L. D. Taylor, A. Lowe, J. R. Cypser and T. E. Johnson, 2012 Expression of a single-copy *hsp-16.2* reporter predicts life span. *J Gerontol A Biol Sci Med Sci* 67: 726-733.
- Raser, J. M., and E. K. O'Shea, 2004 Control of stochasticity in eukaryotic gene expression. *Science* 304: 1811-1814.
- Rea, S. L., D. Wu, J. R. Cypser, J. W. Vaupel and T. E. Johnson, 2005 A stress-sensitive reporter predicts longevity in isogenic populations of *Caenorhabditis elegans*. *Nat Genet* 37: 894-898.
- Sanchez-Blanco, A., and S. K. Kim, 2011 Variable pathogenicity determines individual lifespan in *Caenorhabditis elegans*. *PLoS Genet* 7: e1002047.
- Seewald, A. K., J. Cypser, A. Mendenhall and T. Johnson, 2010 Quantifying phenotypic variation in isogenic *Caenorhabditis elegans* expressing *Phsp-16.2::gfp* by clustering 2D expression patterns. *PLoS One* 5: e11426.

- Terskikh, A., A. Fradkov, G. Ermakova, A. Zaraisky, P. Tan, A. V. Kajava, X. Zhao, S. Lukyanov, M. Matz, S. Kim, I. Weissman and P. Siebert, 2000 "Fluorescent timer": protein that changes color with time. *Science* 290: 1585-1588.
- Visscher, M., S. De Henau, M. H. E. Wildschut, R. M. van Es, I. Dhondt, H. Michels, P. Kemmeren, E. A. Nollen, B. P. Braeckman, B. M. T. Burgering, H. R. Vos and T. B. Dansen, 2016 Proteome-wide Changes in Protein Turnover Rates in *C. elegans* Models of Longevity and Age-Related Disease. *Cell Rep* 16: 3041-3051.
- Yang, J., and J. Tower, 2009 Expression of hsp22 and hsp70 transgenes is partially predictive of drosophila survival under normal and stress conditions. *J Gerontol A Biol Sci Med Sci* 64: 828-838.

REVIEWERS' COMMENTS:

Reviewer #1 (Remarks to the Author):

The authors have addressed my concerns. I think this manuscript should be published as it makes a very interesting and thought-provoking contribution. I do not agree with many of the rather pedantic criticisms of the other referee and think the authors should have the freedom to write their own paper.

I do suggest that the authors change the title to something more general. I understand the change is in response to the referees' comments. But it now almost completely misses the big picture and may lead to the paper being overlooked. The connection to lifespan / penetrance should be in the title.

Reviewer Requests:

REVIEWERS' COMMENTS:

Reviewer #1 (Remarks to the Author):

The authors have addressed my concerns. I think this manuscript should be published as it makes a very interesting and thought-provoking contribution. I do not agree with many of the rather pedantic criticisms of the other referee and think the authors should have the freedom to write their own paper.

I do suggest that the authors change the title to something more general. I understand the change is in response to the referees' comments. But it now almost completely misses the big picture and may lead to the paper being overlooked. The connection to lifespan / penetrance should be in the title.

We thank the reviewer for spending time on another revision. We agree that changing the title to something broader would improve the manuscript and we thank the reviewer for this suggestion.

The new title reads:

"Chaperone biomarkers of lifespan and penetrance track the effective dosages of many other proteins"